# Training Deep Morphological Neural Networks as Universal Approximators

## Abstract

We investigate deep morphological neural networks (DMNNs), studying how changes in algebraic structure affect the expressiveness and trainability of deep architectures. We show that despite the inherent non-linearity of morphological operations, existing deep morphological architectures fail to be universal approximators and exhibit optimization limitations. To address these issues, we introduce architectures incorporating constrained "linear" activations between morphological layers. In the first two architectures, only $O(N)$ parameters (or learnable parameters) per layer of size $N$ belong to the activations, with the remaining parameters constrained to morphological operations. We prove universal approximation results for the proposed architectures and show empirically that they can be successfully trained on standard image classification tasks. Residual connections and weight dropout further improve generalization. Our experiments show that our networks are trainable, without requiring substantially larger parameter counts than comparable linear networks despite the imposed architectural restrictions. Finally, we propose a hybrid linear/morphological architecture and observe accelerated convergence of gradient descent under large batches.

## 1 Introduction

Modern deep learning architectures are predominantly built on linear operations, such as matrix multiplications and convolutions, combined with nonlinear activations (Goodfellow, 2016). This raises a broader question: how do the expressiveness, optimization, and representation properties of deep networks change when the underlying algebraic structure is modified? Recent interest in binary (Yuan & Agaian, 2023) and other non-standard neural architectures reflects growing interest in understanding neural computation beyond the standard linear setting.

Before deep learning, mathematical morphology (Haralick et al., 1987; Heijmans, 1994; Maragos, 1999; 2005; Najman & Talbot, 2010; Serra, 1982; 1988; Soille, 2004) played a central role in image and signal processing. Its operations – dilations and erosions – enabled effective, task-specific feature extraction using max-plus and min-plus algebra. Its historical success makes mathematical morphology a natural framework for studying these questions. Motivated by this, researchers developed models based on morphological operations. One of the earliest was the *Morphological Perceptron (MP)*, which replaces addition and multiplication with max and plus, enabling nonlinear decision boundaries and forming the basis of morphological neural networks (Davidson & Hummer, 1993; Pessoa & Maragos, 2000; Ritter & Sussner, 1996; Ritter & Urcid, 2003; Sussner & Esmi, 2011; Yang & Maragos, 1995). These models offer appealing properties, including compressibility (Dimitriadis & Maragos, 2021; Groenendijk et al., 2023; Zhang et al., 2019) and fast training. The Dilation-Erosion Perceptron (DEP) (de A. Araújo, 2011), trained using CCP (Charisopoulos & Maragos, 2017), blends dilation and erosion via a convex combination.

Despite their advantages, both the MP and DEP have a limitation, which restricts their applicability: Their decision boundaries are axis-aligned (Ritter & Sussner, 1996; Yang & Maragos, 1995). To address this, researchers have proposed approaches to transform the input space into one that is MP-separable or DEP-separable using kernel transformations. For example, Valle (2020) proposed a simple linear transformation as a kernel to map features into a more favorable space.

This idea of mapping inputs to a separable space is not new – it is the foundation of deep learning. This motivates the study of whether deep morphological networks (DMNNs), built on the MP and its variants, can similarly learn effective representations using morphological operations.

Recent works have explored this question with varying levels of success (Hu et al., 2022; Groenendijk et al., 2023; Shen et al., 2022). Notable recent efforts, related to this paper, include the works of Franchi et al. (2020), who demonstrated the potential of deep hybrid morphological-linear networks, Dimitriadis & Maragos (2021), who demonstrated the amenability to pruning of morphological networks, and Angulo-Lopez (2024); Velasco-Forero & Angulo (2022) who proposed representing the combined convolutions, nonlinear activation and max-pooling in ConvNets as a max of erosions or min of dilations by leveraging the morphological representation theory (Maragos, 1989).

However, despite these advancements, most successful approaches rely heavily on standard linear components, making it difficult to isolate the role of the underlying algebraic structure. In this paper, we study, both theoretically and empirically, what minimal algebraic "augmentations" are required for deep morphological networks to achieve expressive and trainable representations. Our contributions can be summarized as follows:

**Contributions.**   In this paper, we address the challenges of training deep morphological networks.

- We show that several existing deep morphological architectures fail to be universal approximators despite their inherent non-linearity, and identify optimization limitations arising from their operations.
- Motivated by these obstructions, we introduce deep morphological architectures incorporating constrained linear activations between morphological layers. In the first two architectures, only $O(N)$ parameters (or learnable parameters) per layer of size $N$ belong to the activations, with the remaining parameters constrained to morphological operations.
- We prove universal approximation results for the deterministic architectures and show empirically that they can be successfully trained on standard image classification tasks despite the imposed architectural constraints.
- We improve the generalization ability of the proposed networks using residual connections and weight dropout.
- We empirically show that the proposed architectures do not require substantially larger parameter counts than comparable linear networks to achieve expressivity in our fully connected pruning experiments.
- Finally, we propose a hybrid linear/morphological architecture and observe that it benefits from large batches with respect to its training convergence under gradient descent.

## 2   Prerequisites

**Tropical Algebra.**   Tropical algebra is a branch of mathematics focused on the study of the tropical semiring, which encompasses both the *min-plus semiring* and the *max-plus semiring* (Butkovič, 2010; Cuninghame-Green, 1979; Maclagan & Sturmfels, 2021; Maragos et al., 2021). The *max-plus semiring*, denoted as $(\mathbb{R}_{\max}, \vee, +)$, consists of the set $\mathbb{R}_{\max} = \mathbb{R} \cup \{-\infty\}$, equipped with two binary operations: $\vee$ (the maximum operator) and $+$ (ordinary addition). Similarly, the *min-plus semiring*, denoted as $(\mathbb{R}_{\min}, \wedge, +)$, is defined over the set $\mathbb{R}_{\min} = \mathbb{R} \cup \{+\infty\}$, with $\wedge$ (the minimum operator) and $+$ as its binary operations. These semirings naturally extend to operations on vectors and matrices. For instance, the *max-plus matrix multiplication* $\boxplus$ and the *min-plus matrix multiplication* $\boxplus'$ for matrices $\mathbf{A} = [a_{ij}]$ and $\mathbf{B} = [b_{ij}]$ is defined as:

$$(\mathbf{A} \boxplus \mathbf{B})_{ij} = \bigvee_k (a_{ik} + b_{kj}), \quad (\mathbf{A} \boxplus' \mathbf{B})_{ij} = \bigwedge_k (a_{ik} + b_{kj}).$$

**Mathematical morphology.**   Modern mathematical morphology (Heijmans, 1994; Serra, 1988) is defined on complete lattices. A partially ordered set $(\mathbb{L}, \preceq)$ is a complete lattice (Birkhoff, 1967) if and only if (iff) every subset $X \subseteq \mathbb{L}$ has a supremum and an infimum, denoted by $\bigvee X$ and $\bigwedge X$ respectively. Consider two complete lattices $\mathbb{L}$ and $\mathbb{M}$. A lattice operator $\delta : \mathbb{L} \to \mathbb{M}$ is called a *dilation* iff it distributes over the supremum of any input (possibly infinite) collection. Dually, an *erosion* is defined as any lattice operator

$\varepsilon : \mathbb{L} \to \mathbb{M}$ that distributes over the infimum. Namely, the following properties hold:

$$\delta \left( \bigvee X \right) = \bigvee \delta(X), \quad \varepsilon \left( \bigwedge X \right) = \bigwedge \varepsilon(X), \quad \forall X \subseteq \mathbb{L}$$

Let $\overline{\mathbb{R}} = \mathbb{R} \cup \{+\infty, -\infty\}$ denote the extended set of real numbers, which is a complete lattice when equipped with its ordinary order. The set $\overline{\mathbb{R}}^n$ of vectors equipped with the partial order $\mathbf{x} \preceq \mathbf{y} \Leftrightarrow x_i \leq y_i, \forall i \in [n]$ forms a complete lattice. There exists a natural way to define a dilation and an erosion from $\overline{\mathbb{R}}^n$ to $\overline{\mathbb{R}}$: Given an input vector $\mathbf{x} \in \overline{\mathbb{R}}^n$ and weights $\mathbf{w}, \mathbf{m} \in \mathbb{R}^n$, a vector dilation $\delta_{\mathbf{w}}$ and erosion $\varepsilon_{\mathbf{m}}$ are defined as follows:

$$\delta_{\mathbf{w}}(\mathbf{x}) = \bigvee_{i \in [n]} (x_i + w_i) = \mathbf{w}^\top \boxplus \mathbf{x}, \quad \varepsilon_{\mathbf{m}}(\mathbf{x}) = \bigwedge_{i \in [n]} (x_i + m_i) = \mathbf{m}^\top \boxplus' \mathbf{x}.$$

The Morphological Perceptron (MP) is simply a dilation (or erosion) as defined above, optionally biased with an addition bias term (i.e. $\mathrm{MP}(\mathbf{x}) = w_0 \vee \delta_{\mathbf{w}}(\mathbf{x})$ for a dilation-based MP). Note that if $\mathbf{x} < +\infty$, then $\mathbf{w}$ is allowed to take the value $-\infty$, and dually if $\mathbf{x} > -\infty$, then $\mathbf{m}$ can take the value $+\infty$. The DEP is a convex combination of a dilation and an erosion (i.e. $\mathrm{DEP}_{\mathbf{w},\mathbf{m}}(\mathbf{x}) = \lambda \delta_{\mathbf{w}}(\mathbf{x}) + (1 - \lambda)\varepsilon_{\mathbf{m}}(\mathbf{x})$, for $\lambda \in [0, 1]$).

Mathematical morphology also employs dilations and erosions defined on sets of functions, which naturally model images. The set $\mathbb{F} = \{f : \mathbb{Z}^n \to \overline{\mathbb{R}}\}$ of extended real-valued functions over the $n$-dimensional grid of integers, equipped with the partial ordering $f \preceq g \Leftrightarrow f(\mathbf{x}) \leq g(\mathbf{x}) \ \forall \mathbf{x}$, becomes a complete lattice. On this set, we can define the *max-plus convolution* $\oplus$ and *min-plus convolution* $\oplus'$ of a function $f \in \mathbb{F}$ with a structuring element (function) $g : \mathbb{Z}^n \to \mathbb{R} \cup \{-\infty\}$, whose domain is $\mathbf{dom}(g) = \{\mathbf{x} : g(\mathbf{x}) > -\infty\}$, as follows:

$$(f \oplus g)(\mathbf{x}) = \bigvee_{\mathbf{y} \in \mathbf{dom}(g)} f(\mathbf{x} - \mathbf{y}) + g(\mathbf{y}), \quad (f \oplus' g)(\mathbf{x}) = \bigwedge_{\mathbf{y} \in \mathbf{dom}(g)} f(\mathbf{x} - \mathbf{y}) - g(\mathbf{y}).$$

The max-plus convolution and $\delta_g(f) = f \oplus g$ and min-plus correlation $\varepsilon_g(f)(\mathbf{x}) = f(\mathbf{x}) \oplus' g(-\mathbf{x})$ are a dilation and erosion on $\mathbb{F}$. By combining these dual morphological operators, we can construct more complex operators such as *opening* $\alpha_g(f) = \delta_g(\varepsilon_g(f))$ and *closing* $\beta_g(f) = \varepsilon_g(\delta_g(f))$. The opening can smooth the input by removing small bright structures, while the closing can fill gaps and remove small dark structures (Serra, 1982; Maragos, 2005).

The *Representation Theorem* of Maragos & Schafer (1987) takes this one step further: it proves that we can represent any increasing, translation-invariant linear shift-invariant filter as a supremum of weighted erosions. A byproduct of this is that we can write any linear perceptron with positive weights that sum up to 1 as a supremum of weighted erosions: If $\alpha_i > 0$ for all $i$ and $\sum_i \alpha_i = 1$, then the following identity holds:

$$\sum_{i=0}^n \alpha_i x_i = \sup_{r_0,\dots,r_{n-1} \in \mathbb{R}} \left[ \min \left\{ x_0 - r_0, \dots, x_{n-1} - r_{n-1}, \ x_n + \frac{\sum_{i=0}^{n-1} \alpha_i r_i}{\alpha_n} \right\} \right].$$

If we allow biased erosions we can relax the condition that the weights must sum up to 1, as long as they sum up to less than 1. For example, we have $\frac{x}{2} = \sup_{r \in \mathbb{R}} \min(x - r, r)$. For details regarding the Representation Theorem, see Appendix E.

## 3 Deep morphological neural networks

Defining deep morphological neural networks (DMNNs) involves combining the fundamental operations of mathematical morphology. In this section, we present the most common DMNNs, and showcase their fundamental problems. Then, we present our proposed architectures under differing set of constraints. Throughout, a superscript $(\cdot)^{(n)}$ is used to indicate when something corresponds to the $n$-th layer, and $N^{(n)}$ refers to the number of units in the $n$-th layer. We use $[k] := \{1, \dots, k\}$. Proofs of our theorems can be found in Appendices A and B.

Existing morphological architectures often compose max-plus or related morphological operators either directly or with standard nonlinear activations (Dimitriadis & Maragos, 2021; de A. Araújo et al., 2017; Franchi et al., 2020; Groenendijk et al., 2023). However, as we show below, such constructions suffer from fundamental expressivity limitations that can already be understood from basic principles of tropical algebra.

**Max-plus MP-based DMNNs.** The most obvious way to building a DMNN is by appending morphological layers consisting of several MPs. Suppose we have $L$ morphological layers, with the $n$-th layer having $N^{(n)}$ biased MPs. We can then write the network in recursive form:

$$x_i^{(n)} = w_{i0}^{(n)} \vee \max_{j \in [N^{(n-1)}]} (x_j^{(n-1)} + w_{ij}^{(n)}),$$

with $\mathbf{x}^{(0)}$ being the input and $\mathbf{x}^{(L)}$ the output.

As mentioned, $(\mathbb{R}_{\max}, \vee, +)$ is a semiring. This means that we have distributivity of $+$ over $\vee$, and associativity of $+$. These properties are enough to prove the associativity of matrix multiplication, i.e. for every matrices $\mathbf{A}, \mathbf{B}, \mathbf{C}$ over $\mathbb{R}_{\max}$, we have $(\mathbf{A} \boxplus \mathbf{B}) \boxplus \mathbf{C} = \mathbf{A} \boxplus (\mathbf{B} \boxplus \mathbf{C}) = \mathbf{A} \boxplus \mathbf{B} \boxplus \mathbf{C}$.

Let $L \geq 2$. The recursive form of the network can be expressed as:

$$\mathbf{x}^{(n)} = \mathbf{w}_0^{(n)} \vee \mathbf{W}^{(n)} \boxplus \mathbf{x}^{(n-1)}, \quad n \in [L],$$

where $\mathbf{w}_0^{(n)} = [w_{10}^{(n)}, \dots, w_{N^{(n)}0}^{(n)}]^\top \in \mathbb{R}^{N^{(n)}}$ is the bias vector, and $\mathbf{W}^{(n)} = (w_{ij}^{(n)})_{ij} \in \mathbb{R}^{N^{(n)} \times N^{(n-1)}}$ is the weight matrix.

We can solve the recursive form for $\mathbf{x}^{(L)}$ by unfolding the recursion, obtaining:

$$\mathbf{x}^{(L)} = \mathbf{w}_{eq,0} \vee \mathbf{W}_{eq} \boxplus \mathbf{x}^{(0)},$$

where

$$\mathbf{w}_{eq,0} = \mathbf{w}_0^{(L)} \vee \bigvee_{k=1}^{L-1} \left( \mathbf{W}^{(L)} \boxplus \mathbf{W}^{(L-1)} \boxplus \cdots \boxplus \mathbf{W}^{(L-(k-1))} \boxplus \mathbf{w}_0^{(L-k)} \right)$$

$$\mathbf{W}_{eq} = \mathbf{W}^{(L)} \boxplus \mathbf{W}^{(L-1)} \boxplus \cdots \boxplus \mathbf{W}^{(1)}.$$

This expanded form shows that the network with $L \geq 2$ is equivalent to a network with a single morphological layer, which is not a universal approximator. Hence, these types of networks require some sort of activation. For example in (Dimitriadis & Maragos, 2021), their "$\delta$ network" has two max-plus MP layers, which effectively reduce to one.

**ASFs and layers combining max-plus and min-plus MPs.** One way to add complexity to the network is by including min-plus MP units. This can be done in two main ways: either by alternating between max-plus MP layers and min-plus MP layers, resulting in what is known as an Alternating Sequential Filter (ASF) (Serra, 1988), or by incorporating both max-plus and min-plus MPs within the same layer, as proposed by Mondal et al. (2019); Dimitriadis & Maragos (2021). While these architectures appear more complex, they were not developed to address the fundamental limitations of max-plus MP-based networks. In fact, we demonstrate that even with the inclusion of min-plus MPs, these networks remain limited in their representational power without the use of activations. By tracing the results of the max and min operations backwards, we can find a dependence of the output on the input and the parameters of the network. We prove the following theorems.

**Theorem 1.** *For any network that **only** uses max-plus and min-plus MPs with input $\mathbf{x} \in \mathbb{R}^d$ and a single output $y(\mathbf{x})$, we have that $y(\mathbf{x})$ is Lipschitz continuous on $\mathbb{R}^d$ and a.e. it holds that either $\nabla y(\mathbf{x}) = 0$ or $\nabla y(\mathbf{x}) = \mathbf{e}_i = [0, \dots, 1, \dots, 0]^\top$ for some $i = i(\mathbf{x})$.*

This result implies that such networks cannot be universal approximators. Consider, for instance, the function $f(\mathbf{x}) = 2x_1$, whose gradient is $\nabla f = 2\mathbf{e}_1 = [2, 0, \dots, 0]^\top$. Since the networks described above can only produce gradients of the form $\mathbf{e}_i$, they are incapable of representing even this simple linear function. For details, refer to Theorem A.5.

Theorem 1 does not contradict the Representation Theorem; rather, it complements it and is motivated by its hypotheses. One of the byproducts of Theorem 1 is that we can represent a linear shift-invariant filter as a supremum of weighted erosions exactly when the filter is increasing and translation-invariant. For details, refer to Appendix E.

**Theorem 2.** *Consider a network that **only** uses max-plus and min-plus MPs with output $\mathbf{y} \in \mathbb{R}^m$. For any given input $\mathbf{x}$, if $\mathbf{y}$ is differentiable with respect to the network parameters, then in each layer $n$, there exists at most $m$ parameters $w_{ij}^{(n)}$ for which the derivative of $\mathbf{y}$ is nonzero.*

This result highlights a fundamental limitation in training such networks: the sparsity of their gradient signals during backpropagation. Consequently, gradient-based optimization methods become highly inefficient.

**DEP-based DMNNs.** Another way of building a DMNN is by using DEP units in each layer (de A. Araújo et al., 2017). For a network with $L$ layers we can write it in recursive form as follows:

$$x_i^{(n)} = f^{(n)}(\lambda_i^{(n)} \max_{j \in [N^{(n-1)}]}(x_j^{(n-1)} + w_{ij}^{(n)}) + (1 - \lambda_i^{(n)}) \min_{j \in [N^{(n-1)}]}(x_j^{(n-1)} + m_{ij}^{(n)})),$$

where $\lambda_i^{(n)} \in [0,1], \forall i \in [N^{(n)}], n \in [L]$, and $f^{(n)}$ a common activation function or identity.

On first sight, this network, a generalization of the previous networks, looks like it solves both problems. By taking the sum, working backwards the gradient disperses in two paths and we have diffusion of the gradient across more parameters and more inputs. In addition, by multiplying the two terms by $\lambda_i^{(n)}$ and $1 - \lambda_i^{(n)}$ we have more diversity in the values of the gradient. However, while it indeed solves the problem of sparse gradients, this network is too not a universal approximator, and in fact very limited in the range of functions it can represent, as the following theorem suggests.

**Theorem 3.** *For existing DEP-based networks with input $\mathbf{x} \in \mathbb{R}^d$ and a single output $y(\mathbf{x})$, we have that $y(\mathbf{x})$ is Lipschitz continuous on $\mathbb{R}^d$ and a.e. it holds that $\nabla y(\mathbf{x}) \succeq 0, \|\nabla y(\mathbf{x})\|_1 \leq 1$.*

This result implies that existing DEP-based networks cannot be universal approximators. To see this, we can consider again the example of the function $f(x) = 2x_1$. For details, refer to Theorem A.7.

In addition to the above problem, DEP-based networks also showcase another problem. The existence of $\lambda_i^{(n)}$ terms makes training slow. To understand why, suppose that at some point, either after initialization or during training, we have $\lambda_i^{(n)} \neq 1/2$. Then, the weights $\mathbf{w}_i^{(n)}, \mathbf{m}_i^{(n)}$ will have to follow distributions with different means in order for the output to be zero-mean. In addition, every time $\lambda_i^{(n)}$ changes, in order for the output to remain zero-mean, the mean of the distributions of $\mathbf{w}_i^{(n)}, \mathbf{m}_i^{(n)}$ has to change, which is very slow in morphological networks due to sparse gradients. In general, having $\lambda \neq 1/2$, fixed or not, hinders trainability. For a detailed discussion we refer the reader to Appendix C.

**Our proposal.** We propose solutions to the above problems that can be summarized as follows:

1. We introduce learnable "linear" activations between the morphological layers of the network. Depending on our constraints, the complexity of the "linear" activations varies.
2. We change the DEP-based architecture in the following ways:
    (a) We use the same weights for the maximum and the minimum.
    (b) We introduce biases.
    (c) We remove the learnable parameters $\lambda_i^{(n)}$ and simply take the sum of the maximum and the minimum.

We develop networks under three different constraint **settings** based on the allocation of parameters:

1. Each layer $n$ of size $N^{(n)}$ has at most $O(N^{(n)})$ parameters allocated for activations, with the remainder constrained to morphological operations.
2. Each layer $n$ of size $N^{(n)}$ has at most $O(N^{(n)})$ *learnable* parameters allocated for activations, with the remainder constrained to morphological operations.
3. No constraints are imposed on the number of parameters used in the "linear" activations.

**Setting 1.** In the most restrictive setting, we define a fully connected network, which we refer to as the *Max-Plus-Min* (MPM) network. The network is recursively defined as follows:

$$x_i^{(n)} = \alpha_i^{(n)}\left(\left(w_{i0}^{(n)} \vee \max_{j \in [N^{(n-1)}]}(x_j^{(n-1)} + w_{ij}^{(n)})\right) + \left(m_{i0}^{(n)} \wedge \min_{j \in [N^{(n-1)}]}(x_j^{(n-1)} + w_{ij}^{(n)})\right)\right).$$

Or, equivalently, as follows:

$$\mathbf{x}^{(n)} = \mathrm{diag}([\alpha_i^{(n)}]_{i \in N^{(n)}}) \left( \left( \mathbf{w}_0^{(n)} \vee \mathbf{W}^{(n)} \boxplus \mathbf{x}^{(n-1)} \right) + \left( \mathbf{m}_0^{(n)} \wedge \mathbf{W}^{(n)} \boxplus' \mathbf{x}^{(n-1)} \right) \right).$$

Here, our proposed morphological layer is the addition of a max-plus MP layer and a min-plus MP layer sharing the same learnable weights $\mathbf{W}^{(n)} \in \mathbb{R}^{N^{(n)} \times N^{(n-1)}}$ but with different learnable biases $\mathbf{w}_0^{(n)}, \mathbf{m}_0^{(n)} \in \mathbb{R}^{N^{(n)}}$. The activation function is a simple scaling operation: after computing the sum of the maximum and minimum, each output $x_i^{(n)}$ is multiplied by a learnable parameter $\alpha_i^{(n)} \in \mathbb{R}$. The final layer is not activated, i.e. $\alpha_i^{(L)}$ is fixed to 1.

For a layer $n$ of size $N^{(n)}$, the activation function introduces only $N^{(n)} \in O(N^{(n)})$ additional parameters, with all remaining parameters dedicated to morphological operations. The presence of learnable parameters $\alpha_i^{(n)}$ and the inclusion of bias terms allow us to establish the following result:

**Theorem 4.** *If the domain of the input is compact (i.e., bounded and closed), the Max-Plus-Min (MPM) network is a universal approximator.*

If we had no biases and the max-plus and min-plus weights were different, then this formulation would be equivalent to a DEP architecture with fixed parameters $\lambda_i^{(n)} = 0.5$ for all $n, i$, followed by a learnable scaling for each output. In the Experiments, we show that: i) by introducing biases and sharing the same weights we get slightly better training accuracy and generalization, and ii) both introducing the learnable scaling and taking $\lambda_i^{(n)} = 0.5$ are essential for the network to be trainable. The importance of learnable scaling is highlighted in the 1D regression toy examples of Fig. 6 in the Appendix.

With universal approximation being a property in the limit, the natural questions that arise from Theorem 4 are i) whether the MPM requires a prohibitively large number of parameters to achieve its stated expressivity guarantee, and more broadly ii) what is the inductive bias of the MPM. In this regard:

- The proof of Theorem 4 reduces universal approximation on bounded domains to the approximation of piecewise affine functions. In particular, for input dimension $d$, the construction implies that the MPM can represent any maxout function considered in the universality proof of Goodfellow et al. (2013) using at most $O(\log d)$ layers of width $O(dK)$, where $K$ denotes the rank of the corresponding maxout construction, which uses $\Theta(dK)$ parameters. However, the construction of Theorem 4 only requires each MPM unit to have $O(1)$ active inputs. Hence, the MPM represents the same function using at most $O(d \log d \cdot K)$ active weights, incurring only an additional $O(\log d)$ factor in active parameters relative to the corresponding maxout construction.
- In practice, although the input dimension $d$ may be large, the complexity of the target function often dominates the approximation problem, and the $K$ needed for a good approximation dominates $d$. The hierarchical construction underlying Theorem 4 suggests an inductive bias in which the slopes, constructed progressively, share intermediate results, potentially allowing reduction of the width $K$ required in practice and compensating for the additional $O(\log d)$ factor relative to the corresponding maxout construction.
- We empirically investigate this behavior through aggressive pre-training pruning experiments. The results suggest that the MPM preserves its performance without requiring a blow-up of active parameters.
- More broadly, the construction underlying Theorem 4 reveals that MPM units are decomposable. Through suitable parameter choices, for bounded inputs MPM units can emulate max-plus or min-plus MPs, and under restricted discrete inputs and weights, they reduce to AND/OR logical operations over subsets of the inputs. Scaling allows modeling of logical negation. This suggests links between deep morphological networks and broader classes of neural architectures based on discrete or non-standard computational structures.
- Finally, activated MPM layers can be interpreted through the lens of tropical geometry. Rank-$K$ maxout units over $d$ inputs correspond to tropical polynomials formed as a maximum of $K$ affine functions, and differences of maxout units correspond to tropical rational mappings. Since the parameter count of a maxout unit scales as $O(Kd)$, practical architectures typically keep $K$ small. ReLU units are an extreme case in which only a single non-zero affine piece is learned. By contrast, activated morphological layers impose the opposite structural constraint. A diagonal scaling followed by a max-plus layer corresponds to a tropical polynomial composed of many affine pieces, each restricted to depend on only a single input.

Combining max-plus and min-plus components then yields a restricted tropical rational mapping. From this perspective, ReLU/maxout architectures constrain the number of affine pieces while allowing each piece to remain fully expressive, whereas activated morphological layers constrain the complexity of each affine piece while allowing many such pieces to participate in the representation.

**Improving generalization.** The Max-Plus-Min (MPM) network struggles with generalization. To mitigate this, we introduce the Residual-Max-Plus-Min (RMPM) network, which incorporates residual connections for layers where input and output dimensions match. Specifically, we add the input to the activated output before propagating it to the next layer. This residual mechanism accelerates training and slightly improves generalization.

**Corollary 5.** *If the domain of the input is compact (i.e., bounded and closed), and assuming no residual skip connection for the output layer, the Residual-Max-Plus-Min (RMPM) network is a universal approximator.*

To further improve generalization, we encourage the RMPM to learn robust representations by introducing *weight dropout* during training – randomly dropping a percentage of weights during training forward passes.

**Setting 2.** In Setting 1, the activation function is effectively a linear transformation with a learnable diagonal matrix. However, a richer class of linear transformations is desirable to enhance gradient diffusion. In Setting 2, our goal is to achieve this expressivity while maintaining a count of *learnable* activation parameters of at most $O(N^{(n)})$ per layer. Additionally, we prefer that the transformation be full-rank to maximize information propagation.

This motivates our second architecture, the *Max-Plus-Min-SVD* (MPM-SVD) network, defined as follows:

$$\mathbf{y}^{(n)} = \left(\mathbf{w}_0^{(n)} \vee \mathbf{W}^{(n)} \boxplus \mathbf{x}^{(n-1)}\right) + \left(\mathbf{m}_0^{(n)} \wedge \mathbf{W}^{(n)} \boxplus' \mathbf{x}^{(n-1)}\right),$$

$$\mathbf{x}^{(n)} = U^{(n)} \mathrm{diag}(\sigma_1^{(n)}, \ldots, \sigma_{N^{(n)}}^{(n)})(V^{(n)})^\top \mathbf{y}^{(n)}.$$

Again, $\mathbf{W}^{(n)} \in \mathbb{R}^{N^{(n)} \times N^{(n-1)}}$ and $\mathbf{w}_0^{(n)}, \mathbf{m}_0^{(n)} \in \mathbb{R}^{N^{(n)}}$ are learnable morphological parameters. $\{\sigma_i^{(n)}\}_i \in \mathbb{R}^{N^{(n)}}$ are learnable scaling parameters, while $U^{(n)}, V^{(n)} \in \mathbb{R}^{N^{(n)} \times N^{(n)}}$ are **fixed**, random orthonormal matrices. These matrices are initialized as follows: we sample a matrix using Glorot initialization (Glorot & Bengio, 2010) and compute its singular value decomposition (SVD) to obtain $U, V$, ensuring a well-conditioned transformation. While the number of learnable scaling parameters per layer remains $N^{(n)} \in O(N^{(n)})$, the total parameter count, including the fixed matrices $U$ and $V$, scales as $\Theta((N^{(n)})^2)$. The final layer is not activated, i.e. $\sigma_i^{(L)}$ are fixed to 1 and $U^{(L)} = V^{(L)} = I_{N^{(L)}}$.

At this point we should emphasize that networks such as those of Mondal et al. (2019); Dimitriadis & Maragos (2021); Valle (2020) incorporate fully connected linear layers and do not fall in Settings 1 or 2, but in Setting 3. To the best of our knowledge, we are the first to provide trainable networks in settings as restrictive as 1 and 2.

**Setting 3.** In this setting, we impose no constraints on the number of parameters in the linear transformations. The proposed network follows a hybrid architecture, alternating between linear and our proposed morphological layers. We refer to this model as the Hybrid-MLP, defined as follows:

$$\mathbf{y}^{(n)} = \mathbf{A}^{(n)} \mathbf{x}^{(n-1)} + \mathbf{b}^{(n)},$$

$$\mathbf{x}^{(n)} = \left(\mathbf{w}_0^{(n)} \vee \mathbf{W}^{(n)} \boxplus \mathbf{y}^{(n)}\right) + \left(\mathbf{m}_0^{(n)} \wedge \mathbf{W}^{(n)} \boxplus' \mathbf{y}^{(n)}\right),$$

Here, $\mathbf{A}^{(n)} \in \mathbb{R}^{N^{(n)} \times N^{(n-1)}}, \mathbf{b}^{(n)} \in \mathbb{R}^{N^{(n)}}, \mathbf{W}^{(n)} \in \mathbb{R}^{N^{(n)} \times N^{(n)}}, \mathbf{w}_0^{(n)} \in \mathbb{R}^{N^{(n)}}, \mathbf{m}_0^{(n)} \in \mathbb{R}^{N^{(n)}}$ are all trainable parameters. The Hybrid-MLP is effectively an MLP whose ReLU activations have been replaced with our proposed morphological layers. We should note that the biases $\mathbf{b}^{(n)}$ are theoretically not necessary.

**Theorem 6.** *If the domain of the input is compact (i.e., bounded and closed), the Hybrid-MLP is a universal approximator. In fact, any fully connected ReLU or maxout network is a special case of the Hybrid-MLP.*

**Convolutional networks.** So far we have focused on building fully connected DMNNs. We can extend our insights from these networks and build convolutional networks. Convolutional networks will be based on the morphological convolution. The morphological layer we propose takes the sum of a dilation and an erosion with shared weights and different biases. For Setting 1, for a layer with $N^{(n)}$ output channels, we activate the layer by linearly convoluting each channel with a learnable $3 \times 3$ matrix, obtaining $9N^{(n)} \in O(N^{(n)})$ parameters in total. For Setting 2, we activate the layer by a linear convolutional layer, which has weight matrix $A^{(n)} \in \mathbb{R}^{N^{(n)} \times N^{(n)} \times 3 \times 3}$, initialized according to Glorot. For each $i, j \in [3]$ we write $A_{:,:,i,j}^{(n)} = U_{i,j}^{(n)} \Sigma_{i,j}^{(n)} (V_{i,j}^{(n)})^\top$, fix $U_{i,j}^{(n)}, V_{i,j}^{(n)}$, and take $\Sigma_{i,j}^{(n)}$ to be a learnable diagonal matrix, obtaining $9N^{(n)} \in O(N^{(n)})$ learnable parameters. For Setting 3, we alternate between linear convolutional layers and our proposed morphological convolutional layers.

## 4 Experiments

To test the efficacy of our networks, we conduct experiments on the MNIST, Fashion-MNIST, (Deng, 2012; Xiao et al., 2017), and CIFAR-10 (Krizhevsky & Hinton, 2009) datasets. Unless otherwise stated, we use a batch size of 64, a random 80/20 train-validation split, and the Adam optimizer (Kingma & Lei Ba, 2015) with a learning rate of 0.001 for 50 epochs. Our loss function is the Cross Entropy Loss. Throughout, we report mean and std of accuracy for different runs. The networks were initialized according to the remarks of Appendix C. For additional experimental details, additional experiments, and number of parameters of each model, refer to Appendix D.

### 4.1 Fully Connected Networks

In this section, we evaluate various fully connected networks on the MNIST and Fashion-MNIST datasets. Our primary objectives are: i) to demonstrate that our proposed networks are trainable, and ii) to highlight the necessity of our proposed modifications through systematic ablation. We train the following networks:

- **MLP**: A standard ReLU-activated multilayer perceptron.
- **MP**: A max-plus MP-based DMNN.
- **DEP**: A non-activated DEP-based DMNN.
- **DEP** ($\lambda = 1/2$): A non-activated DEP-based DMNN with fixed $\lambda = 1/2$.
- **Act-MP**: A max-plus MP-based DMNN with activation applied according to our proposed method in Setting 1.
- **Act-DEP**: A DEP-based DMNN with learnable $\lambda_i^{(n)}$, activated according to our proposed method in Setting 1.
- **Act-DEP** ($\lambda = 3/4$): A DEP-based DMNN with fixed $\lambda_i^{(n)} = 3/4$, activated according to our proposed method in Setting 1.
- **Act-DEP** ($\lambda = 1/2$): A DEP-based DMNN with fixed $\lambda_i^{(n)} = 1/2$, activated according to our proposed method in Setting 1.
- **MPM**: Our proposed Max-Plus-Min network for Setting 1.
- **RMPM**: Our proposed Residual-Max-Plus-Min network for Setting 1.
- **RMPM-Drop**: An RMPM with 0.3 weight dropout, trained for 200 epochs.
- **MPM-SVD**: Our proposed Max-Plus-Min-SVD network for Setting 2.

All networks consist of 5 hidden layers of size 256. Dropout makes convergence slower, hence RMPM-Drop was trained for 200 epochs. The results are reported in Tables 1a, 1b. As expected, learnable "linear" activation is crucial for training these networks, with activated networks consistently outperforming their non-activated counterparts. Moreover, both MPM and Act-DEP ($\lambda = 1/2$) outperform the models with learnable $\lambda$ and fixed $\lambda = 3/4$, highlighting the importance of summing the maximum and the minimum for trainability.

MPM further benefits from biases and using the same weights for both the dilation and the erosion, resulting in a slight edge in training accuracy and generalization. The networks Act-DEP ($\lambda = 1/2$), MPM, RMPM, RMPM-Drop, and MPM-SVD successfully train (i.e. reach satisfactory train accuracy peaks and

Table 1: Train (peak) and test accuracy of fully connected networks.

(a) Accuracies on MNIST.

| Network | Train (%) | Test (%) |
|---|---|---|
| MLP | $99.88 \pm 0.04$ | $98.01 \pm 0.08$ |
| MP | $31.24 \pm 1.51$ | $31.59 \pm 1.28$ |
| DEP | $76.63 \pm 3.46$ | $76.51 \pm 3.36$ |
| DEP ($\lambda = 1/2$) | $76.96 \pm 1.13$ | $77.64 \pm 1.03$ |
| Act-MP | $66.17 \pm 14.17$ | $65.27 \pm 13.97$ |
| Act-DEP | $84.51 \pm 1.16$ | $84.15 \pm 1.03$ |
| Act-DEP ($\lambda = 3/4$) | $94.04 \pm 0.37$ | $92.29 \pm 0.64$ |
| Act-DEP ($\lambda = 1/2$) | $99.15 \pm 0.23$ | $94.43 \pm 0.26$ |
| MPM | $99.82 \pm 0.04$ | $94.66 \pm 0.13$ |
| RMPM | $99.99 \pm 0.00$ | $95.52 \pm 0.22$ |
| RMPM-Drop | $99.85 \pm 0.03$ | $97.49 \pm 0.12$ |
| MPM-SVD | $99.99 \pm 0.00$ | $96.14 \pm 0.04$ |

(b) Accuracies on Fashion-MNIST.

| Network | Train (%) | Test (%) |
|---|---|---|
| MLP | $98.17 \pm 0.12$ | $88.82 \pm 0.23$ |
| MP | $22.34 \pm 3.04$ | $22.05 \pm 2.93$ |
| DEP | $66.41 \pm 2.08$ | $65.30 \pm 2.38$ |
| DEP ($\lambda = 1/2$) | $70.99 \pm 2.12$ | $70.30 \pm 2.29$ |
| Act-MP | $49.37 \pm 9.84$ | $48.50 \pm 9.54$ |
| Act-DEP | $68.52 \pm 2.26$ | $66.79 \pm 2.25$ |
| Act-DEP ($\lambda = 3/4$) | $82.82 \pm 0.89$ | $79.26 \pm 0.64$ |
| Act-DEP ($\lambda = 1/2$) | $95.13 \pm 0.59$ | $82.58 \pm 0.39$ |
| MPM | $98.42 \pm 0.01$ | $82.86 \pm 0.17$ |
| RMPM | $99.66 \pm 0.03$ | $84.24 \pm 0.26$ |
| RMPM-Drop | $96.07 \pm 0.19$ | $86.88 \pm 0.19$ |
| MPM-SVD | $99.63 \pm 0.02$ | $84.72 \pm 0.12$ |

convergence; for convergence results refer to Appendix D) and achieve training accuracy comparable to that of a standard MLP. Among morphological networks without dropout, RMPM demonstrates slightly better generalization than MPM, while MPM-SVD – operating under a different setting – achieves the best generalization. With weight dropout, RMPM-Drop improves generalization significantly, achieving on MNIST/Fashion-MNIST test accuracies 97.49%/86.88%, 0.52%/1.94% lower compared to the linear MLP.

## 4.2 Convolutional networks

In this section, we evaluate various convolutional networks on the MNIST, Fashion-MNIST, and CIFAR-10 datasets. We train the following networks:

- **LeNet-5**: A variant of the standard, linear LeNet-5.
- **MPM-LeNet-5**: A morphological LeNet-5 according to Setting 1. Both convolutional and fully connected layers are morphological.
- **MPM-SVD-LeNet-5**: A morphological LeNet-5 according to Setting 2. Both convolutional and fully connected layers are morphological.
- **ResNet-20**: A variant of the standard, linear ResNet-20.
- **MPM-ResNet-20**: A morphological ResNet-20 according to Setting 1, trained for 100 epochs. All convolutional layers are morphological.

For MPM-LeNet-5 and MPM-SVD-LeNet-5, morphological convolutions were implemented with a LogSum-Exp scheme for compatibility with PyTorch. For MPM-ResNet-20, we developed a CUDA module for PyTorch implementing max-plus convolution under its strict definition. For all LeNet-5 networks, a slight variation was adopted: incorporating max-pooling instead of average pooling and using $5 \times 5$ kernels with a padding of 1 for all convolutional layers. For ResNet-20 networks, we used max-pooling instead of stride for down-sampling. Although the validation accuracy converged within 50 epochs, for MPM-ResNet-20 the training accuracy took longer to converge, and thus we trained it for 100 epochs. We did not use any form of dropout. The results are reported in Tables 2, 3, 4. On MNIST, it is clear that the networks we propose are trainable. It is also clear that they benefit from incorporating convolutions, showcasing a clear jump in generalization compared to fully connected networks. However, they lag behind the linear LeNet-5. On Fashion-MNIST, the MPM-LeNet-5 and MPM-SVD-LeNet-5 networks struggled to train as effectively as their fully connected counterparts. The deeper MPM-ResNet-20 network, on the other hand, trained successfully and reached 89.38% test accuracy, comparable to the linear LetNet-5 but slightly lower than the linear ResNet-20. On CIFAR-10, MPM-ResNet-20 reached satisfactory training accuracy, and 62.14% test accuracy. Although the accuracy is lower than that of a linear ResNet-20, the results show that our proposed

Table 2: Train (peak) and test accuracy of convolutional networks on MNIST.

| Network | Train (%) | Test (%) |
|---|---|---|
| LeNet-5 | $99.99 \pm 0.01$ | $99.03 \pm 0.07$ |
| MPM-LeNet-5 | $98.46 \pm 0.76$ | $97.08 \pm 0.30$ |
| MPM-SVD-LeNet-5 | $99.21 \pm 0.19$ | $97.25 \pm 0.28$ |

Table 3: Train (peak) and test accuracy of convolutional networks on Fashion-MNIST.

| Network | Train (%) | Test (%) |
|---|---|---|
| LeNet-5 | $99.20 \pm 0.07$ | $90.14 \pm 0.10$ |
| MPM-LeNet-5 | $85.22 \pm 1.56$ | $82.12 \pm 1.47$ |
| MPM-SVD-LeNet-5 | $88.57 \pm 0.72$ | $84.52 \pm 0.28$ |
| ResNet-20 | $99.69 \pm 0.05$ | $92.42 \pm 0.04$ |
| MPM-ResNet-20 | $96.30 \pm 0.22$ | $89.38 \pm 0.26$ |

morphological networks are capable of learning meaningful representations. To the best of our knowledge, this is the first demonstration of training such networks on CIFAR-10. For comparison, a linear LeNet-5 typically achieves around 60–65% test accuracy on this dataset.

### 4.3 Pruning

We study pruning as a means of investigating whether the proposed architectures require prohibitively large active parameter counts to achieve expressive representations in practice. We consider two pruning methods: i) unstructured $\ell_1$ pruning, and ii) the pre-training pruning method SNIP (Lee et al., 2019). The results are reported in Tables 5 and 6.

For $\ell_1$ pruning, in each layer a given fraction $r$ of low-importance weights are set to zero (i.e., they are not structurally removed), where importance is measured by the absolute value of the weight. In practice, this produces sparse parameter matrices that can be stored more efficiently.

SNIP is applied as standard: redundant weights are identified and are completely removed from the network. For SNIP, we apply a slightly modified version of the standard procedure. Since SNIP relies on gradient-based importance scores and morphological networks exhibit sparse gradients (Theorem 2), directly applying SNIP can produce unstable pruning scores. To mitigate this issue, we apply SNIP after a short warm-up phase and aggregate importance scores across multiple batches. The same procedure is used for both linear and morphological networks.

For aggressively pruned fully connected MPMs, we additionally maintain validity masks to avoid degenerate max/min units with all inputs removed.

We can see that, even after aggressive pre-training pruning, the fully connected RMPM maintains stable performance, indicating that an excessive number of active parameters are not required for the networks to achieve satisfactory expressivity.

Table 4: Train (peak) and test accuracy of convolutional networks on CIFAR-10.

| Network | Train (%) | Test (%) |
|---|---|---|
| ResNet-20 | $99.10 \pm 0.11$ | $81.74 \pm 0.27$ |
| MPM-ResNet-20 | $95.34 \pm 0.80$ | $62.14 \pm 0.22$ |

Table 5: Performance under $\ell_1$-based weight masked MLP and MPM for various masking ratios on MNIST and Fashion-MNIST.

| Masking ratio | MNIST | | Fashion-MNIST | |
|---|---|---|---|---|
| | MLP | MPM | MLP | MPM |
| 0.85 | $71.93 \pm 5.48$ | $93.59 \pm 0.65$ | $54.12 \pm 4.38$ | $80.83 \pm 1.09$ |
| 0.875 | $58.52 \pm 4.63$ | $93.02 \pm 0.73$ | $34.46 \pm 5.50$ | $79.92 \pm 1.17$ |
| 0.9 | $38.20 \pm 8.89$ | $92.21 \pm 1.01$ | $17.17 \pm 2.23$ | $79.19 \pm 0.99$ |
| 0.925 | $17.19 \pm 3.53$ | $90.88 \pm 1.01$ | $13.29 \pm 2.02$ | $77.53 \pm 1.74$ |
| 0.95 | $11.51 \pm 1.80$ | $78.65 \pm 13.99$ | $10.70 \pm 1.19$ | $74.94 \pm 0.16$ |
| 0.975 | $9.88 \pm 0.25$ | $64.37 \pm 10.75$ | $10.27 \pm 0.19$ | $57.11 \pm 4.62$ |

Table 6: Performance of SNIP pruned MLP and RMPM for various pruning ratios on MNIST and Fashion-MNIST.

| MLP (initial params: 466698) | | | | RMPM (initial params: 469268) | | | |
|---|---|---|---|---|---|---|---|
| Pruning ratio | Params (kept) | MNIST | F-MNIST | Pruning ratio | Params (kept) | MNIST | F-MNIST |
| 0.9875 | 5839 | $95.41 \pm 0.24$ | $85.05 \pm 0.06$ | 0.9875 | 5895 | $94.61 \pm 0.16$ | $84.32 \pm 0.23$ |
| 0.9900 | 4672 | $94.98 \pm 0.16$ | $84.32 \pm 0.13$ | 0.9875 | 4722 | $94.22 \pm 0.07$ | $83.72 \pm 0.12$ |
| 0.9925 | 3506 | $93.97 \pm 0.42$ | $82.92 \pm 0.48$ | 0.9875 | 3549 | $93.26 \pm 0.22$ | $82.95 \pm 0.17$ |
| 0.9950 | 2340 | $88.41 \pm 4.99$ | $79.09 \pm 1.33$ | 0.9875 | 2376 | $90.17 \pm 0.27$ | $79.98 \pm 0.22$ |

### 4.4 Hybrid-MLP: Gradient descent with large batches

In this section, we train the proposed Hybrid-MLP with 5 linear and 5 morphological hidden layers of size 256 each, using different batch sizes and report the results in Table 7. Our findings indicate that the network requires a large batch size to be trainable, suggesting that the inclusion of morphological layers introduces significant noise in the gradient estimation of stochastic optimization methods like Adam. However, for sufficiently large batch sizes, Adam exhibits rapid convergence. This is evident in Figures 1a, 1b, 1c, and 1d, which compare the training and validation accuracy of a standard MLP, a Maxout network (Goodfellow et al., 2013), and a Hybrid-MLP, each with five layers of size 256, trained with Adam using a batch size of 6400. Under sufficiently large batch sizes, the Hybrid-MLP exhibits stable and rapid optimization dynamics, suggesting that reducing gradient noise may play an important role in training deep morphological architectures.

## 5   Conclusion

We studied deep morphological neural networks from the perspective of how alternative algebraic structures affect the expressiveness and trainability of deep architectures. Our analysis revealed fundamental algebraic and optimization limitations of existing deep morphological networks, motivating the introduction of constrained linear activations between morphological layers. We showed that these minimal augmentations

Table 7: Performance of Hybrid-MLP on MNIST and Fashion-MNIST for different batch sizes

| | MNIST | | | Fashion-MNIST | | |
|---|---|---|---|---|---|---|
| | 64 | 640 | 6400 | 64 | 640 | 6400 |
| Train Acc. (peak) (%) | $33.54 \pm 2.89$ | $98.52 \pm 0.18$ | $99.96 \pm 0.06$ | $35.13 \pm 2.79$ | $92.68 \pm 0.40$ | $98.90 \pm 0.19$ |
| Train Acc. (last epoch) (%) | $18.66 \pm 5.28$ | $16.76 \pm 1.31$ | $99.81 \pm 0.27$ | $19.76 \pm 6.20$ | $32.32 \pm 15.92$ | $98.36 \pm 0.27$ |
| Validation Acc. (%) | $33.37 \pm 2.60$ | $96.76 \pm 0.21$ | $97.63 \pm 0.15$ | $34.61 \pm 2.69$ | $88.62 \pm 0.02$ | $88.96 \pm 0.19$ |
| Test Acc. (%) | $33.59 \pm 3.20$ | $96.74 \pm 0.23$ | $97.42 \pm 0.15$ | $34.64 \pm 2.40$ | $87.89 \pm 0.24$ | $88.15 \pm 0.24$ |

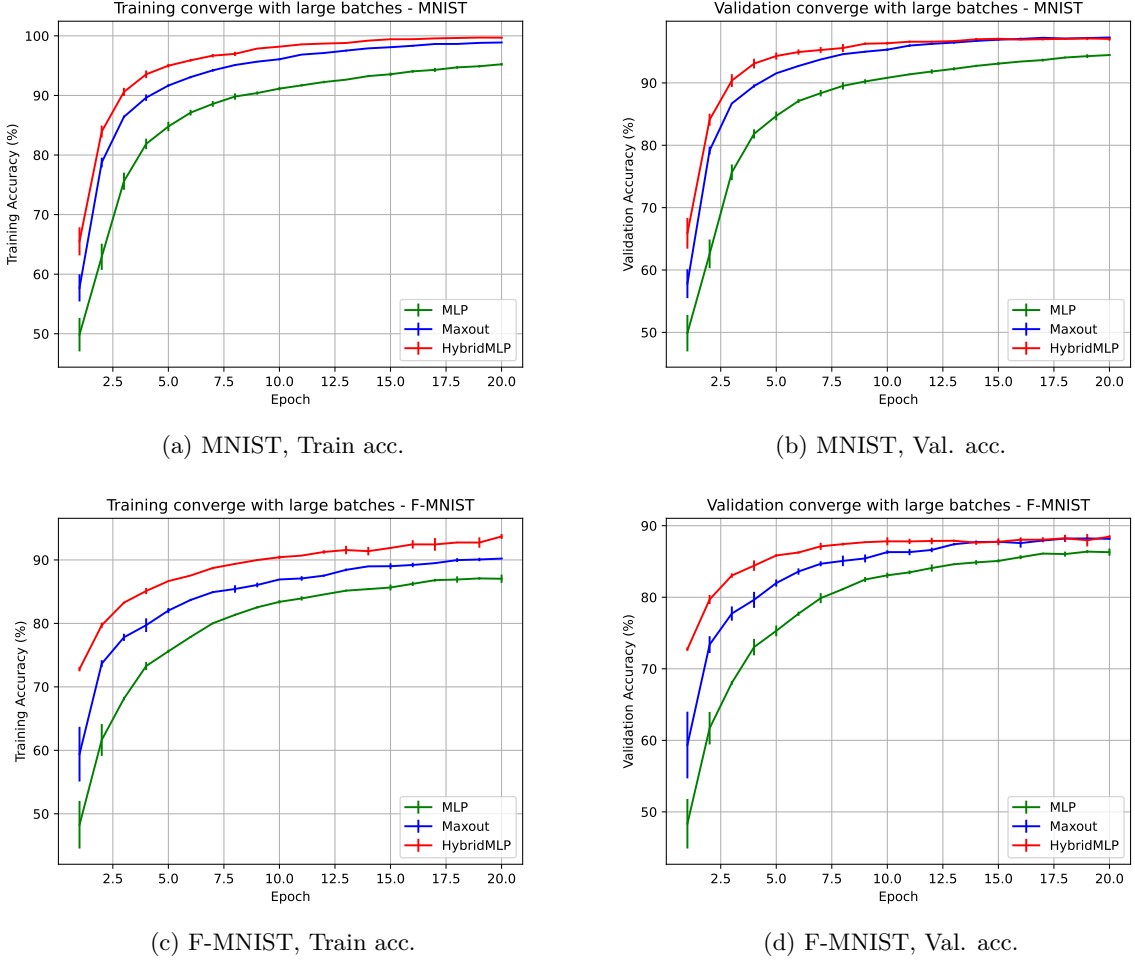

(a) MNIST, Train acc.

(b) MNIST, Val. acc.

(c) F-MNIST, Train acc.

(d) F-MNIST, Val. acc.

Figure 1: Convergence rate of different models on MNIST and Fashion-MNIST for 6400 batch size

restore universal approximation while remaining compatible with successful training on standard image classification tasks. Residual connections and weight dropout further improved generalization, and our fully connected pruning experiments demonstrated that the resulting architectures remain competitive without requiring prohibitively large active parameter counts. More broadly, our results suggest that deep learning architectures built on non-standard algebraic operations may require carefully structured linear components to achieve expressive and trainable representations. We hope this work motivates further study of neural architectures beyond the standard linear setting and improved optimization methods for sparse-gradient regimes.

**Broader Impact Statement**

This work is primarily theoretical and methodological, studying how alternative algebraic structures affect the expressiveness and trainability of deep neural architectures. We do not anticipate ethical concerns beyond those already associated with standard neural network theory research. The proposed methods were evaluated only on standard image classification benchmarks and do not involve sensitive data, human subjects, or high-risk deployment settings. More broadly, architectures with constrained parameterizations may contribute to future research on computationally efficient learning systems.

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

## Organization of the appendix

The Appendix is organized as follows:

Appendix A establishes that previous architectures are not universal approximators. Specifically, we prove Theorems 1 and 3 and show that these results are sufficient to conclude non-universality with Theorems A.5 and A.7. Additionally, we prove Theorem 2, which highlights the sparse gradient signal in MP-based networks.

Appendix B proves Theorems 4 and 6 and Corollary 5, demonstrating that our proposed networks are universal approximators.

Appendix C investigates the initialization of morphological neural networks. In particular, apart from the activation of their layers, we showcase the main problem that makes training and initializing morphological networks difficult. We show how initializing our proposed networks is easier than initializing MP-based networks or DEP-based networks with $\lambda \neq 1/2$. Finally, we explain how the networks in our experiments were initialized, and the reasoning behind it.

Appendix D presents our experimental setup details, compute resources required, and our declaration of LLM usage. In addition, it includes additional experimental results, including experiments on simple regression tasks, and the full training history of networks that we claimed were trainable but for which only peak training accuracy was reported in the main body.

Appendix E presents the relationship between our results and the Representation Theorem (Maragos & Schafer, 1987). Specifically, it showcases the differences in assumptions made, where these matter in the proof of our theorems, and shows also that our theoretical results are actually complementary to the Representation Theorem.

## A  Results on previous work

In this appendix we establish our results regarding previous architectures. Before we proceed with our proofs, we give the definition of a universal approximator.

**Definition A.0** (Universal approximator). *We say that a class of functions $\mathcal{F}$ is a universal approximator on a set $\mathcal{D} \subseteq \mathbb{R}^n$ if and only if it is a dense subclass of the class of continuous functions on $\mathcal{D}$ under the $\|\cdot\|_\infty$ function norm. Equivalently, for any continuous $g$ defined on $\mathcal{D}$, there exists a sequence $\{f_i\}_{i \in \mathbb{N}} \subseteq \mathcal{F}$ such that $f_i \to g$ uniformly.*

We say that a network architecture is a universal approximator on $\mathcal{D}$ if the class of functions it defines is a universal approximator on $\mathcal{D}$. Most of our claims hold for compact $\mathcal{D}$, which for $\mathbb{R}^n$ correspond to bounded and closed domains by the Heine–Borel theorem. With a slight abuse of terminology, whenever we say that a network is (or is not) a universal approximator, we mean that the defined architecture is (or is not) a universal approximator.

For our proofs, we are going to need the following lemmas.

**Lemma A.1.** *Let $\mathcal{B} \subseteq \mathbb{R}$ be a non-trivial compact interval, and let $\mathcal{F}$ be the class of Lipschitz continuous, single-variate functions $f : \mathcal{B} \to \mathbb{R}$ with $|f'(x)| \leq 1$ a.e.. The class $\mathcal{F}$ is not a universal approximator on $\mathcal{B}$ (i.e. it is not a dense subclass of the class of continuous functions on $\mathcal{B}$).*

*Proof.* Since $\mathcal{B}$ has non-empty interior, there exists $x_0$ and $0 < r < 1$ such that $[x_0 - r, x_0 + r] \subseteq \mathcal{B}$. Let $g(x) = x/r$, a continuous function on $\mathcal{B}$. For the sake of contradiction, suppose that $\mathcal{F}$ is a universal approximator. Then, for every $\varepsilon > 0$ we can find $f \in \mathcal{F}$ such that $|f(x) - g(x)| < \varepsilon, \forall x \in \mathcal{B}$. In particular, $f(x_0) < g(x_0) + \varepsilon = x_0/r + \varepsilon$ and $f(x_0 + r) > g(x_0 + r) - \varepsilon = x_0/r + 1 - \varepsilon$. It follows that $f(x_0 + r) - f(x_0) > 1 - 2\varepsilon$. However, f is Lipschitz continuous, and hence also absolutely continuous. Thus, it is differentiable a.e., and it holds that $f(x_0 + r) - f(x_0) = \int_{x_0}^{x_0+r} f'(x)dx$. But, due to the bound on the derivates of $f$, we have that $\int_{x_0}^{x_0+r} f'(x)dx < \int_{x_0}^{x_0+r} 1dx = r \Rightarrow r > 1 - 2\varepsilon, \forall \varepsilon > 0 \Rightarrow r \geq 1$, a contradiction. $\qquad\square$

The previous lemma can be generalized to functions of multiple variables.

**Lemma A.2.** *Let $\mathcal{B} \subseteq \mathbb{R}^d$ be a compact, convex domain with non-empty interior, and let $\mathcal{F}$ be the class of Lipschitz continuous functions $f : \mathcal{B} \to \mathbb{R}$ with $\|\nabla f(\mathbf{x})\|_1 \leq 1$ a.e.. The class $\mathcal{F}$ is not a universal approximator on $\mathcal{B}$.*

*Proof.* First, we prove that if $\mathcal{H}$ is a universal approximator on $\mathcal{B}$, $\mathbf{x}_0 \in \text{int}(\mathcal{B})$, $\hat{\mathbf{v}}$ is a directional vector, and $\widetilde{\mathcal{B}} = \{t \in \mathbb{R} : \mathbf{x}_0 + t\hat{\mathbf{v}} \in \mathcal{B}\}$, then $\widetilde{\mathcal{H}} = \{f(\mathbf{x}_0 + t\hat{\mathbf{v}})|f \in \mathcal{H}\}$ is a universal approximator on $\widetilde{\mathcal{B}}$. Let $\tilde{g} : \widetilde{\mathcal{B}} \to \mathbb{R}$ be a continuous function. Since $\widetilde{\mathcal{B}}$ is a compact interval, we extend the function continuously on $\mathbb{R}$. For every $\mathbf{x} \in \mathcal{B}$ if $\mathbf{x}_0 + t\mathbf{v}$ is its projection on the line, we define $g(\mathbf{x}) = \tilde{g}(t)$. In the end, we obtain a continuous $g$ on $\mathcal{B}$ for which $\tilde{g}(t) = g(\mathbf{x}_0 + t\hat{\mathbf{v}})$. Since $\mathcal{H}$ is a universal approximator of $\mathcal{B}$, we can find a sequence $f_1, f_2, \dots$ of functions in $\mathcal{H}$ such that $f_n \to g$ uniformly on $\mathcal{B}$, i.e. $\sup_{\mathbf{x} \in \mathcal{B}}(f_n(\mathbf{x}) - g(\mathbf{x})) \to 0 \Rightarrow \sup_{t \in \widetilde{\mathcal{B}}}(f_n(\mathbf{x}_0 + t\hat{\mathbf{v}}) - \tilde{g}(t)) \to 0$. Thus, $f_1(\mathbf{x}_0 + t\hat{\mathbf{x}}), f_2(\mathbf{x}_0 + t\hat{\mathbf{x}}), \dots$ approximates the continuous function $\tilde{g}$, and $\widetilde{\mathcal{H}}$ is a universal approximator on $\widetilde{\mathcal{B}}$.

Next, we take some $\mathbf{x}_0 \in \text{int}(\mathcal{B})$ and $\hat{\mathbf{v}} = (1, 0, \dots, 0)$. To prove that $\mathcal{F}$ is not a universal approximator on $\mathcal{B}$, it suffices to prove that $\widetilde{\mathcal{F}} = \{f(\mathbf{x}_0 + t\hat{\mathbf{x}})|f \in \mathcal{F}\}$ is not a universal approximator on $\widetilde{\mathcal{B}}$. Notice that $\widetilde{\mathcal{F}}$ and $\widetilde{\mathcal{B}}$ satisfy all requirements of Lemma A.1. Indeed, $\mathcal{B}$ is compact, convex, with non-empty interior, and thus $\widetilde{\mathcal{B}}$ is a non-trivial compact interval. In addition, let $\tilde{f} \in \widetilde{\mathcal{F}}$. Then, we have that

1. $\tilde{f}$ is Lipschitz: $|\tilde{f}(t_2) - \tilde{f}(t_1)| = |f(\mathbf{x}_0 + t_2\hat{\mathbf{v}}) - f(\mathbf{x}_0 + t_1\hat{\mathbf{v}})| \leq L\|\mathbf{x}_0 + t_2\hat{\mathbf{v}} - \mathbf{x}_0 + t_1\hat{\mathbf{v}}\| = L|t_2 - t_1|\|\hat{\mathbf{v}}\| = L|t_2 - t_1|$

2. $|\tilde{f}'(t)| \leq 1$ a.e.: Since $f$ is Lipschitz, by Rademacher's theorem, it is differentiable a.e. on $\text{int}(\mathcal{B})$. Since $\mathcal{B}$ is convex, its boundary has zero measure, and hence its interior $\text{int}(\mathcal{B})$ has same measure as $\mathcal{B}$. This means that $f$ is differentiable a.e. on $\mathcal{B}$. This in turn means that the directional derivative of $f$ in the direction $\hat{\mathbf{v}}$ (i.e. the derivative of $\tilde{f}$) equals $\langle \nabla f, \hat{\mathbf{v}} \rangle = (\nabla f)_1$ a.e.. It also holds that $|(\nabla f)_1| \leq \|\nabla f\|_1 \leq 1$ a.e..

Hence, by Lemma A.1, $\widetilde{\mathcal{F}}$ is not a universal approximator on $\widetilde{\mathcal{B}}$, and $\mathcal{F}$ is not a universal approximator on $\mathcal{B}$. $\qquad\square$

Before we proceed with the proof of Theorem 1, we will need the following two auxiliary lemmas.

**Lemma A.3.** *Let $\mathcal{B} \subseteq \mathbb{R}^d$ and $f_1, f_2, \dots, f_n : \mathcal{B} \to \mathbb{R}$ be Lipschitz continuous functions with Lipschitz constants $L_1, \dots$. Then, their pointwise maximum and point-wise minimum are Lipschitz continuous functions.*

*Proof.* We can prove the statement for two functions $f_1$ and $f_2$, and for multiple functions it will follow from induction on $n$.

Notice that $\max(f_1, f_2) = \frac{f_1 + f_2 + |f_1 - f_2|}{2}$, and $\min(f_1, f_2) = \frac{f_1 + f_2 - |f_1 - f_2|}{2}$. Since the addition and scaling of Lipschitz functions is Lipschitz, it suffices to show that $|f_1 - f_2|$ is Lipschitz. By triangle inequality, we have that

$$\left||f_1(\mathbf{x}_1) - f_2(\mathbf{x}_1)| - |f_1(\mathbf{x}_2) - f_2(\mathbf{x}_2)|\right| \leq |f_1(\mathbf{x}_1) - f_2(\mathbf{x}_1) - f_1(\mathbf{x}_2) + f_2(\mathbf{x}_2)|$$

$$\leq |f_1(\mathbf{x}_1) - f_1(\mathbf{x}_2)| + |f_2(\mathbf{x}_1) - f_2(\mathbf{x}_2)| \leq (L_1 + L_2)\|\mathbf{x}_1 - \mathbf{x}_2\|, \forall \mathbf{x}_1, \mathbf{x}_2$$

Thus, $\max(f_1, f_2), \min(f_1, f_2)$ are Lipschitz. By induction, $\max_{i \in [n]} f_i, \min_{i \in [n]} f_i$ are Lipschitz. $\qquad\square$

**Lemma A.4.** *Let open $\mathcal{B} \subseteq \mathbb{R}^d$, $f_1, f_2, \dots, f_n : \mathcal{B} \to \mathbb{R}$, and $g = \max_{i \in [n]}(f_i), h = \min_{i \in [n]}(f_i)$. Let $\mathbf{x}_0 \in \mathcal{B}$ and $I = \{i \in [n] : f_i(\mathbf{x}_0) = g(\mathbf{x}_0)\}, J = \{i \in [n] : f_i(\mathbf{x}_0) = h(\mathbf{x}_0)\}$. Suppose that $f_i$:differentiable on $\mathbf{x}_0, \forall i \in [n]$. Then, we have that*

$$g : \text{diff. on } \mathbf{x}_0 \Rightarrow \nabla g(\mathbf{x}_0) = \nabla f_i(\mathbf{x}_0), \forall i \in I.$$

$$h : \text{diff. on } \mathbf{x}_0 \Rightarrow \nabla h(\mathbf{x}_0) = \nabla f_j(\mathbf{x}_0), \forall j \in J.$$

*Proof.* We will prove only the part of the statement involving the maximum, with the proof of the statement for the minimum being similar. Suppose that $g$ is differentiable on $\mathbf{x}_0$ and that there exists $i \in I$ such that $\nabla g(\mathbf{x}_0) \neq \nabla f_i(\mathbf{x}_0)$. Then, we have that

$$\nabla(f_i - g)(\mathbf{x}_0) \neq \mathbf{0} \Rightarrow \exists \hat{\mathbf{v}} : \nabla(f_i - g)(\mathbf{x}_0) \cdot \hat{\mathbf{v}} > 0 \Rightarrow \exists \hat{\mathbf{v}} : \frac{\partial(f_i - g)}{\partial \hat{\mathbf{v}}}(\mathbf{x}_0) > 0$$

But it also holds that $(f_i - g)(\mathbf{x}_0) = 0$. Hence, for small enough $t > 0$, we have that

$$(f_i - g)(\mathbf{x}_0 + t\hat{\mathbf{v}}) > 0$$

$$\Rightarrow f_i(\mathbf{x}_0 + t\hat{\mathbf{v}}) > g(\mathbf{x}_0 + t\hat{\mathbf{v}}) = \max_{j \in [n]}(f_j(\mathbf{x}_0 + t\hat{\mathbf{v}})),$$

which is a contradiction. $\qquad\square$

**Remark.** *In Lemma A.4 it is implicitly proven that if $g$ is differentiable at $\mathbf{x}_0$, then the gradients of $f_i$ at $\mathbf{x}_0$ for $i \in I$ will be equal.*

Next, we restate and prove Theorem 1.

**Theorem 1.** *For any network that only uses max-plus and min-plus MPs with input $\mathbf{x} \in \mathbb{R}^d$ and a single output $y(\mathbf{x})$, we have that $y(\mathbf{x})$ is Lipschitz continuous on $\mathbb{R}^d$ and a.e. it holds that either $\nabla y(\mathbf{x}) = 0$ or $\nabla y(\mathbf{x}) = \mathbf{e}_i = [0, \dots, 1, \dots, 0]^\top$ for some $i = i(\mathbf{x})$.*

*Proof.* First, we prove that $y$ is Lipschitz continuous. We use induction on the layers of the network to prove that the output of each unit is Lipschitz continuous. For the base case $n = 0$, notice that each input $i$ can be thought of as a projection $\mathbf{x} \to x_i$, which is Lipschitz continuous. Suppose that the outputs $x_j^{(n)}$ of all units of the $n$-th layer are Lipschitz continuous. Without loss of generality, suppose the $i$-th unit of the $(n+1)$-th layer is a max-plus MP. We have that $x_j^{(n)} + w_{ij}^{(n+1)}$ is Lipschitz for all $j \in [N^{(n)}]$, $w_{i0}^{(n+1)}$ is Lipschitz, and hence by Lemma A.3 $x_i^{(n+1)} = w_{i0}^{(n+1)} \vee \max_{j \in [n]}(x_j^{(n)} + w_{ij}^{(n+1)})$ is Lipschitz continuous. Thus, by induction, the outputs of all units of the networks, including output $y$ of the last unit, are Lipschitz continuous.

Next, we prove the remaining part of the theorem. To simplify notation, we denote $\mathbf{e}_0 = \mathbf{0} = [0, \dots, 0]^\top$.

To prove our theorem, we use induction on the number of layers. For the base case, i.e. the input, we have that $\nabla x_i = \mathbf{e}_i$ everywhere. Suppose that the claim holds true for the $n$-th layer. Take any MP unit of the $(n+1)$-th layer, say the $i$-th unit, and suppose without loss of generality that it is a max-plus unit. We have that

$$x_i^{(n+1)}(\mathbf{x}) = w_{i0}^{(n+1)} \vee \max_j(x_j^{(n)}(\mathbf{x}) + w_{ij}^{(n+1)}).$$

We set $f_{i0}^{(n+1)} = w_{i0}^{(n+1)}, f_{ij}^{(n+1)} = x_j^{(n)} + w_{ij}^{(n+1)}$. Notice that by Rademacher's theorem, we have that $x_i^{(n+1)}(\mathbf{x})$ is differentiable a.e. on $\mathbb{R}^d$. In addition, by the inductive hypothesis, we have that $x_j^{(n)}(\mathbf{x})$ is differentiable a.e. with $(\nabla x_j^{(n)}(\mathbf{x}) = \mathbf{e}_{k_j(\mathbf{x})}$ for some $k_j(\mathbf{x}))$ a.e., and hence $f_{ij}^{(n+1)}$ is differentiable a.e. with $(\nabla f_{ij}^{(n+1)} = \nabla x_j^{(n)} = \mathbf{e}_{k_j(\mathbf{x})}$ for some $k_j(\mathbf{x}))$ a.e.. Moreover, $w_{i0}^{(n+1)}$ is differentiable everywhere with $\nabla f_{i0}^{(n+1)} = \nabla w_{i0}^{(n+1)} = \mathbf{e}_0$. This means that Lemma A.4 holds a.e., and we have that a.e. $\nabla x_i^{(n+1)}(\mathbf{x}) = \nabla f_{ij}^{(n+1)}(\mathbf{x}), \forall j \in J$, where $J = \{j : x_i^{(n+1)}(\mathbf{x}) = f_{ij}^{(n+1)}(\mathbf{x})\}$. This means that a.e.: 1) $k_j(\mathbf{x}) = k(\mathbf{x}), \forall j \in J$, and 2) $\nabla x_i^{(n+1)}(\mathbf{x}) = \mathbf{e}_{k(\mathbf{x})}$. This concludes the induction. $\qquad\square$

Notice that the above proof also gives as a way to find the derivative (i.e. the $i = i(\mathbf{x})$ of the statement): It suffices to work backwards, following the paths where the maximum or the minimun is attained. If we end up with two or more "leafs" as inputs/biases, then the function is not differentiable. Otherwise, if we end up at a unique leaf, which can either be an input or a bias, then the index of the input (0 for the bias) is the $i = i(\mathbf{x})$ we are looking for. Refer to Figure 2a for a differentiable example, and Figure 2b for a non-differentiable example.

Combining Theorem 1 and Lemma A.2, we obtain the following result.

**Theorem A.5.** *Networks that only use max-plus and min-plus MPs are not universal approximators.*

*Proof.* We prove that they are not universal approximators on any compact, convex domain $\mathcal{B} \subseteq \mathbb{R}^d$ with non-empty interior, which is stronger that not being universal approximators on $\mathbb{R}^d$. Notice that by Theorem 1 we have that a.e. either $\nabla y(\mathbf{x}) = 0$ or $\nabla y(\mathbf{x}) = \mathbf{e}_{i(\mathbf{x})} \Rightarrow \|\nabla y(\mathbf{x})\|_1 \leq 1$. Hence, $\|\nabla y(\mathbf{x})\|_1 \leq 1$ a.e. on $\mathbb{R}^n$. If we restrict $y$ on $\mathcal{B}$, we have a compact, convex, domain $\mathcal{B}$ with non-empty interior and a Lipschitz continuous $y(\mathbf{x})$ with $\|\nabla y(\mathbf{x})\|_1 \leq 1$ a.e. on $\mathcal{B}$. From Lemma A.2 it follows that networks that only use max-plus and min-plus MPs are not universal approximators on $\mathcal{B}$. $\square$

Before we proceed with the proof of Theorem 2, we prove the following lemma.

**Lemma A.6.** *Let open $\mathcal{B} \subseteq \mathbb{R}^d$, continuous $f_1, \ldots, f_n : \mathcal{B} \to \mathbb{R}$, and $g = \max_{i \in [n]}(f_i), h = \min_{i \in [n]}(f_i)$. If for some $\mathbf{x}_0 \in \mathcal{B}$ we have that the maximum is attained only at some $j \in [n]$ (i.e. $f_j(\mathbf{x}_0) > f_i(\mathbf{x}_0), \forall i \neq j$) and $f_j$ : differentiable on $\mathbf{x}_0$, then $g$ : differentinable on $\mathbf{x}_0$ with $\nabla g(\mathbf{x}_0) = \nabla f_j(\mathbf{x}_0)$. Similarly for minimum.*

*Proof.* We only prove the statement for the maximum, with the minimum being similar. Since $f_i$: continuous on $\mathbf{x}_0$ and $f_j(\mathbf{x}_0) > f_i(\mathbf{x}_0), \forall i \neq j$, we have that in a small enough neighborhood $N(\mathbf{x}_0, \varepsilon)$ it holds that $f_j(\mathbf{x}) > f_i(\mathbf{x}), \forall i \neq j, \mathbf{x} \in N(\mathbf{x}_0, \varepsilon)$. Then $g(\mathbf{x}) = f_j(\mathbf{x}), \forall \mathbf{x} \in N(\mathbf{x}_0, \varepsilon) \Rightarrow g$ : differentiable on $\mathbf{x}_0$ with $\nabla g(\mathbf{x}_0) = \nabla f_j(\mathbf{x}_0)$. $\square$

In a similar fashion to Theorem 1, we can prove Theorem 2.

**Theorem 2.** *Consider a network that only uses max-plus and min-plus MPs with output $\mathbf{y} \in \mathbb{R}^m$. For any given input $\mathbf{x}$, if $\mathbf{y}$ is differentiable with respect to the network parameters, then in each layer $n$, there exists at most $m$ parameters $w_{ij}^{(n)}$ for which the derivative of $\mathbf{y}$ is nonzero.*

*Proof.* We work backwards, following the paths where the maximum and the minimum is attained. If the maximum or the minimum is attained only for one argument, we keep track of the argument's "slack" and continue the path from this argument. We continue this process until either 1) we reach a "leaf" (i.e. an input or a bias), or 2) we find a maximum or minimum that is attained for multiple arguments and we get a "split".

In the second case, we have that the output is not differentiable with respect to the weights: If the split happens for the path of the $k$-th output, WLOG at a max-plus MP, and $w_{ij}^{(n)}$ is a weight such that the maximum is attained at $j$, then by letting a small $dw > 0$, smaller than the minimum of the recorded slacks, and $(w_{ij}^{(n)})' = w_{ij}^{(n)} + dw$, we have that all the units along the path are incremented by $dw$, and $y_k' = y_k + dw \Rightarrow dy_k = dw$. If, on the other hand, we let $dw < 0$, then the MP unit where the split happened remains unchanged and $y_k' = y_k \Rightarrow dy_k = 0$. Hence, $\mathbf{y}$ is not differentiable with respect to $w_{ij}^{(n)}$.

In the first case, each output $y_k$ has a path from $y_k$ to a "leaf". We have that the output is differentiable with respect to the weights, and the statement holds. We will prove this using induction on the nodes of the path. We will prove the following stronger statement: If $i_k(n) \geq 0$ is the unique node of layer $n$ that belongs to the path of $y_k$, and $\text{path}_k(n)$ is the set of weights that belong to the path of $y_k$ and are before the $n$-th layer, then for every $n$ it holds that $x_{i_k(n)}^{(n)}$ is differentiable with respect to the weights and $\nabla_{\mathbf{w}} x_{i_k(n)}^{(n)} = \sum_{w \in \text{path}_k(n)} \mathbf{e}_w$ (here we denote $x_0^{(n)} = 0$). Notice that since we have no "splits", paths can only merge, and hence the statement is indeed stronger.

Take the path of $y_k$. The statement obviously holds true for the first node of the path, i.e. the leaf. Suppose it holds true for the $n$-th node. Then, we have that $f_{i_k(n+1), i_k(n)} = x_{i_k(n)}^{(n)} + w_{i_k(n+1), i_k(n)}^{(n+1)}$ is differentiable in terms of the weights with $\nabla_{\mathbf{w}} f_{i_k(n+1), i_k(n)} = \sum_{w \in \text{path}_k(n)} \mathbf{e}_w + \mathbf{e}_{w_{i_k(n+1), i_k(n)}^{(n+1)}} = \sum_{w \in \text{path}_k(n+1)} \mathbf{e}_w$. Also since the maximum or minimum is attained only for $i_k(n)$, by Lemma A.6 we have that $x_{i_k(n+1)}^{(n+1)}$ is differentiable with respect to the weights, and $\nabla_{\mathbf{w}} x_{i_k(n+1)}^{(n+1)} = \nabla_{\mathbf{w}} f_{i_k(n+1), i_k(n)} = \sum_{w \in \text{path}_k(n+1)} \mathbf{e}_w$, which concludes the induction. $\square$

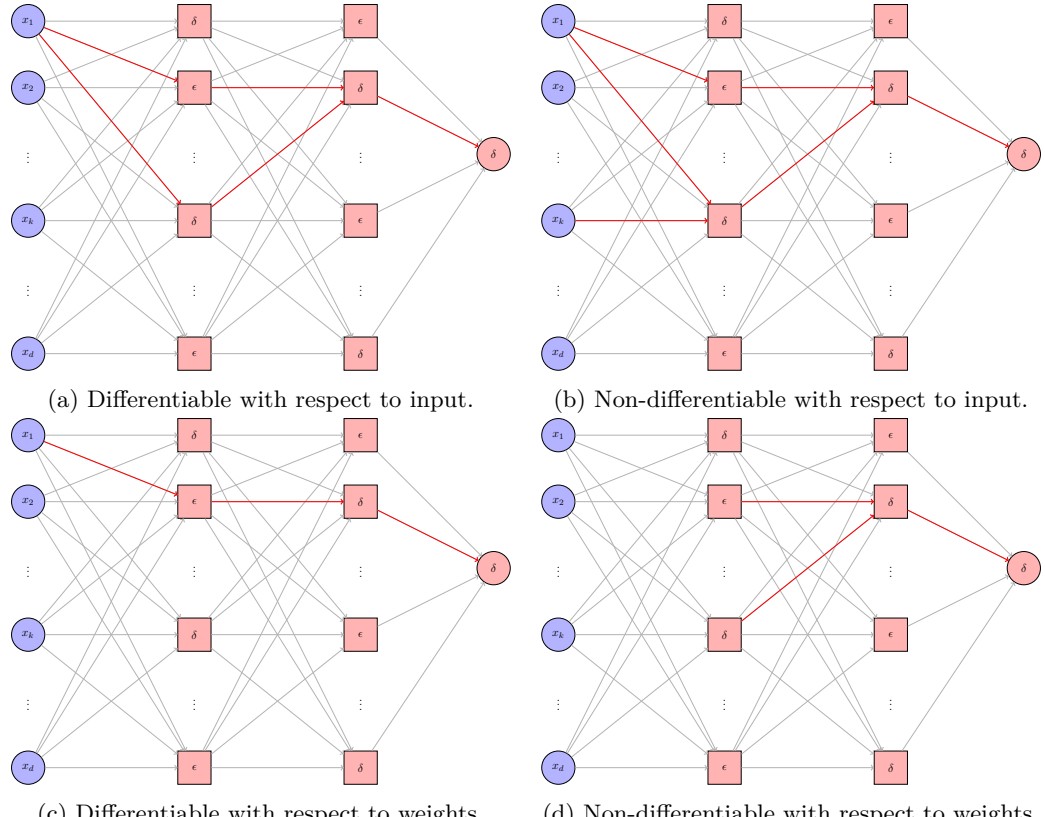

(a) Differentiable with respect to input.  (b) Non-differentiable with respect to input.

(c) Differentiable with respect to weights.  (d) Non-differentiable with respect to weights.

Figure 2: Cases of differentiable and non-differentiable networks with respect to input and weights.

The above proof also gives us a way to calculate the derivative with respect to any weight. Refer to Figure 2c for a differentiable example, and to Figure 2d for a non-differentiable example.

Continuing, we restate and prove Theorem 3. We consider two cases: the non-activated DEP-based network (i.e. $f^{(n)} = 1, \forall n$), and the activated DEP-based network with a common activation function (i.e. sigmoid as in (de A. Araújo et al., 2017), tanh, ReLU, ELU, or Leaky ReLU). For both cases, we prove a stronger variant of Theorem 3.

**Theorem 3** (non-activated)**.** *For any non-activated DEP-based network with input $\mathbf{x} \in \mathbb{R}^d$ and a single output $y(\mathbf{x})$, we have that $y(\mathbf{x})$ is Lipschitz continuous on $\mathbb{R}^d$ and a.e. it holds that $\nabla y(\mathbf{x}) \succeq 0, \|\nabla y(\mathbf{x})\|_1 = 1$.*

*Proof.* First, we prove that $y$ is Lipschitz continuous. We again use induction on the layers of the network to prove that the output of each unit is Lipschitz continuous. For the base case $n = 0$, notice that each input $i$ can be thought of as a projection $\mathbf{x} \to x_i$, which is Lipschitz continuous. Suppose that the outputs $x_j^{(n)}$ of all units of the $n$-th layer are Lipschitz continuous. We have that $x_j^{(n)} + w_{ij}^{(n+1)}$ is Lipschitz for all $j \in [N^{(n)}]$, and hence by Lemma A.3 $f_i^{(n+1)} = \max_{j \in [n]}(x_j^{(n)} + w_{ij}^{(n+1)})$ is Lipschitz continuous for all $i \in [N^{(n+1)}]$. Similarly, $g_i^{(n+1)} = \min_{j \in [n]}(x_j^{(n)} + m_{ij}^{(n+1)})$ is Lipschitz continuous. Since scaling and addtion preserve Lipschitz continuity, we conclude that $x_i^{(n+1)} = \lambda_i^{(n+1)} f_i^{(n+1)} + (1 - \lambda_i^{(n+1)}) g_i^{(n+1)}$ is also Lipschitz continuous. Thus, by induction, the outputs of all units of the networks, including output $y$ of the last unit, are Lipschitz continuous.

Next, we prove the remaining part of the theorem. To prove our theorem, we use induction on the number of layer. For the base case, i.e. the input, we have that $\nabla x_i = \mathbf{e}_i \Rightarrow \nabla x_i \succeq 0, \|\nabla x_i\|_1 = 1$ everywhere. Take

any DEP unit of the $(n+1)$-th layer, say the $i$-th unit. We have that

$$f_i^{(n+1)}(\mathbf{x}) = \max_j(x_j^{(n)}(\mathbf{x}) + w_{ij}^{(n+1)}),$$

$$g_i^{(n+1)}(\mathbf{x}) = \min_j(x_j^{(n)}(\mathbf{x}) + m_{ij}^{(n+1)}),$$

$$x_i^{(n+1)}(\mathbf{x}) = \lambda_i^{(n+1)} f_i^{(n+1)}(\mathbf{x}) + (1 - \lambda_i^{(n+1)})g_i^{(n+1)}(\mathbf{x}).$$

Notice that by Rademacher's theorem, we have that $f_i^{(n+1)}$ and $g_i^{(n+1)}$ are differentiable a.e. on $\mathbb{R}^d$ (since they are Lipschitz, see previous paragraph). In addition, by the inductive hypothesis, we have that $x_j^{(n)}$ is differentiable a.e., and hence $x_j^{(n)} + w_{ij}^{(n+1)}$ and $x_j^{(n)} + m_{ij}^{(n+1)}$ are differentiable a.e.. This means that Lemma A.4 holds a.e., and we have that a.e. $\nabla f_i^{(n+1)}(\mathbf{x}) = \nabla(x_j^{(n)}(\mathbf{x}) + w_{ij}^{(n+1)}) = \nabla x_j^{(n)}(\mathbf{x}), \forall j \in J_1, \nabla g_i^{(n+1)}(\mathbf{x}) = \nabla(x_j^{(n)}(\mathbf{x}) + w_{ij}^{(n+1)}) = \nabla x_j^{(n)}(\mathbf{x}), \forall j \in J_2$, where $J_1 = \{j : f_i^{(n+1)}(\mathbf{x}) = x_j^{(n)}(\mathbf{x}) + w_{ij}^{(n+1)}\}, J_2 = \{j : g_i^{(n+1)}(\mathbf{x}) = x_j^{(n)}(\mathbf{x}) + w_{ij}^{(n+1)}\}$. This means that a.e. $x_i^{(n+1)}$ is differentiable, with $\nabla x_i^{(n+1)} = \lambda_i^{(n+1)} \nabla x_{j_1}^{(n)} + (1-\lambda_i^{(n+1)}) \nabla x_{j_2}^{(n)}$, for some $j_1 \in J_1, j_2 \in J_2$. Since $\lambda_i^{(n+1)} \in [0,1]$ and a.e. $\nabla x_{j_1}^{(n)} \succeq 0, \nabla x_{j_2}^{(n)} \succeq 0$, we have also that a.e. $\nabla x_i^{(n+1)} \succeq 0$. Therefore, we also have a.e. that $\|\nabla x_i^{(n+1)}\|_1 = \sum_k (\nabla x_i^{(n+1)})_k = \sum_k (\lambda_i^{(n+1)} \nabla x_{j_1}^{(n)} + (1 - \lambda_i^{(n+1)}) \nabla x_{j_2}^{(n)})_k = \lambda_i^{(n+1)} \sum_k (\nabla x_{j_1}^{(n)})_k + (1 - \lambda_i^{(n+1)}) \sum_k (\nabla x_{j_2}^{(n)})_k = \lambda_i^{(n+1)} + 1 - \lambda_i^{(n+1)} = 1$. This concludes the induction. □

At this point, we should note that using standard non-linear activations, as is done by some existing work such as (de A. Araújo et al., 2017), who use a sigmoid function, does not solve the problem (and if anything, it makes it worse).

**Theorem 3** (activated). *For activated DEP-based networks with $L$ layers in total, of which $\widetilde{L}$ are activated by a common activation function $f$ with $0 \leq f' \leq s \leq 1$ a.e., input $\mathbf{x} \in \mathbb{R}^d$ and a single output $y(\mathbf{x})$, we have that $y(\mathbf{x})$ is Lipschitz continuous on $\mathbb{R}^d$ and a.e. it holds that $\nabla y(\mathbf{x}) \succeq 0, \|\nabla y(\mathbf{x})\|_1 \leq s^{\widetilde{L}} \leq 1$.*

*Proof.* We denote with $y_i^{(n)}$ the non-activated outputs, and $x_i^{(n)}$ the outputs after activation (where as activation we can either have a common activation function or an identity). Again, we denote with $f_i^{(n)}$ the output of the maximum operations, and by $g_i^{(n)}$ the output of the minimum operation.

Because the composition of Lipschitz continuous functions is Lipschitz continuous, and the common activation functions are Lipschitz, with a proof similar to that of the previous theorem it follows immediately that both $y_i^{(n)}$ and $x_i^{(n)}$ are Lipschitz continuous for all $n, i$.

We will prove that after $l \leq \widetilde{L}$ activated layers, we have that a.e. it holds that $\nabla x_i^{(n)} \succeq 0, \|\nabla x_i^{(n)}\|_1 \leq s^l$. We will prove this with induction. The base case is similar to that of the previous theorem. Suppose that the statement holds for the $n$-th layer, i.e. if $l$ activated layers precede the $n$-th output, then a.e. it holds that $\nabla x_i^{(n)} \succeq 0, \|\nabla x_i^{(n)}\|_1 \leq s^l$. With an argument similar to that of the previous theorem, we have that a.e. $\nabla y_i^{(n+1)} = \lambda_i^{(n+1)} \nabla x_{j_1}^{(n)} + (1 - \lambda_i^{(n+1)}) \nabla x_{j_2}^{(n)}$, for some $j_1 \in J_1, j_2 \in J_2$, where $J_1, J_2$ are defined as in the above proof. Since $\lambda_i^{(n+1)} \in [0,1]$ and a.e. $\nabla x_{j_1}^{(n)} \succeq 0, \nabla x_{j_2}^{(n)} \succeq 0$, we also have that a.e. $\nabla y_i^{(n+1)} \succeq 0$. Also, a.e. $\|\nabla y_i^{(n+1)}\|_1 \leq \lambda_i^{(n+1)} \|\nabla x_{j_1}^{(n)}\|_1 + (1 - \lambda_i^{(n+1)}) \|\nabla x_{j_2}^{(n)}\|_1 \leq \lambda_i^{(n+1)} s^l + (1 - \lambda_i^{(n+1)}) s^l = s^l$. If the $(n+1)$-th layer is not activated, then we have that $x_i^{(n+1)} = y_i^{(n+1)}$, and then, $l$ activated layers precede the $(n+1)$-th output, and we have that a.e. $\nabla x_i^{(n+1)} = \nabla y_i^{(n+1)} \succeq 0, \|\nabla x_i^{(n+1)}\|_1 = \|\nabla y_i^{(n+1)}\|_1 \leq s^l$. If the $(n+1)$-th layer is activated, then by the chain rule, because the derivative of all common activation functions exists a.e. and is bounded by $0$ and $s$, and because $\nabla y_i^{(n)} \succeq 0$, we have that a.e. $\nabla x_i^{(n+1)} \succeq 0$, $\|\nabla x_i^{(n+1)}\|_1 \leq s\|\nabla y_i^{(n+1)}\|_1 \leq s^{l+1}$. This concludes the induction.

□

Combining Theorem 3 and Lemma A.2, we obtain the following result.

**Theorem A.7.** *Existing DEP-based networks are not universal approximators.*

*Proof.* We prove that they are not universal approximators on any compact, convex domain $\mathcal{B} \subseteq \mathbb{R}^d$ with non-empty interior, which is stronger that not being universal approximators on $\mathbb{R}^d$. Notice that by Theorem 3 we have that a.e. $\nabla y(\mathbf{x}) \succeq 0, \|\nabla y(\mathbf{x})\|_1 \leq 1$. Hence, $\|\nabla y(\mathbf{x})\|_1 \leq 1$ a.e. on $\mathbb{R}^n$. If we restrict $y$ on $\mathcal{B}$, we have a compact, convex, domain $\mathcal{B}$ with non-empty interior and a Lipschitz continuous $y(\mathbf{x})$ with $\|\nabla y(\mathbf{x})\|_1 \leq 1$ a.e. on $\mathcal{B}$. From Lemma A.2 it follows that DEP-based networks are not universal approximators on $\mathcal{B}$. $\square$

At this point, it is important to note that Theorem A.7 can be easily generalized to hold for any activation function with a bounded derivative, which also includes more exotic activation functions.

## B  Proposed networks are universal approximators

In this appendix we provide the proofs of Theorems 4 and 6 regarding the universality of our proposed networks.

**Theorem 4.** *If the domain of the input is compact, the Max-Plus-Min (MPM) network is a universal approximator.*

First, we give an overview of the ideas of the proof. The main idea is that, if $C$ is a large enough constant, then for some max-plus-min unit, by letting some input weights hover around $+C$, some other input weights hover around $-C$ and the rest to hover around 0, we effectively untangle the max-plus-min unit: The third type of weights are rendered idle, the first type of weights contribute only to the maximum, the second type of weights contribute only to the minimum, and after summing the result the $+C$ and $-C$ cancel out. This way, we can control which inputs contribute to the maximum, which to the nimimum, and which to neither. This way, we can do the following: 1) We can build functions of the form $\mathbf{a}^\top \mathbf{x} + b$, 2) We can split the network into parallel networks each of which build their own function, 3) we can feed some of them into a maximum, and the rest into a minimum, to build a function of the form $\max_{i \in [n]}(\mathbf{a}_i^\top \mathbf{x} + b_i) + \min_{i \in [n+m]-[n]}(\mathbf{a}_i^\top \mathbf{x} + b_i)$, which can be shown to be a universal approximator based on the proof of universality of the Maxout networks (Goodfellow et al., 2013).

*Proof.* A byproduct of the proof of universality of Maxout networks, is that the class of functions of the form $\max_{k \in [K]}(\mathbf{a}_k^\top \mathbf{x} + b_k) - \max_{m \in [M]}(\mathbf{c}_m^\top \mathbf{x} + d_m) = \max_{k \in [K]}(\mathbf{a}_k^\top \mathbf{x} + b_k) + \min_{m \in [M]}((-\mathbf{c})_m^\top \mathbf{x} - d_m) = \max_{k \in [K]}(\mathbf{a}_k^\top \mathbf{x} + b_k) + \min_{m \in [M]}(\tilde{\mathbf{c}}_m^\top \mathbf{x} + \tilde{d}_m)$ is a universal approximator on $\mathbb{R}^d$. Thus, it suffices to prove that over any compact domain, we can build any function of the above form using an MPM. We restrict ourselves further by assuming that $\mathbf{w}_0^{(n)} = \mathbf{m}_0^{(n)}$, for all layers $n$.

First, we show how we can build linear functions of the form $\mathbf{a}^\top \mathbf{x} + b$. Let a bounded domain $\mathcal{B}$ and let some $R > 0$ such that $\mathcal{B} \subseteq B_1(\mathbf{0}, R)$, where $B_1(\mathbf{0}, R)$ denotes the $\ell_1$ ball centered at the origin with $\ell_1$ radius $R$. Let $C > 0$ be a large constant to be determined. Write $\mathbf{a}^\top \mathbf{x} + b = a_1 x_1 + \ldots + a_d x_d + b$. First, we build each $a_i x_i$ term. Take the first max-plus-min layer to have $d$ outputs and $d$ inputs, with weights

$$w_{ii}^{(1)} = +C,$$

$$w_{ij}^{(1)} = 0, \quad i \neq j, j \neq 0,$$

$$w_{i0}^{(1)} = -C.$$

Then, for the $i$-th output we have that

$$x_i + w_{ii}^{(1)} = x_i + C > x_j + 0 = x_j + w_{ij}^{(1)}, \quad j \neq i,$$

$$x_i + w_{ii}^{(1)} = x_i + C > -C = w_{i0}^{(1)},$$

$$x_j + w_{ij}^{(1)} = x_j > -C = w_{i0}^{(n)}$$

$$\Rightarrow w_{i0}^{(1)} \vee \max_{j \in [d]}(x_j + w_{ij}^{(1)}) = x_i + C,$$

$$w_{i0}^{(1)} \wedge \min_{j \in [d]}(x_j + w_{ij}^{(1)}) = -C$$

$$\Rightarrow y_i^{(1)} = (w_{i0}^{(1)} \vee \max_{j \in [d]}(x_j + w_{ij}^{(1)})) + (w_{i0}^{(1)} \wedge \min_{j \in [d]}(x_j + w_{ij}^{(1)})) = x_i + C - C = x_i$$

For the inequalities to hold, it suffices that $C > \|\mathbf{x}\|_1$. Take the activation of the first layer to be

$$\alpha_i^{(1)} = a_i$$

Then, for the $i$-th activated output we have that

$$x_i^{(1)} = a_i y_i^{(1)} = a_i x_i \tag{1}$$

Next, we will start summing up the terms. First, we sum up $a_{d-1}x_{d-1}$ and $a_d x_d$. Take a max-plus-min layer with $d-1$ outputs and $d$ inputs. The weights of the second layer are as follows:

$$w_{(d-1)(d-1)}^{(2)} = +C, \quad w_{(d-1)d}^{(2)} = -C, \quad w_{(d-1)j}^{(2)} = 0, \forall j < d-1,$$

$$w_{ii}^{(2)} = +C, \quad i < d-1,$$

$$w_{ij}^{(2)} = 0, \quad i \neq j, i < d-1, j \neq 0$$

$$w_{i0}^{(2)} = -C, \quad i < d-1.$$

Similarly with before, for $i < d-1$ we have that

$$x_i^{(1)} + w_{ii}^{(2)} = x_i^{(1)} + C > x_j^{(1)} + 0 = x_j^{(1)} + w_{ij}^{(2)}, \quad j \neq i,$$

$$x_i^{(1)} + w_{ii}^{(2)} = x_i^{(1)} + C > -C = w_{i0}^{(2)},$$

$$x_j^{(1)} + w_{ij}^{(2)} = x_j^{(1)} > -C = w_{i0}^{(2)}$$

$$\Rightarrow w_{i0}^{(2)} \vee \max_{j \in [n]}(x_j^{(1)} + w_{ij}^{(2)}) = x_i^{(1)} + C,$$

$$w_{i0}^{(2)} \wedge \min_{j \in [n]}(x_j^{(1)} + w_{ij}^{(2)}) = -C$$

$$\Rightarrow y_i^{(2)} = (w_{i0}^{(2)} \vee \max_{j \in [d]}(x_j^{(1)} + w_{ij}^{(2)})) + (w_{i0}^{(2)} \wedge \min_{j \in [d]}(x_j^{(1)} + w_{ij}^{(2)})) = x_i^{(1)} + C - C = x_i^{(1)} = a_i x_i$$

For $i = d-1$ we get the sum of the terms $a_{d-1}x_{d-1}, a_d x_d$. We have that

$$x_{d-1}^{(1)} + w_{(d-1)(d-1)}^{(2)} = x_{d-1}^{(1)} + C > x_j^{(1)} + 0 = x_j^{(1)} + w_{(d-1)j}^{(2)}, j < d-1,$$

$$x_{d-1}^{(1)} + w_{(d-1)(d-1)}^{(2)} = x_{d-1}^{(1)} + C > x_d^{(1)} - C = x_d^{(1)} + w_{(d-1)d}^{(2)},$$

$$x_{d-1}^{(1)} + w_{(d-1)(d-1)}^{(2)} = x_{d-1}^{(1)} + C > 0 = w_{(d-1)0}^{(2)},$$

$$\Rightarrow w_{(d-1)0}^{(2)} \vee \max_{j \in [d]}(x_{d-1}^{(1)} + w_{(d-1)j}^{(2)}) = x_{d-1}^{(1)} + C,$$

$$x_d^{(1)} + w_{(d-1)d}^{(2)} = x_d^{(1)} - C < x_j^{(1)} + 0 = x_j^{(1)} + w_{(d-1)j}^{(2)}, j < d-1,$$

$$x_d^{(1)} + w_{(d-1)d}^{(2)} = x_d^{(1)} - C < x_{d-1}^{(1)} + C = x_{d-1}^{(1)} + w_{(d-1)(d-1)}^{(2)},$$

$$x_d^{(1)} + w_{(d-1)d}^{(2)} = x_d^{(1)} - C < 0 = w_{(d-1)0}^{(2)},$$

$$\Rightarrow w_{(d-1)0}^{(2)} \wedge \min_{j \in [d]}(x_{d-1}^{(1)} + w_{(d-1)j}^{(2)}) = x_d^{(1)} - C,$$

$$\Rightarrow y_{d-1}^{(2)} = (w_{(d-1)0}^{(2)} \vee \max_{j \in [d]}(x_{d-1}^{(1)} + w_{(d-1)j}^{(2)})) + (w_{(d-1)0}^{(2)} \wedge \min_{j \in [d]}(x_{d-1}^{(1)} + w_{(d-1)j}^{(2)})) = x_{d-1}^{(1)} + C + x_d^{(1)} - C$$

$$= x_{d-1}^{(1)} + x_d^{(1)} = a_{d-1}x_{d-1} + a_d x_d$$

For the inequalities to hold, it suffices that $C > \|\mathbf{x}^{(1)}\|_1 = |a_1 x_1| + \ldots + |a_d x_d|$. Take the activation of the second layer to be

$$\alpha_i^{(2)} = 1$$

Then, for the $i$-th activated output we have that

$$x_i^{(2)} = y_i^{(2)} = a_i x_i, \quad i < d - 1, \quad x_{d-1}^{(2)} = y_{d-1}^{(2)} = a_{d-1}x_{d-1} + a_d x_d \tag{2}$$

We repeat this process for a total of $d$ layers ($1^{st}$ layer for multiplication with $a_i$, the rest for adding the terms) until all the terms $a_i x_i$ have been summed up and we end up with a single output. To add the bias term $b$, we add one final layer with 1 output and 1 input, with weights defined as follows:

$$w_{11}^{(d+1)} = b, \quad w_{10}^{(d+1)} = 0$$

Then, we have that

$$x_1^{(d+1)} = y_1^{(d+1)} = (w_{10}^{(d+1)} \vee (x_1^{(d)} + w_{11}^{(d+1)})) + (w_{10}^{(d+1)} \wedge (x_1^{(d)} + w_{11}^{(d+1)}))$$

$$= w_{10}^{(d+1)} + (x_1^{(d)} + w_{11}^{(d+1)}) = 0 + x_1^{(d)} + b = a_1 x_1 + \ldots + a_d x_d + b \tag{3}$$

For all the inequalities to hold, it suffices that the following hold:

$$C > \|\mathbf{x}\|_1,$$

$$C > \|\mathbf{x}^{(1)}\|_1 = |a_1 x_1| + \ldots + |a_d x_d|,$$

$$C > \|\mathbf{x}^{(2)}\|_1 = |a_1 x_1| + \ldots + |a_{d-2}x_{d-2}| + |a_{d-1}x_{d-1} + a_d x_d|,$$

$$\vdots$$

$$C > \|\mathbf{x}^{(d)}\|_1 = |a_1 x_1 + \ldots + a_d x_d|$$

For the above to hold, it suffices that the following holds

$$C > (1 + \max_{i \in [d]} |a_i|)\|\mathbf{x}\|_1$$

We simply choose $C = (1 + \max_{i \in [d]} |a_i|)R$ and we are done.

So far we have shown how to built the single output function $\mathbf{a}^\top \mathbf{x} + b$. We can also build networks with multiple outputs each of which is an affine function of the input.

Say we want to build a network which outputs the functions $\mathbf{a}_1^\top \mathbf{x} + b_1, \ldots, \mathbf{a}_K^\top \mathbf{x} + b_K$. We can use the exact same method as the one used for the single function $\mathbf{a}^\top \mathbf{x} + b$. Specifically, we build "parallel" networks, each calculating a single function independent of each other.

Again, let a bounded domain $\mathcal{B}$ and let some $R > 0$ such that $\mathcal{B} \subseteq B_1(\mathbf{0}, R)$, where $B_1(\mathbf{0}, R)$ denotes the $\ell_1$ ball centered at the origin with $\ell_1$ radius $R$. Let $C > 0$ be a large constant to be determined. Write $\mathbf{a}_k^\top \mathbf{x} + b_k = a_{k,1}x_1 + \ldots + a_{k,d}x_d + b_k$. First, we build each $a_{k,i}x_i$ term for all $k \in [K], i \in [d]$. Take the first max-plus-min layer to have $Kd$ outputs and $d$ inputs, with weights

$$w_{(i+(k-1)d),i}^{(1)} = +C,$$

$$w_{(i+(k-1)d),j}^{(1)} = 0, \quad i \neq j, j \neq 0,$$

$$w_{(i+(k-1)d),0}^{(1)} = -C,$$

where $k \in [K]$. Then, similarly to before, if $C > \|\mathbf{x}\|_1$ we have that

$$y_{i+(k-1)d}^{(1)} = x_i + C - C = x_i,$$

where $k \in [K]$. For the activations, take

$$\alpha_{i+(k-1)d}^{(1)} = a_{k,i}, \quad k \in [K], i \in [d].$$

Then, for the $(i + (k-1)d)$-th activated output we have that

$$x_{i+(k-1)d}^{(1)} = a_{ki} y_{i+(k-1)d}^{(1)} = a_{k,i} x_i, \quad k \in [K], i \in [d]$$

Next, we start summing up the terms. First, we take the sums $a_{1,(d-1)} x_{d-1} + a_{1,d} x_d$, $\ldots$, $a_{K,(d-1)} x_{d-1} + a_{K,d} x_d$. We use the same method as the one used previously. Take a max-plus-min layer with $K(d-1)$ outputs and $Kd$ inputs. The weights of the second layer are as follows:

$$w_{k(d-1),k(d-1)}^{(2)} = +C, \quad w_{k(d-1),kd}^{(2)} = -C, \quad w_{k(d-1),j}^{(2)} = 0, \forall k \in [K], j \neq kd, k(d-1),$$

$$w_{i+(k-1)(d-1),i+(k-1)(d-1)}^{(2)} = +C, \quad k \in [K], i < d-1,$$

$$w_{i+(k-1)(d-1),j}^{(2)} = 0, \quad k \in [K], i < d-1, j \neq i+(k-1)(d-1), j \neq 0,$$

$$w_{i+(k-1)(d-1),0}^{(2)} = -C, \quad k \in [K], i < d-1,$$

Similarly to before, if $C > \|\mathbf{x}^{(1)}\|_1 = |a_{11} x_1| + \ldots + |a_{1d} x_d| + \ldots + |a_{Kd} x_d|$, then we have

$$y_{i+(k-1)(d-1)}^{(2)} = a_{ki} x_i, \quad k \in [K], i < d-1$$

$$y_{k(d-1)}^{(2)} = a_{k,(d-1)} x_{d-1} + a_{k,d} x_d, \quad k \in [K]$$

Take the activation of the second layer to be $a_{i+(k-1)(d-1)}^{(2)} = 1, \quad i \in [d-1], k \in [K]$. Then, for the $i$-th activated output we have that

$$x_{i+(k-1)(d-1)}^{(2)} = y_{i+(k-1)(d-1)}^{(2)} = a_{k,i} x_i, \quad i < d-1, k \in [K],$$

$$x_{k(d-1)}^{(2)} = y_{k(d-1)}^{(2)} = a_{k,(d-1)} x_{d-1} + a_{k,d} x_d, \quad k \in [K]$$

We repeat this process for a total of $d$ layers until all the terms have been summed up and we end up with $K$ outputs. To add the bias terms $b_k$ we add one final layer with $K$ outputs and $K$ inputs, with weights defined as follows:

$$w_{kk}^{(d+1)} = b_k + C, \quad w_{k0}^{(d+1)} = -C, \quad w_{kj}^{(d+1)} = 0, j \neq k$$

Notice the slight deviation from the previous method. If $C > \|\mathbf{x}^{(d)}\|_1 + \max_{k \in [K]} |b_k|$, then we have that

$$x_k^{(d+1)} = y_k^{(d+1)} = a_{k1} x_1 + \ldots + a_{kd} x_d + b_k$$

For the inequalities to hold, it suffices that the following hold:

$$C > \|\mathbf{x}\|_1,$$

$$C > \|\mathbf{x}^{(1)}\|_1 = |a_{11} x_1| + \ldots + |a_{1d} x_d| + \ldots + |a_{Kd} x_d|,$$

$$C > \|\mathbf{x}^{(2)}\|_1 = |a_{11} x_1| + \ldots + |a_{1(d-1)} x_{d-1} + a_{1d} x_d| + \ldots + |a_{K(d-1} x_{d-1} + a_{Kd} x_d|,$$

$$\vdots$$

$$C > \|\mathbf{x}^{(d)}\|_1 + \max_{k \in [K]} |b_k| = |a_{11}x_1 + \ldots + a_{1d}x_d| + \ldots + |a_{K1}x_1 + \ldots + a_{Kd}x_d| + \max_{k \in [K]} |b_k|$$

For the above to hold, it suffices that the following holds

$$C > (1 + K \max_{i \in [d], k \in [K]} |a_{ki}|) \|\mathbf{x}\|_1 + \max_{k \in [K]} |b_k|$$

We simply take $C > (1 + K \max_{i \in [d], k \in [K]} |a_{ki}|) R + \max_{k \in [K]} |b_k|$ and we are done.

To finish the proof, we will prove that we can build functions of the form $\max_{k \in [K]}(\mathbf{a}_k^\top \mathbf{x} + b_k) + \min_{m \in [M]}(\mathbf{c}_m^\top \mathbf{x} + d_m)$. First, using the previous method, build a network with $K + M$ outputs, of which the first $K$ outputs are the functions $\mathbf{a}_k^\top \mathbf{x} + b_k, k \in [K]$, and the last $M$ outputs are the functions $\mathbf{c}_m^\top \mathbf{x} + d_m, m \in [M]$. Activate the final layer with an identity activation. Let $C' > 0$ be a large constant to be determined. Place another final layer with 1 output and $K + M$ inputs. The weights of the new final layer are as follows

$$w_{10}^{(d+2)} = 0, \quad w_{1i}^{(d+2)} = +C', i \in [K], \quad w_{1(K+i)}^{(d+2)} = -C', i \in [M]$$

Then, we have that

$$\mathbf{a}_k^\top \mathbf{x} + b_k + C' > 0 > \mathbf{c}_m^\top \mathbf{x} + d_m, -C', k \in [K], m \in [M],$$

and hence, the output will be given by

$$x_1^{(d+2)} = y_1^{(d+2)} = \max_{k \in [K]}(\mathbf{a}_k^\top \mathbf{x} + b_k + C) + \min_{m \in [M]}(\mathbf{c}^\top \mathbf{x} + d_k - C) = \max_{k \in [K]}(\mathbf{a}_k^\top \mathbf{x} + b_k) + \min_{m \in [M]}(\mathbf{c}^\top \mathbf{x} + d_k).$$

For the above inequalities to hold, it suffices that

$$C' > (1 + (K + M) \max(\max_{i \in [d], k \in [K]} |a_{ki}|, \max_{i \in [d], m \in [M]} |c_{mi}|)) R + \max(\max_{k \in [K]} |b_k|, \max_{m \in [M]} |d_m|).$$

This finishes the proof. $\qquad\square$

We should point out that for the purpose of simplifying the proof, we summed up the terms one-by-one, resulting in $\Theta(d)$ layers. However, one can easily sum up the terms in a hierchical fashion, resulting in $\Theta(\log(d))$ layers.

The above proof can also be extended to the RMPM networks. It suffices to adjust steps (1), (2), and (3) to work with residual connections. We avoid another lengthy derivation, and instead show how these steps can be adjusted.

**Corollary 5.** *If the domain of the input is compact (i.e., bounded and closed), and assuming no residual skip connection for the output layer, the Residual-Max-Plus-Min (RMPM) network is a universal approximator.*

*Proof.* To avoid repetition with Theorem 4, in this proof wherever dots (...) are used in equations, they indicate arguments that are dominated and never chosen by max or min. Also, in contrast to Theorem 4, here the biases of the max and the min are not taken to be the same.

(1) sets the weights of the MPM appropriately so that the first layer performs scaling for each input. For the RMPM, if the first layer contains a residual skip connection, then (1) becomes

$$x_i^{(1)} = x_i + \tilde{a}_i x_i = (1 + \tilde{\alpha}_i) x_i.$$

For the RMPM we select $\tilde{a}_i = a_i - 1$, and then the outputs of the first layer are matching by using the same large constants.

(2) performs either a pass-through for a variable, or pair-wise addition of others. Since we now have skip connections, we can instead set outputs to 0 or allow a pass-through respectively to get the same behavior. In particular, for $i < d - 1$, instead of setting $w_{ii}^{(2)} = +C$, we set $w_{i0}^{(2)} = C \neq m_{i0}^{(2)} = -C$, and after including the residual skip we get

$$y_i^{(2)} = x_i^{(1)} + (C \vee \max_{j \in [d]}(\ldots)) + (-C \wedge \min_{j \in [d]}(\ldots)) = x_i^{(1)} + C - C = x_i^{(1)},$$

and we recover the same output as the MPM. For $i = d - 1$, instead of setting $w^{(2)}_{(d-1)(d-1)} = +C$, we set $w^{(2)}_{(d-1)0} = +C \neq m^{(2)}_{(d-1)0} = 0$, and after including the residual skip we get

$$y^{(2)}_{d-1} = x^{(1)}_{d-1} + (+C \vee \max_{j \in [d]}(\ldots)) + (0 \wedge \min(x^{(1)}_d - C, \ldots)) = x^{(1)}_{d-1} + C + x^{(1)}_d - C,$$

and we again recover the same output as the MPM. Choosing the same large constants suffices.

Finally, (3) adds a bias to the input. Instead of setting $w^{(d+1)}_{11} = b$, we set $w^{(d+1)}_{10} = C + b \neq m^{(d+1)}_{10} = -C$, and after including the residual skip we get

$$x^{(d+1)}_1 = x^{(d)}_i + (b + C \vee \max(\ldots)) + (-C \wedge \min(\ldots)) = x^{(d)}_i + b,$$

and we recover the same output as the MPM. Choosing the same large constants suffices.

With (1), (2), and (3) properly adjusted, the remainder of the proof of Theorem 4 can also be adjusted since it uses the same exact decompositions (with the exception of the final merging layer, which we took by assumption to not have a residual skip connection). $\qquad\square$

Before we proceed with our final theorem on the universality of Hybrid-MLP, we are going to need the following two lemmas regarding ReLU and Maxout networks.

**Lemma B.1.** *Suppose we are given a ReLU activated fully connected linear network with $L$ linear layers $(A^{(n)}, b^{(n)}), n \in [L]$, which can be recursively defined as follows*

$$\mathbf{y}^{(n)} = \mathbf{A}^{(n)} \mathbf{x}^{(n-1)} + \mathbf{b}^{(n)}, \quad n \in [L],$$

$$\mathbf{x}^{(n)} = \max(\mathbf{0}, \mathbf{y}^{(n)}), \quad n \in [L-1], \quad \mathbf{x}^{(L)} = \mathbf{y}^{(L)}$$

*where max is taken element-wise. Then, for every $n \in [L]$ it holds that*

$$\|\mathbf{y}^{(n)}\|_1 \leq \left(\prod_{i=1}^n \|\mathbf{A}^{(i)}\|_1\right) \|\mathbf{x}\|_1 + \sum_{i=1}^n \left(\prod_{j=i+1}^n \|\mathbf{A}^{(j)}\|_1\right) \|\mathbf{b}^{(i)}\|_1.$$

*Proof.* We will prove that for every $n \in \{0, \ldots, L-1\}$ it holds that

$$\|\mathbf{x}^{(n)}\|_1 \leq \left(\prod_{i=1}^n \|\mathbf{A}^{(i)}\|_1\right) \|\mathbf{x}\|_1 + \sum_{i=1}^n \left(\prod_{j=i+1}^n \|\mathbf{A}^{(j)}\|_1\right) \|\mathbf{b}^{(i)}\|_1.$$

For $n = 0$ the inequality obviously holds. For $n \in [L-1]$ we have that

$$\|\mathbf{x}^{(n)}\|_1 = \|\max(\mathbf{0}, \mathbf{y}^{(n)})\|_1 = \sum_j |\max(0, y^{(n)}_j)| \leq \sum_j |y^{(n)}_j| = \|\mathbf{y}^{(n)}\|_1 = \|\mathbf{A}^{(n)}\mathbf{x}^{(n-1)} + \mathbf{b}^{(n)}\|_1.$$

By the sub-additivity of vector norm $\|\cdot\|_1$ and sub-multiplicativity of matrix operator norm $\|\cdot\|_1$ we have that

$$\|\mathbf{x}^{(n)}\|_1 \leq \|\mathbf{A}^{(n)}\|_1 \|\mathbf{x}^{(n-1)}\|_1 + \|\mathbf{b}^{(n)}\|_1$$

By expanding the recurrence, we have that

$$\|\mathbf{x}^{(n)}\|_1 \leq \|\mathbf{A}^{(n)}\|_1 \|\mathbf{x}^{(n-1)}\|_1 + \|\mathbf{b}^{(n)}\|_1$$

$$\leq \|\mathbf{A}^{(n)}\|_1 \|\mathbf{A}^{(n-1)}\|_1 \|\mathbf{x}^{(n-2)}\|_1 + \|\mathbf{b}^{(n)}\|_1 + \|\mathbf{A}^{(n)}\|_1 \|\mathbf{b}^{(n-1)}\|_1$$

$$\leq \ldots \leq \left(\prod_{i=1}^n \|\mathbf{A}^{(i)}\|_1\right) \|\mathbf{x}\|_1 + \sum_{i=1}^n \left(\prod_{j=i+1}^n \|\mathbf{A}^{(j)}\|_1\right) \|\mathbf{b}^{(i)}\|_1.$$

Then, for every $n \in [L]$ we have that

$$\|\mathbf{y}^{(n)}\|_1 \leq \|\mathbf{A}^{(n)}\|_1 \|\mathbf{x}^{(n-1)}\|_1 + \|\mathbf{b}^{(n)}\|_1,$$

and the result follows immediately. $\qquad\square$

**Lemma B.2.** *Suppose we are given a Maxout fully connected linear network with $L$ layers and pooling of $P$ which can be recursively defined as follows*

$$\mathbf{y}_p^{(n)} = \mathbf{A}_p^{(n)}\mathbf{x}^{(n-1)} + \mathbf{b}_p^{(n)}, \quad n \in [L-1], p \in [P],$$

$$\mathbf{x}^{(n)} = \max_{p \in [P]}(\mathbf{y}_p^{(n)}), \quad n \in [L-1], \quad \mathbf{x}^{(L)} = \mathbf{y}^{(L)} = \mathbf{A}^{(L)}\mathbf{x}^{(L-1)} + \mathbf{b}^{(L)}$$

*where max is taken element-wise. Then, for every $n \in [L-1]$ it holds that*

$$\sum_{p=1}^{P} \|\mathbf{y}_p^{(n)}\|_1 \leq \left( \prod_{i=1}^{n} \sum_{p \in [P]} \|\mathbf{A}_p^{(i)}\|_1 \right) \|\mathbf{x}\|_1 + \sum_{i=1}^{n} \left( \prod_{j=i+1}^{n} \sum_{p \in [P]} \|\mathbf{A}_p^{(j)}\|_1 \right) \left( \sum_{p \in [P]} \|\mathbf{b}_p^{(i)}\|_1 \right)$$

*Proof.* First, we will prove that for every $n \in \{0, \ldots, L-1\}$ it holds that

$$\|\mathbf{x}^{(n)}\|_1 \leq \left( \prod_{i=1}^{n} \sum_{p \in [P]} \|\mathbf{A}_p^{(i)}\|_1 \right) \|\mathbf{x}\|_1 + \sum_{i=1}^{n} \left( \prod_{j=i+1}^{n} \sum_{p \in [P]} \|\mathbf{A}_p^{(j)}\|_1 \right) \left( \sum_{p \in [P]} \|\mathbf{b}_p^{(i)}\|_1 \right).$$

For $n = 0$ the inequality obviously holds. For $n \in [L-1]$ we have that

$$\|\mathbf{x}^{(n)}\|_1 = \| \max_{p \in [P]}(\mathbf{y}_p^{(n)})\|_1 = \sum_j | \max_{p \in [P]}(y_{pj}^{(n)})| \leq \sum_j \sum_{p \in [P]} |y_{pj}^{(n)}| = \sum_{p \in [P]} \|\mathbf{y}_p^{(n)}\|_1$$

$$= \sum_{p \in [P]} \|\mathbf{A}_p^{(n)}\mathbf{x}^{(n-1)} + \mathbf{b}_p^{(n)}\|_1$$

By the sub-additivity of vector norm $\| \cdot \|_1$ and sub-multiplicativity of matrix operator norm $\| \cdot \|_1$ we have that

$$\|\mathbf{x}^{(n)}\|_1 \leq \left( \sum_{p \in [P]} \|\mathbf{A}_p^{(n)}\|_1 \right) \|\mathbf{x}^{(n)}\|_1 + \left( \sum_{p \in [P]} \|\mathbf{b}_p^{(n)}\|_1 \right)$$

By expanding the recurrence, we have that

$$\|\mathbf{x}^{(n)}\|_1 \leq \left( \sum_{p \in [P]} \|\mathbf{A}_p^{(n)}\|_1 \right) \|\mathbf{x}^{(n)}\|_1 + \left( \sum_{p \in [P]} \|\mathbf{b}_p^{(n)}\|_1 \right)$$

$$\leq \left( \sum_{p \in [P]} \|\mathbf{A}_p^{(n)}\|_1 \right) \left( \sum_{p \in [P]} \|\mathbf{A}_p^{(n-1)}\|_1 \right) \|\mathbf{x}^{(n-2)}\|_1 + \left( \sum_{p \in [P]} \|\mathbf{b}_p^{(n)}\|_1 \right)$$

$$+ \left( \sum_{p \in [P]} \|\mathbf{A}_p^{(n)}\|_1 \right) \left( \sum_{p \in [P]} \|\mathbf{b}_p^{(n-1)}\|_1 \right)$$

$$\leq \ldots \leq \left( \prod_{i=1}^{n} \sum_{p \in [P]} \|\mathbf{A}_p^{(i)}\|_1 \right) \|\mathbf{x}\|_1 + \sum_{i=1}^{n} \left( \prod_{j=i+1}^{n} \sum_{p \in [P]} \|\mathbf{A}_p^{(j)}\|_1 \right) \left( \sum_{p \in [P]} \|\mathbf{b}_p^{(i)}\|_1 \right).$$

Then, for every $n \in [L-1]$ we have that

$$\sum_{p \in [P]} \|\mathbf{y}_p^{(n)}\|_1 \leq \left( \sum_{p \in [P]} \|\mathbf{A}_p^{(n)}\|_1 \right) \|\mathbf{x}^{(n-1)}\|_1 + \left( \sum_{p \in [P]} \|\mathbf{b}_p^{(n)}\|_1 \right),$$

and the result follows immediately. □

**Theorem 6.** *If the domain of the input is compact, the Hybrid-MLP is a universal approximator. In fact, any fully connected ReLU or maxout network is a special case of the Hybrid-MLP.*

*Proof.* We will show that for any ReLU and Maxout networks, if the domain $\mathcal{B}$ is bounded (i.e. there exists $R$ such that $\mathcal{B} \subseteq B_1(\mathbf{0}, R)$), then there exists a Hybrid-MLP network that gives the same output. By our construction it will immediately follow that for bounded domains, ReLU and Maxout networks are special cases of the Hybrid-MLP.

First, we focus on ReLU activated networks. Take a network as defined in Lemma B.1. Layer-by-layer, we will replace its ReLU activations with our proposed morphological layer. Pick an activated layer $n \in [L-1]$ and some large constant $C > 0$ to be determined. Remove the ReLU activations and replace them by our morphological layer:

$$\mathbf{y}^{(n)} = \mathbf{A}^{(n)} \mathbf{x}^{(n-1)} + \mathbf{b}^{(n)},$$

$$\mathbf{x}^{(n)} = \left( \mathbf{w}_i^{(n)} \vee \mathbf{W}^{(n)} \boxplus \mathbf{y}^{(n)} \right) + \left( \mathbf{m}_i^{(n)} \wedge \mathbf{W}^{(n)} \boxplus' \mathbf{y}^{(n)} \right).$$

The weights of the morphological layer are defined as follows:

$$w_{i0}^{(n)} = +C, \quad m_{i0}^{(n)} = -C,$$

$$w_{ii}^{(n)} = +C,$$

$$w_{ij}^{(n)} = 0, j \neq i.$$

Then, for the $i$-th output, we have that

$$x_i^{(n)} = (C \vee (y_i^{(n)} + C) \vee \max_{j \neq i}(y_j^{(n)})) + ((-C) \wedge (y_i^{(n)} + C) \wedge \min_{j \neq i}(y^{(n)})) = (C \vee (y_i^{(n)} + C)) + (-C)$$

$$= (C + \max(0, y_i^{(n)})) + (-C) = \max(0, y_i^{(n)}),$$

and the output is the same as the ReLU network. For the above to hold, it suffices that $C > \|\mathbf{y}^{(n)}\|_1$. According to Lemma B.1, it suffices to pick

$$C > \left( \prod_{i=1}^n \|\mathbf{A}^{(n)}\|_1 \right) R + \sum_{i=1}^n \left( \prod_{j=i+1}^n \|\mathbf{A}^{(i)}\|_1 \right) \|\mathbf{b}^{(n)}\|_1.$$

We continue with Maxout networks. Take a network as defined in Lemma B.2. If we concatenate the vectors $\mathbf{y}_p^{(n)}$, then maxout networks are effectively a maxpooling layer. We will replace this maxpooling layer with our morphological layer.

Pick a maxout-activated layer $n \in [L-1]$ and some large constant $C > 0$ to be determined. Concatenate the vectors $\mathbf{y}_p^{(n)}$ to form the vector $\mathbf{g}^{(n)}$. Suppose that the $n$-th layer has output size $N$. Then, the vector $\mathbf{g}^{(n)}$ has dimension $NP$. The $i$-th output $x_i^{(n)}$ will be given by

$$x_i^{(n)} = \max_{j=i,i+N,\ldots,i+(P-1)N}(g_j^{(n)})$$

Remove the maxpooling layer, and replace it by our morphological layer of output size $N$ and input size $NP$:

$$\mathbf{y}_p^{(n)} = \mathbf{A}_p^{(n)} \mathbf{x}^{(n-1)} + \mathbf{b}^{(n)},$$

$$\mathbf{g}^{(n)} = \text{concat}_{p \in [P]}(\mathbf{y}_p^{(n)}),$$

$$\mathbf{x}^{(n)} = \left( \mathbf{w}_i^{(n)} \vee \mathbf{W}^{(n)} \boxplus \mathbf{g}^{(n)} \right) + \left( \mathbf{m}_i^{(n)} \wedge \mathbf{W}^{(n)} \boxplus' \mathbf{g}^{(n)} \right).$$

The weights of the morphological layer are defined as follows:

$$w_{i,0}^{(n)} = +C, m_{i,0}^{(n)} = -C, \quad i \in [N],$$

$$w_{i,i+(k-1)N}^{(n)} = +C, \quad i \in [N], k \in [P],$$

$$w_{i,j}^{(n)} = 0, \quad j \neq i + (k-1)N, \text{for some } k \in [P]$$

Then, for the $i$-th output, we have that

$$x_i^{(n)} = \left( -C \vee \max_{j=i,\dots,i+(P-1)N} (g_j^{(n)} + C) \vee \max_{j \notin \{i,\dots,i+(P-1)N\}} (g^{(n)}) \right)$$

$$+ \left( -C \wedge \min_{j=i,\dots,i+(P-1)N} (g_j^{(n)} + C) \wedge \min_{j \notin \{i,\dots,i+(P-1)N\}} (g^{(n)}) \right)$$

$$= \left( \max_{j=i,\dots,i+(P-1)N} (g_j^{(n)} + C) \right) + (-C) = \max_{j=i,\dots,i+(P-1)N} (g_j^{(n)}),$$

and the output is the same as the Maxout network. For the above to hold, it suffices that $C > \|\mathbf{g}^{(n)}\|_1 = \sum_{p \in [P]} \|\mathbf{y}_p^{(n)}\|_1$. According to Lemma B.2 it suffices to pick

$$C > \left( \prod_{i=1}^{n} \sum_{p \in [P]} \|\mathbf{A}_p^{(i)}\|_1 \right) R + \sum_{i=1}^{n} \left( \prod_{j=i+1}^{n} \sum_{p \in [P]} \|\mathbf{A}_p^{(j)}\|_1 \right) \left( \sum_{p \in [P]} \|\mathbf{b}_p^{(i)}\|_1 \right).$$

So far we have proven that for bounded domains, we can build any ReLU or Maxout network with a Hyrbid-MLP. And in fact by our construction, ReLU and Maxout networks are special cases of the Hybrid-MLP. Since these networks are universal approximators, it follows that Hybrid-MLP will be a universal approximator on compact domains. □

## C  On the initialization of morphological neural networks

In this appendix, we study what makes training morphological networks difficult, apart from the lack of activations. We showcase that a good initialization is crucial, and that this is easier to achieve with our networks as opposed with previous architectures. Finally, we explain how we initialized the networks in our experiments. Most of the claims in this appendix are qualitative and lack formal proofs. However, we still believe that they serve an important purpose in the understanding of morphological networks.

**Optimization landscape.** First, we start with some examples that will allow us to gain an idea of the optimization landscapes we are faced against.

1. Consider an unbiased max-plus MP with 2 inputs and a single output, and an unbiased linear perceptron with 2 inputs and a single output. Consider 3 samples $(-1.7, 1; 2.3), (5, -2.2; 3.7), (1, 1; 4.7)$. The loss functions of the two percetrons are illustrated in Figure 3a. As we can see, for this particular dataset, the loss function of the MP (blue) has a smaller global minimum than the loss function of the linear perceptron (green). However, the loss function of the linear perceptron is a simple paraboloid, which is easy to optimize using gradient methods. On the other hand, the loss function of the MP is more complicated, comprised of pieces with their own local minima. For most initializations, solving with a gradient method would result in reaching these local minima, which are worse than the global minimum of the paraboloid loss of the linear perceptron.

2. Now consider a hybrid network, with a first layer of 2 unbiased max-plus MPs of 2 inputs, and a single output unbiased linear perceptron. We fix the weights of the output perceptron to 1, and the weights $w_{12} = w_{21} = 0$, study the loss function for variable weights $w_{11}, w_{22}$. Consider again 3 samples $(1.2, -2.4; 1.4), (-3.36, 2.34; 2.16), (-2.1, -1.5; 2.4)$. The loss function of the network is illustrated in Figure 3b. As we can see, for this dataset, the loss function has 9 local minima. Depending on the initialization, all local minima are likely to be reached, with most of them being a lot worse than the global minimum.

From the above, we understand that a good initialization is crucial for training morphological networks.

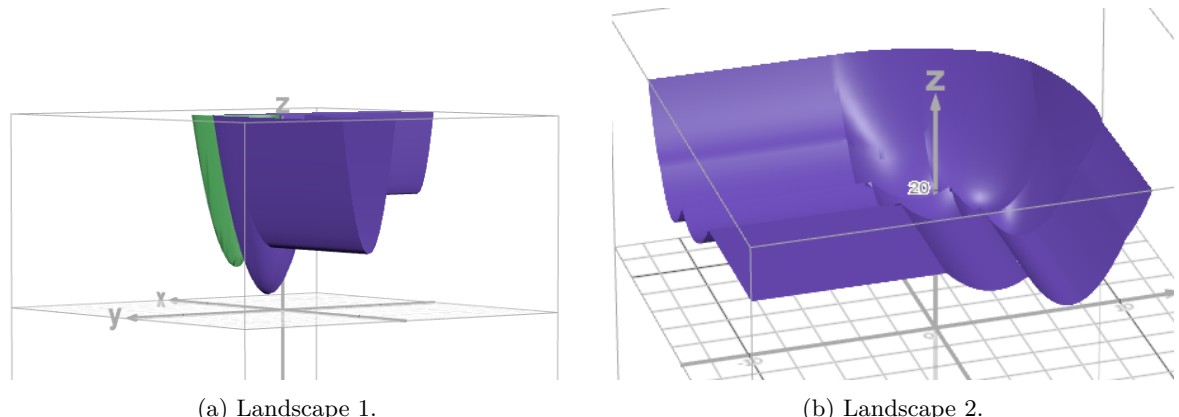

(a) Landscape 1.                     (b) Landscape 2.

Figure 3: Different landscapes of morphological networks.

Table 8: Linear perceptron trained on shifting its mean with batch size of 1.

| Epoch | $w_1$ | $w_2$ | $w_3$ | $w_4$ | $w_5$ | $w_6$ | $w_7$ | $w_8$ | $w_9$ | $w_{10}$ | mean | std |
|---|---|---|---|---|---|---|---|---|---|---|---|---|
| 1 | -0.1 | 0.1 | -0.1 | -0.1 | 0.1 | 0.1 | 0.1 | 0.1 | 0.1 | 0.1 | 0.04 | 0.10 |
| 2 | -0.04 | 0.17 | -0.16 | -0.08 | 0.18 | 0.12 | 0.16 | 0.12 | 0.14 | 0.17 | 0.08 | 0.12 |
| 3 | 0 | 0.23 | -0.21 | -0.05 | 0.25 | 0.12 | 0.21 | 0.14 | 0.17 | 0.23 | 0.11 | 0.15 |
| $\vdots$ | $\vdots$ | $\vdots$ | $\vdots$ | $\vdots$ | $\vdots$ | $\vdots$ | $\vdots$ | $\vdots$ | $\vdots$ | $\vdots$ | $\vdots$ | $\vdots$ |
| 998 | 9.98 | 9.99 | 9.99 | 9.98 | 9.97 | 9.97 | 10 | 9.95 | 9.97 | 9.97 | 9.98 | 0.01 |
| 999 | 9.98 | 9.99 | 9.99 | 9.99 | 9.97 | 9.97 | 10 | 9.95 | 9.97 | 9.97 | 9.98 | 0.01 |
| 1000 | 9.98 | 9.99 | 9.99 | 9.99 | 9.97 | 9.97 | 10 | 9.95 | 9.98 | 9.97 | 9.98 | 0.01 |

**Shift of the mean.** Let us now study the optimization of morphological networks from a different viewpoint. Let $d = 10$. Suppose we want to approximate the function $f(\mathbf{x}) = \mathbf{10}^\top \mathbf{x} = 10x_1 + \ldots + 10x_d$ using a linear perceptron $y_1(\mathbf{x}; \mathbf{w}) = \mathbf{w}^\top \mathbf{x}$. Also, suppose we want to approximate the function $g(\mathbf{x}) = \max_i(x_i + 10)$ using a max-plus MP $y_2(\mathbf{x}; \mathbf{m}) = \max_i(x_i + m_i)$. Obviously, both approximations are possible with zero error by letting $\mathbf{w} = \mathbf{m} = \mathbf{10} = (10, \ldots, 10)$.

For both problems, a good initialization of the weights $\mathbf{w}, \mathbf{m}$ would be with a mean of 10. However, such initializations are not common, because we can not know a-priori the distribution of the weights of the function we want to approximate. As such, let us initialize the weights to 0, and let us test how efficient both networks are at changing the mean of the distribution of their weights.

First, we use Adam with a batch size of 1 for 1000 epochs. In each epoch, we generate a training sample at random according to a standard distribution and our target functions and introduce no noise. Ideally, the networks should learn the weights $\mathbf{w} = \mathbf{m} = \mathbf{10}$, i.e. they should be able to shift the mean of the distribution of their weights, maintaining a standard deviation close to 0. The results are reported in Tables 8 and 9. We used a learning rate of 0.1 as this optimized the results of the MP. As we can see, the linear perceptron worked efficiently, meaning that it eventually managed to shift the mean to 10. The MP, on the other hand, did not manage to change the mean of the distribution of its weights to 10, and it also introduced a lot of variance in the values of the weights.

We repeat the experiment with a batch size of 100. The results are reported in Tables 10 and 11. Again, the linear perceptron excelled, learning the new mean successfully. The MP, while it performed better than the previous experiment, still failed to shift the mean of its weights to 10.

From the above, we understand that MPs have trouble at changing the mean of the distribution of their weights, especially when using smaller batch sizes. This is due to the sparse gradient signals that they produce. As such, we are in a unprecedented situation. Standard linear networks struggle with the initialization of the standard deviation of their weights, with a wrong initialization causing exploding and vanishing gradi-

Table 9: MP trained on shifting its mean with batch size of 1.

| Epoch | $w_1$ | $w_2$ | $w_3$ | $w_4$ | $w_5$ | $w_6$ | $w_7$ | $w_8$ | $w_9$ | $w_{10}$ | mean | std |
|---|---|---|---|---|---|---|---|---|---|---|---|---|
| 1 | 0. | 0.1 | 0. | -0. | -0. | 0. | -0. | -0. | -0. | 0. | 0.01 | 0.03 |
| 2 | 0.07 | 0.17 | 0. | -0. | -0. | 0. | -0. | -0. | -0. | 0. | 0.02 | 0.06 |
| 3 | 0.13 | 0.22 | 0. | -0. | -0. | 0.06 | -0. | -0. | -0. | 0. | 0.04 | 0.08 |
| ⋮ | ⋮ | ⋮ | ⋮ | ⋮ | ⋮ | ⋮ | ⋮ | ⋮ | ⋮ | ⋮ | ⋮ | ⋮ |
| 998 | 2.67 | 0.86 | 0.51 | 11.67 | 0.52 | 4.27 | 1.82 | 1.7 | 1.96 | 1.06 | 2.70 | 3.35 |
| 999 | 2.67 | 0.86 | 0.51 | 11.67 | 0.52 | 4.27 | 1.82 | 1.7 | 1.96 | 1.06 | 2.70 | 3.35 |
| 1000 | 2.67 | 0.86 | 0.51 | 11.66 | 0.52 | 4.27 | 1.82 | 1.7 | 1.96 | 1.06 | 2.70 | 3.34 |

Table 10: Linear perceptron trained on shifting its mean with batch size of 100.

| Epoch | $w_1$ | $w_2$ | $w_3$ | $w_4$ | $w_5$ | $w_6$ | $w_7$ | $w_8$ | $w_9$ | $w_{10}$ | mean | std |
|---|---|---|---|---|---|---|---|---|---|---|---|---|
| 1 | 0.1 | 0.1 | 0.1 | 0.1 | 0.1 | 0.1 | 0.1 | 0.1 | 0.1 | 0.1 | 0.10 | 0.00 |
| 2 | 0.2 | 0.2 | 0.2 | 0.19 | 0.2 | 0.2 | 0.18 | 0.2 | 0.2 | 0.2 | 0.20 | 0.01 |
| 3 | 0.3 | 0.29 | 0.3 | 0.29 | 0.3 | 0.3 | 0.26 | 0.3 | 0.3 | 0.29 | 0.29 | 0.01 |
| ⋮ | ⋮ | ⋮ | ⋮ | ⋮ | ⋮ | ⋮ | ⋮ | ⋮ | ⋮ | ⋮ | ⋮ | ⋮ |
| 998 | 10. | 10. | 10. | 10. | 10. | 10. | 10. | 10. | 10. | 10. | 10.00 | 0.00 |
| 999 | 10. | 10. | 10. | 10. | 10. | 10. | 10. | 10. | 10. | 10. | 10.00 | 0.00 |
| 1000 | 10. | 10. | 10. | 10. | 10. | 10. | 10. | 10. | 10. | 10. | 10.00 | 0.00 |

Table 11: MP trained on shifting its mean with batch size of 100.

| Epoch | $w_1$ | $w_2$ | $w_3$ | $w_4$ | $w_5$ | $w_6$ | $w_7$ | $w_8$ | $w_9$ | $w_{10}$ | mean | std |
|---|---|---|---|---|---|---|---|---|---|---|---|---|
| 1 | 0.1 | 0.1 | 0.1 | 0.1 | 0.1 | 0.1 | 0.1 | 0.1 | 0.1 | 0.1 | 0.10 | 0.00 |
| 2 | 0.2 | 0.2 | 0.2 | 0.2 | 0.2 | 0.2 | 0.2 | 0.2 | 0.2 | 0.2 | 0.20 | 0.00 |
| 3 | 0.29 | 0.3 | 0.3 | 0.3 | 0.3 | 0.3 | 0.3 | 0.3 | 0.29 | 0.3 | 0.30 | 0.00 |
| ⋮ | ⋮ | ⋮ | ⋮ | ⋮ | ⋮ | ⋮ | ⋮ | ⋮ | ⋮ | ⋮ | ⋮ | ⋮ |
| 998 | 6.03 | 5.03 | 10.75 | 6.52 | 7.28 | 10.67 | 6.7 | 10.61 | 6.67 | 6.59 | 7.69 | 2.14 |
| 999 | 6.03 | 5.03 | 10.75 | 6.52 | 7.28 | 10.66 | 6.7 | 10.62 | 6.67 | 6.59 | 7.69 | 2.14 |
| 1000 | 6.03 | 5.03 | 10.76 | 6.52 | 7.28 | 10.65 | 6.7 | 10.63 | 6.67 | 6.59 | 7.69 | 2.14 |

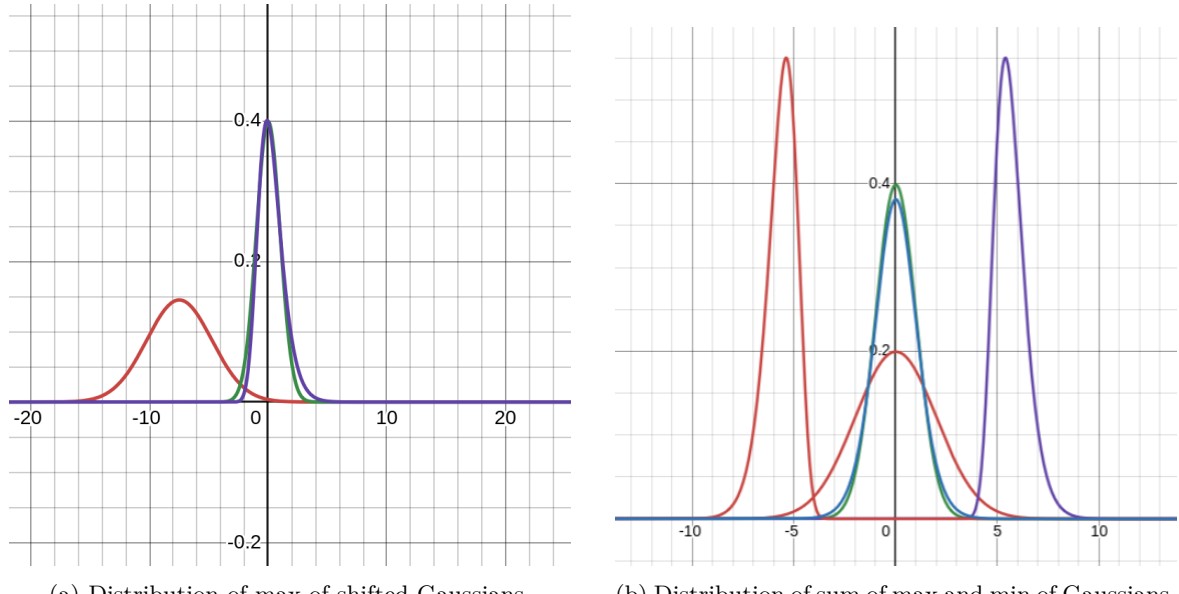

(a) Distribution of max of shifted Gaussians.

(b) Distribution of sum of max and min of Gaussians.

Figure 4: Distributions of MP and MPM networks.

ents (Glorot & Bengio, 2010; He et al., 2015). In contrast, morphological networks struggle with the wrong initialization of the mean of their weights.

Based on the above, we conclude that attention has to be paid when initializing morphological networks. We have to ensure that the weights of each MP of the network are initialized so that their mean will not have to change much throughout training. This remark is not limited only to initialization, but also throughout training: During training we should not make changes that require an MP to drastically change the mean of its weights.

**Why $\lambda = 1/2$?**   In the main body, we argued that choosing $\lambda \neq 1/2$ (or allowing $\lambda$ to be learnable) hinders trainability, leading us to propose the sum of the maximum and minimum as a stable alternative. Here, we analyze this issue in detail.

Consider the extreme case where $\lambda = 1$, reducing the DEP operation to an unbiased max-plus MP layer. Suppose we have an MP-based morphological layer with $n = 256$ inputs, where the weights $w_{ij}$ follow a normal distribution, either at initialization or during training. As assumed in prior works (Glorot & Bengio, 2010; He et al., 2015), let the input $\mathbf{x}$ also follow a normal distribution, independent of the weights. The output of the layer is then given by $y_i = \max_j(x_j + w_{ij})$. Since $x_j$ and $w_{ij}$ are independent zero-mean Gaussian variables with variance 1, their sum follows a Gaussian distribution with zero mean and variance 2. Consequently, the output distribution follows the "maximum of Gaussians" distribution. This results in a shift towards positive values (nonzero mean) and a reduction in variance for large $n$. Specifically, a bound on the mean of the output is given by $\sqrt{4 \log(n)}$ (Sivaraman, 2011). The variance can be estimated using the Fisher–Tippett–Gnedenko theorem (Fisher & Tippett, 1928), though we omit the detailed derivation.

For proper initialization, following the principles of Glorot and Kaiming initialization (Glorot & Bengio, 2010; He et al., 2015), the output should ideally follow a zero-mean normal distribution. However, correcting the output distribution by adjusting the weights is a highly nontrivial inverse problem. At best, one can attempt to correct the mean and variance heuristically. For instance, in our case with $n = 256$, setting the weights to have a mean of $-7.5$ and variance 2.56 approximates a normal distribution, as shown in Figure 4a. However, this process is cumbersome, fragile, and highly sensitive to small deviations of $n$, making it impractical for initializing morphological networks. As a result, the max-plus MP-based layer is effectively unusable in practice, even if activated.

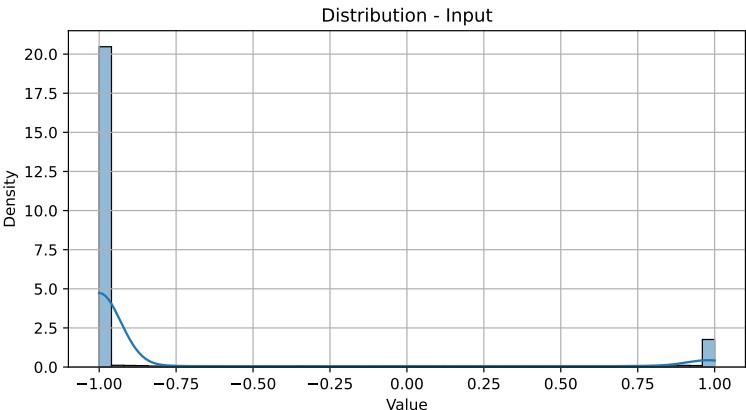

Figure 5: Distribution of MNIST values.

A similar issue arises with DEP-based layers when $\lambda$ is either learnable or fixed at $\lambda \neq 1/2$. When $\lambda$ is fixed but not $1/2$, the weight distributions for the maximum and minimum terms must be initialized differently to ensure a zero-mean output. When $\lambda$ is learnable, even if initialized near $1/2$, any deviation during training forces the layer to adjust the mean of its weights dynamically. Morphological networks, however, are not well suited for such adaptive mean corrections, further compromising their trainability.

On the other hand, setting fixed $\lambda = 1/2$ ensures that the output has zero mean and maintains a relatively stable variance, even for small variations in $n$. In Figure 4b, we illustrate the distribution of the sum of the maximum and minimum, assuming that for sufficiently large $n$, the two terms are approximately uncorrelated. Additionally, the variance can be further adjusted to 1 by appropriately scaling the activations. This makes networks with fixed $\lambda = 1/2$ trainable.

**Initialization in experiments.** In our experiments, we did not ablate the effect of initialization. Rather, we tried to initialize each network properly according to the above remarks and give it as much of an advantage as possible. Specifically:

1. **Max-Plus MP-based Networks:** As discussed earlier, properly initializing this network required setting the weight variance to a value greater than 1 and the mean to a negative value. After extensive trial and error, we found that initializing the weights with a mean of $-5/3$ and a standard deviation of 3 yielded the best results. Therefore, this initialization was used for all MP-based networks in our experiments.

2. **DEP ($\lambda = 3/4$):** Properly initializing this network proved to be impractical. As a result, we used the same initialization as for DEP networks with $\lambda = 1/2$.

3. **DEP (learnable $\lambda$):** The parameter $\lambda$ was initialized from a uniform distribution $U([0, 1])$, which has a mean of $1/2$. Given this, the ideal weight initialization follows the same approach as DEP networks with $\lambda = 1/2$.

4. **DEP ($\lambda = 1/2$) and MPM:** The initialization of networks with our proposed fixed $\lambda = 1/2$ and MPM networks is significantly simpler than that of MP-based networks. All morphological layers are initialized to follow a standard distribution.

We also optimized the initialization of the linear activations (which correct the variance of the output). For fully connected networks of Setting 1, we found that initializing according to a zero-mean Gaussian distribution with standard deviation $1/3.46$ yielded the best results (in general, initialization of activations for Setting 1 does not make significant differences. One could choose to initialize the activations to $1/2$ and obtain good results). For convolutional networks of Setting 1, each convolution kernel was simply initialized

Table 12: Parameter count of all models.

| Network Params. | MLP | MP | DEP | DEP ($\lambda = 1/2$) | Act-MP | Act-DEP |
|---|---|---|---|---|---|---|
| | 466698 | 466698 | 932106 | 930816 | 467978 | 933386 |

| Network Params. | Act-DEP ($\lambda = 3/4$) | Act-DEP ($\lambda = 1/2$) | MPM | RMPM | RMPM-Drop | MPM-SVD |
|---|---|---|---|---|---|---|
| | 932096 | 932096 | 469268 | 469268 | 469268 | 469268 |

| Network Params. | LeNet-5 | MPM-LeNet-5 | MPM-SVD-LeNet-5 | ResNet-20 | MPM-ResNet-20 | |
|---|---|---|---|---|---|---|
| | 61696 | 63304 | 63304 | 271994/272282 | 278378/278666 | |

according to a standard distribution. For Settings 2 and 3, the linear layers were initialized according to Glorot.

Some exceptions: i) For MNIST, the input images do not follow a zero-mean distribution (see Figure 5). For this reason, and to keep the output zero-mean, the first morphological layer of the networks was initialized to zero. ii) For the morphological ResNet-20 networks, all morphological layers were initialized following the standard distribution for all datasets, and the linear activations were initialized to give the average of the max and the min (making the initialization of the morphological ResNet-20 models extremely simple).

## D  Additional experiments

Before we continue with the additional experiments, we provide the details of our experimental setup and compute resources required: All experiments can be reproduced from the provided python notebooks by sequentially running all of their cells. Notebooks "finalMNNs*.ipynb" include the majority of experiments. Notebook "regressionMNNs.ipynb" includes additional experiments (and experiments on regression tasks presented in Appendix C and this Appendix). Notebooks "snip_experiments*.ipynb" include the SNIP pruning experiments. Notebooks "resnet_20_experiments*.ipynb" include the experiments on ResNet-20 models. The GPUs we used were the NVIDIA GeForce RTX 2080 Ti and the RTX 3060, both of which have 12GB of memory. One GPU is sufficient to run each notebook (i.e. 12GB of GPU memory suffice to run each notebook). For notebooks "finalMNNs*.ipynb" and "snip_experiments*.ipynb", each epoch of training took less that 30 seconds, and a whole cell of training took less than 30 minutes (with the exception of RMPM-Drop, which was trained for 200 epochs). The other cells took negligible time to complete. Each "finalMNNs*.ipynb" notebook required about a day to run from start to finish. Each "snip_experiments*.ipynb" notebook took about 6 hours to run. For notebooks "resnet_20_experiments*.ipynb", each epoch of MPM-ResNet-20 training took about 1.5 minutes, longer than the 30 seconds required by the linear ResNet-20. We believe this is due to the inefficiency of our CUDA implementation of max-plus convolution. Each "resnet_20_experiments*.ipynb" took about 6 hours to run.

**Declaration of LLM Usage:** We made use of LLMs, and specifically ChatGPT, as a programming assistant for tasks like writing boilerplate code, code auto-completion, fixing errors and debugging. Any code generated by ChatGPT was rigorously checked for correctness.

**Parameter count of models:** Table 12 presents the parameter counts of all models used in our experiments. For the ResNet-20 models, two numbers appear: one for gray-scale datasets such as F-MNIST, and one for RGB datasets such as CIFAR-10.

Next, we present our additional experimental results. Specifically, we first provide experiments on Hybrid-MLP, which was not studied in the main text. Then, we provide experiments on simple regression tasks, and the full training history of networks that we claimed were trainable but for which only peak training accuracy was reported in the main body.

**Regression experiments.**  Continuing, we present experiments on simple regression tasks. Notice that on classification tasks (like the ones presented in the main body), it is not necessary for the network to be a universal approximator, since we work with logits. This makes regression a more ideal task for showcasing the need for our "linear" activations. We perform regression of simple single-variate single-output functions sampled with zero-mean i.i.d. gaussian noise. To ablate the effect of our "linear" activations, we use 3 fully

connected networks, each with two hidden layers of size 100: 1) a simple ReLU-activated MLP, 2) the MPM network, and 3) a non-activated MPM network. The functions which we sample are the following: 1) $6\sin(x)$, 2) $x^2$, and 3) $20x$. The results are presented in Figures 6a, 6b, 6c. As expected, the non-activated MPM fails to approximate the underlying functions. The MLP and MPM however, achieve a successful regression of the samples.

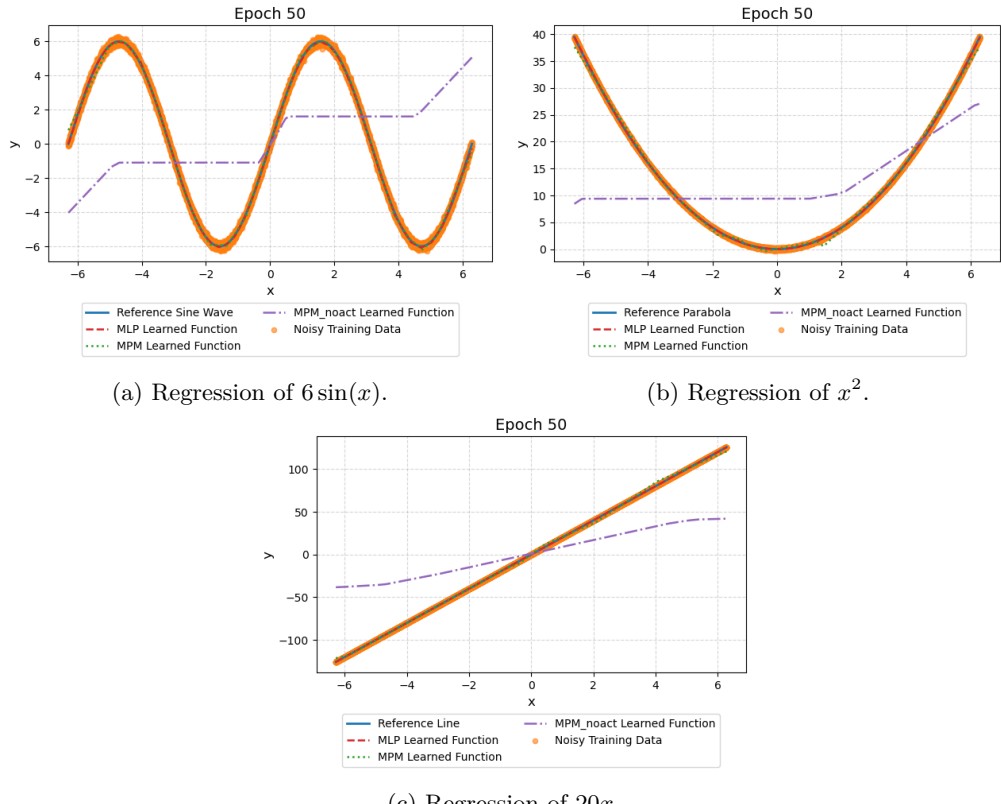

(a) Regression of $6\sin(x)$.  (b) Regression of $x^2$.

(c) Regression of $20x$.

Figure 6: Regression using MLP, MPM, and non-activated MPM on noisy samples for different underlying functions.

**Full training history.**  Finally, we present the full training history of our trainable networks. These results serve as proof that the networks not only achieve satisfactory peak training accuracy, but also manage to converge. In Table 13 we provide the full training history of the networks Act-DEP($\lambda = 1/2$), MPM, RMPM, and MPM-SVD on MNIST, including training accuracy and validation accuracy. In Table 14 we provide the full training history of the networks Act-DEP($\lambda = 1/2$), MPM, RMPM, and MPM-SVD on Fashion-MNIST, including training accuracy and validation accuracy.

Table 13: Full training history, MNIST.

| Epoch | Train | | | | Validation | | | |
|---|---|---|---|---|---|---|---|---|
| | ActDEP($\lambda = 1/2$) | MPM | RMPM | MPM-SVD | ActDEP($\lambda = 1/2$) | MPM | RMPM | MPM-SVD |
| 1 | 56.03 | 85.83 | 88.57 | 90.04 | 56.33 | 85.85 | 87.81 | 89.68 |
| 2 | 84.13 | 91.19 | 92.71 | 94.06 | 84.05 | 90.55 | 91.75 | 92.99 |
| 3 | 88.25 | 93.21 | 94.47 | 95.95 | 87.88 | 92.35 | 93.27 | 94.38 |
| 4 | 90.18 | 94.34 | 95.60 | 97.03 | 89.85 | 93.22 | 94.08 | 94.90 |
| 5 | 91.72 | 95.01 | 96.40 | 97.73 | 91.18 | 93.62 | 94.62 | 95.16 |
| 6 | 92.69 | 95.65 | 96.93 | 98.25 | 91.71 | 94.01 | 94.87 | 95.38 |
| 7 | 93.53 | 96.11 | 97.33 | 98.61 | 92.30 | 94.12 | 94.89 | 95.43 |
| 8 | 94.18 | 96.58 | 97.83 | 98.98 | 92.73 | 94.36 | 95.15 | 95.71 |
| 9 | 94.58 | 96.88 | 98.16 | 98.87 | 93.01 | 94.61 | 95.32 | 95.43 |
| 10 | 95.01 | 97.16 | 98.42 | 99.42 | 93.38 | 94.78 | 95.40 | 95.92 |
| 11 | 95.38 | 97.44 | 98.74 | 99.47 | 93.50 | 94.84 | 95.41 | 95.78 |
| 12 | 95.72 | 97.76 | 98.82 | 99.55 | 93.67 | 94.80 | 95.33 | 95.83 |
| 13 | 96.00 | 97.81 | 99.15 | 99.67 | 93.72 | 94.93 | 95.42 | 95.80 |
| 14 | 96.26 | 98.10 | 99.25 | 99.67 | 93.97 | 95.09 | 95.48 | 95.77 |
| 15 | 96.45 | 98.21 | 99.41 | 99.66 | 94.04 | 94.81 | 95.64 | 95.83 |
| 16 | 96.72 | 98.36 | 99.50 | 99.71 | 94.19 | 94.72 | 95.47 | 95.74 |
| 17 | 96.78 | 98.44 | 99.55 | 99.84 | 94.27 | 94.99 | 95.55 | 95.83 |
| 18 | 97.02 | 98.65 | 99.62 | 99.79 | 94.42 | 94.89 | 95.42 | 95.72 |
| 19 | 97.16 | 98.67 | 99.67 | 99.79 | 94.47 | 94.97 | 95.33 | 95.72 |
| 20 | 97.33 | 98.83 | 99.73 | 99.84 | 94.52 | 95.03 | 95.38 | 95.78 |
| 21 | 97.46 | 98.90 | 99.77 | 99.85 | 94.48 | 95.00 | 95.30 | 95.72 |
| 22 | 97.61 | 99.01 | 99.82 | 99.94 | 94.59 | 95.04 | 95.42 | 95.99 |
| 23 | 97.73 | 99.02 | 99.85 | 99.74 | 94.56 | 95.09 | 95.37 | 95.92 |
| 24 | 97.88 | 99.13 | 99.84 | 99.65 | 94.59 | 95.08 | 95.45 | 95.52 |
| 25 | 98.00 | 99.18 | 99.83 | 99.90 | 94.69 | 95.10 | 95.18 | 95.87 |
| 26 | 98.08 | 99.14 | 99.90 | 99.86 | 94.67 | 94.93 | 95.40 | 95.74 |
| 27 | 98.21 | 99.32 | 99.92 | 99.94 | 94.58 | 95.04 | 95.49 | 96.00 |
| 28 | 98.28 | 99.39 | 99.92 | 99.95 | 94.74 | 94.92 | 95.33 | 96.15 |
| 29 | 98.34 | 99.36 | 99.92 | 99.74 | 94.72 | 94.94 | 95.38 | 95.86 |
| 30 | 98.45 | 99.45 | 99.89 | 99.56 | 94.71 | 94.84 | 95.33 | 95.62 |
| 31 | 98.52 | 99.54 | 99.93 | 99.72 | 94.82 | 94.91 | 95.38 | 95.77 |
| 32 | 98.64 | 99.50 | 99.96 | 99.96 | 94.78 | 94.69 | 95.59 | 96.04 |
| 33 | 98.72 | 99.54 | 99.93 | 99.82 | 94.76 | 94.89 | 95.34 | 95.81 |
| 34 | 98.73 | 99.62 | 99.95 | 99.95 | 94.67 | 94.88 | 95.34 | 96.08 |
| 35 | 98.86 | 99.60 | 99.91 | 99.94 | 94.75 | 94.69 | 95.41 | 96.12 |
| 36 | 98.90 | 99.67 | 99.94 | 99.99 | 94.78 | 94.72 | 95.44 | 96.06 |
| 37 | 98.99 | 99.66 | 99.96 | 99.99 | 94.73 | 94.80 | 95.40 | 96.20 |
| 38 | 99.04 | 99.73 | 99.98 | 99.67 | 94.70 | 94.72 | 95.52 | 95.64 |
| 39 | 99.06 | 99.74 | 99.98 | 99.97 | 94.73 | 94.73 | 95.51 | 96.05 |
| 40 | 99.06 | 99.74 | 99.97 | 99.99 | 94.68 | 94.59 | 95.39 | 96.15 |
| 41 | 99.13 | 99.78 | 99.85 | 99.99 | 94.67 | 94.61 | 95.35 | 96.17 |
| 42 | 99.19 | 99.73 | 99.92 | 99.99 | 94.77 | 94.58 | 95.35 | 96.22 |
| 43 | 99.25 | 99.79 | 99.97 | 99.99 | 94.75 | 94.44 | 95.53 | 96.16 |
| 44 | 99.26 | 99.76 | 99.98 | 99.12 | 94.81 | 94.81 | 95.76 | 95.34 |
| 45 | 99.30 | 99.78 | 99.98 | 99.83 | 94.72 | 94.62 | 95.64 | 96.05 |
| 46 | 99.35 | 99.77 | 99.98 | 99.88 | 94.83 | 94.63 | 95.52 | 96.01 |
| 47 | 99.33 | 99.81 | 99.98 | 99.99 | 94.68 | 94.58 | 95.52 | 96.33 |
| 48 | 99.35 | 99.80 | 99.75 | 99.99 | 94.72 | 94.77 | 95.24 | 96.41 |
| 49 | 99.47 | 99.86 | 99.96 | 99.99 | 94.75 | 94.67 | 95.38 | 96.37 |
| 50 | 99.47 | 99.81 | 99.99 | 99.99 | 94.68 | 94.75 | 95.51 | 96.38 |

Table 14: Full training history, Fashion-MNIST.

| Epoch | Train | | | | Validation | | | |
|---|---|---|---|---|---|---|---|---|
| | ActDEP($\lambda = 1/2$) | MPM | RMPM | MPM-SVD | ActDEP($\lambda = 1/2$) | MPM | RMPM | MPM-SVD |
| 1 | 27.90 | 75.07 | 80.46 | 83.01 | 27.97 | 74.61 | 79.55 | 81.33 |
| 2 | 71.51 | 82.53 | 85.12 | 87.56 | 70.70 | 80.81 | 82.62 | 84.49 |
| 3 | 78.22 | 84.83 | 87.20 | 89.27 | 77.37 | 82.25 | 83.78 | 84.58 |
| 4 | 81.36 | 86.30 | 89.29 | 91.91 | 79.85 | 82.89 | 84.55 | 85.55 |
| 5 | 82.71 | 87.35 | 90.57 | 93.06 | 81.01 | 83.12 | 84.53 | 85.76 |
| 6 | 83.59 | 88.46 | 91.88 | 93.62 | 81.38 | 83.43 | 84.97 | 85.18 |
| 7 | 84.54 | 89.22 | 92.74 | 95.35 | 82.20 | 83.11 | 84.72 | 85.66 |
| 8 | 85.31 | 90.11 | 93.60 | 95.86 | 82.45 | 83.53 | 84.56 | 85.82 |
| 9 | 85.99 | 90.73 | 94.42 | 96.66 | 82.72 | 83.58 | 84.72 | 85.73 |
| 10 | 86.43 | 91.53 | 94.92 | 96.89 | 82.98 | 83.50 | 84.61 | 85.16 |
| 11 | 86.89 | 91.93 | 95.59 | 97.56 | 82.97 | 83.65 | 84.71 | 85.51 |
| 12 | 87.49 | 92.54 | 95.72 | 97.80 | 83.17 | 83.39 | 84.28 | 85.17 |
| 13 | 87.88 | 92.63 | 96.39 | 97.66 | 83.32 | 83.27 | 84.24 | 85.32 |
| 14 | 88.37 | 93.21 | 96.79 | 97.19 | 83.51 | 83.40 | 84.27 | 84.70 |
| 15 | 88.62 | 93.79 | 97.06 | 98.55 | 83.61 | 83.40 | 84.32 | 85.28 |
| 16 | 89.09 | 93.75 | 97.34 | 98.24 | 83.42 | 83.41 | 84.29 | 84.88 |
| 17 | 89.40 | 94.40 | 97.41 | 98.99 | 83.46 | 83.43 | 83.91 | 85.22 |
| 18 | 89.64 | 94.64 | 97.68 | 98.60 | 83.51 | 83.19 | 83.97 | 85.30 |
| 19 | 90.02 | 94.88 | 98.09 | 98.77 | 83.74 | 83.20 | 83.95 | 84.52 |
| 20 | 90.12 | 94.98 | 98.03 | 98.79 | 83.52 | 83.58 | 83.92 | 85.06 |
| 21 | 90.50 | 95.31 | 98.43 | 99.16 | 83.66 | 83.33 | 83.86 | 84.70 |
| 22 | 90.75 | 95.71 | 98.45 | 98.66 | 83.74 | 83.42 | 83.51 | 84.69 |
| 23 | 90.95 | 95.77 | 98.68 | 99.13 | 83.53 | 83.28 | 83.69 | 84.78 |
| 24 | 91.31 | 95.90 | 98.80 | 99.31 | 83.42 | 83.28 | 83.47 | 84.81 |
| 25 | 91.49 | 96.13 | 98.67 | 99.40 | 83.57 | 83.11 | 83.49 | 85.22 |
| 26 | 91.76 | 96.21 | 98.60 | 98.97 | 83.33 | 82.83 | 83.06 | 84.95 |
| 27 | 92.01 | 96.62 | 98.94 | 98.75 | 83.38 | 83.12 | 83.38 | 84.36 |
| 28 | 92.26 | 96.75 | 98.69 | 98.89 | 83.40 | 82.92 | 83.07 | 84.53 |
| 29 | 92.41 | 96.68 | 98.93 | 99.42 | 83.44 | 82.66 | 83.38 | 84.89 |
| 30 | 92.61 | 96.69 | 99.12 | 98.96 | 83.06 | 82.53 | 83.08 | 84.41 |
| 31 | 92.81 | 97.06 | 99.15 | 99.43 | 83.08 | 82.49 | 83.03 | 84.97 |
| 32 | 93.06 | 97.07 | 99.25 | 99.18 | 83.40 | 82.42 | 82.99 | 84.27 |
| 33 | 92.97 | 97.29 | 99.24 | 99.45 | 83.12 | 82.64 | 83.14 | 85.03 |
| 34 | 93.25 | 97.43 | 99.36 | 99.59 | 83.25 | 82.59 | 83.21 | 84.64 |
| 35 | 93.43 | 97.25 | 99.20 | 99.16 | 83.02 | 82.34 | 82.80 | 84.66 |
| 36 | 93.47 | 97.44 | 99.38 | 99.00 | 83.03 | 82.11 | 82.63 | 84.30 |
| 37 | 93.65 | 97.65 | 99.29 | 99.64 | 83.22 | 82.35 | 82.99 | 84.74 |
| 38 | 93.97 | 97.80 | 99.51 | 99.56 | 83.21 | 82.12 | 82.64 | 84.86 |
| 39 | 93.78 | 97.67 | 99.41 | 98.62 | 83.15 | 82.42 | 83.03 | 84.89 |
| 40 | 94.23 | 97.78 | 99.24 | 98.30 | 83.22 | 82.21 | 82.59 | 83.72 |
| 41 | 94.25 | 97.89 | 99.26 | 99.40 | 83.12 | 82.08 | 82.55 | 84.83 |
| 42 | 94.39 | 97.98 | 99.50 | 99.03 | 83.10 | 82.29 | 82.85 | 85.25 |
| 43 | 94.55 | 98.08 | 99.56 | 99.42 | 82.93 | 82.50 | 82.63 | 84.66 |
| 44 | 94.65 | 98.13 | 99.56 | 99.59 | 83.20 | 82.17 | 82.50 | 85.00 |
| 45 | 94.77 | 97.97 | 99.49 | 99.62 | 83.12 | 81.87 | 82.90 | 84.89 |
| 46 | 94.86 | 98.29 | 99.51 | 99.51 | 83.11 | 82.17 | 82.88 | 84.63 |
| 47 | 94.88 | 98.31 | 99.46 | 99.45 | 82.96 | 81.89 | 82.47 | 84.60 |
| 48 | 94.98 | 98.36 | 99.51 | 99.47 | 82.86 | 82.20 | 82.70 | 84.95 |
| 49 | 95.08 | 98.41 | 99.53 | 99.65 | 82.92 | 81.99 | 82.85 | 85.00 |
| 50 | 95.06 | 98.30 | 99.62 | 99.37 | 82.74 | 81.92 | 82.83 | 84.86 |

# E Connection to the Representation Theorem

In this appendix, we highlight the connection between our theoretical results and the *Representation Theorem* of Maragos & Schafer (1987). For completeness, we restate the theorem below.

**Theorem E.1** (Representation Theorem). *Let $h(\mathbf{x}), \mathbf{x} \in \mathbb{Z}^d$ be the finite-extent impulse response of an $m$-dimensional linear shift-invariant filter $\Gamma(f) = f * h$, which is defined on a class $\mathcal{S}$ of real-valued discrete-domain signals $f : \mathbb{Z}^d \to \mathbb{R}$ closed under translation. If $h$ satisfies the following conditions:*

1. *Increasing: $h(\mathbf{x}) \geq 0, \forall \mathbf{x}$,*

2. *Translation-invariant: $\sum_{\mathbf{x}} h(\mathbf{x}) = 1$,*

*then we can represent the linear operator as a supremum of weighted erosions:*

$$\Gamma(f)(\mathbf{x}) = (h * f)(\mathbf{x}) = \bigvee_{g \in \mathrm{Bas}(\Gamma)} \bigwedge_{\mathbf{y} \in \mathbb{Z}^d} f(\mathbf{y}) - g(\mathbf{y} - \mathbf{x}),$$

*where the basis $\mathrm{Bas}(\Gamma)$ is given by*

$$\mathrm{Bas}(\Gamma) = \{g \in \mathcal{S} : \sum_{\mathbf{y} \in spt(h)} h(\mathbf{y})g(-\mathbf{y}) = 0 \ \wedge \ g(-\mathbf{x}) = -\infty \Leftrightarrow h(\mathbf{x}) = 0\}$$

**Example 1.** *If we take $h(n) = \alpha\delta(n) + (1-\alpha)\delta(n-1)$ with $0 < \alpha < 1$, then a corresponding basis function $g \in \mathrm{Bas}(\Gamma)$ takes the form:*

$$g(n) = \begin{cases} r, & n = 0, \\ -\frac{\alpha r}{1-\alpha}, & n = -1, \\ -\infty, & otherwise, \end{cases} \quad for \ r \in \mathbb{R}.$$

*Substituting into the Representation Theorem, we obtain:*

$$\alpha x_n + (1-\alpha)x_{n-1} = \sup_{r \in \mathbb{R}} \left[ \min \left\{ x_n - r, \ x_{n-1} + \frac{\alpha r}{1-\alpha} \right\} \right].$$

*For example, setting $n = 1$ yields the following identity:*

$$\alpha x_1 + (1-\alpha)x_0 = \sup_{r \in \mathbb{R}} \left[ \min \left\{ x_1 - r, \ x_0 + \frac{\alpha r}{1-\alpha} \right\} \right]. \tag{4}$$

*The LHS of this identity is a linear perceptron with weights $\alpha_1 = \alpha, a_0 = 1 - \alpha$ with $\alpha > 0$, i.e. a linear perceptron with positive weights that sum up to 1. Notice that this condition follows from the "Increasing" and "Translation-invariant" hypotheses of the Representation Theorem.*

*More generally, consider a linear perceptron with inputs $x_0, x_1, \ldots, x_n$ and weights $\alpha_0, \ldots, \alpha_n$, where $\alpha_i > 0$ for all $i$ and $\sum_i \alpha_i = 1$. Then, the following identity holds:*

$$\sum_{i=0}^{n} \alpha_i x_i = \sup_{r_0, \ldots, r_{n-1} \in \mathbb{R}} \left[ \min \left\{ x_0 - r_0, \ldots, x_{n-1} - r_{n-1}, \ x_n + \frac{\sum_{i=0}^{n-1} \alpha_i r_i}{\alpha_n} \right\} \right]. \tag{5}$$

*Furthermore, if we relax the normalization constraint and only require the weights to be positive and sum to less than 1, the result still holds if we allow biased erosions. For instance, taking $\alpha = 1/2$ and evaluating equation (4) with $x_0 = 0$ and $x_1 = x$, we find:*

$$\frac{x}{2} = \sup_{r \in \mathbb{R}} \left[ \min \left\{ x - r, \ r \right\} \right].$$

*Similarly, setting $x_n = 0$ in (5) yields the identity:*

$$\sum_{i=0}^{n-1} \alpha_i x_i = \sup_{r_0, \ldots, r_{n-1} \in \mathbb{R}} \left[ \min \left\{ x_0 - r_0, \ldots, x_{n-1} - r_{n-1}, \ \frac{\sum_{i=0}^{n-1} \alpha_i r_i}{1 - \sum_{i=0}^{n-1} \alpha_i} \right\} \right]. \tag{6}$$

*In the above identities we can even allow the weights to be non-negative and they will still hold true: if a weight $\alpha_i = 0$, then we can always take $r_i \to -\infty$ and the $i$-th term of the erosion vanishes.*

**Conclusion:** Based on the above example, we conclude that the Representation Theorem allows us to build any linear perceptron with weights greater than or equal to 0 whose sum is less than or equal to 1, provided we are allowed to take suprema over infinite domains and biased erosions. If the weights sum up to 1, then we use (5); if they sum up to less than 1 then we use (6). We can combine the two forms by writing the generalized operator $y(\mathbf{x}) = \sup_{r \in \mathbb{R}^n} \min_{j \in [n] \cup \{0\}}(x_j + w(j, r))$, where $w(j, r)$ is continuous in terms of $r$, $x_0 = 0$ is fixed, and depending on the case we can either take $w(j, r) = +\infty$, in which case the bias vanishes, or $w(j, r) < +\infty$, in which case we have biased erosions.

It is trivial to see that the inverse of the above also holds true: If we can build any linear perceptron with non-negative weights that sum up to less than or equal to 1, then we can build an increasing, translation-invariant LSI filter.

**Notice that the Representation Theorem states nothing about linear perceptrons with weights that are less than $0$ or that sum up to more than $1$.**

This is where Theorem 1 comes in. There are three main differences between Theorem 1 and the Representation Thoerem:

1. In Theorem 1 suprema and infima are only over finite domains, exactly how they are taken in neural networks. This is a condition for the proof to work.

2. In Theorem 1 we assume the use of max-plus and min-plus MPs, i.e. the weights of the supremum is not inside the previous infimum like they are in the idenities obtained from the Representation Theorem.

3. If we take the limit as the domain tends to be infinite, then Theorem 1 proves the impossibility of representing linear perceptrons with negative weights, or weights that sum up to more that 1.

First, we see how having suprema and infima over finite domains is implicitly used in the proof. This is done in the following two ways:

1. When we have a finite family of functions, with each being differentiable a.e., then they are **simultaneously** differentiable a.e. (i.e. the Lebesgue-measure of the set of points for which at least one of the functions is non-differentiable is zero).

2. When we have a supremum or infimum over a finite domain, then it is definitely attained.

Let us see the above in more detail. In the proof of Theorem 1, we write $x_i^{(n+1)} = \max_j f_{ij}^{(n+1)}$. We note that each $f_{ij}^{(n+1)}$ is differentiable a.e.. Then, because $j$ runs over a finite (and hence countable) set, $f_{ij}^{(n+1)}$ for the different $j$, and $x_i^{(n+1)}$ are simultaneously differentiable a.e.. This allows us to apply Lemma A.4 a.e., etc.

Notice that the same argument can not be used when dealing with suprema and infima over uncountably infinite domains. For example, it is $\frac{x}{2} = \sup_{r \in \mathbb{R}} \min(x - r, r) = \sup_{r \in \mathbb{R}} f_r(x)$. For every $r$ the function $f_r(x)$ is non-differentiable only for $x = 2r$, and hence it is a.e. differentiable. However, it does not hold that $f_r(x)$ are simultaneously differentiable a.e., because for every $x$, there is the function $f_{x/2}$ which is not differentiable on $x$. In fact, for no $x$ are the functions $f_r(x)$ simultaneously differentiable. Hence, Lemma A.4 can never be applied.

The problem with the above is that $\mathbb{R}$ has non-zero measure. The next logical question would be, what if we allow the suprema and infima to be taken over a infinite but countable domain. In fact, due to continuity, we can write $\frac{x}{2} = \sup_{q \in \mathbb{Q}} \min(x - q, q) = \sup_{q \in \mathbb{Q}} f_q(x)$. Then, for every $x \notin \mathbb{Q}$ it holds that $f_q(x)$ are simultaneously differentiable, i.e. they are simultaneously differentiable a.e., and Lemma A.4 can be applied a.e.. However, now the proof stops working for a different reason. Since we are taking a supremum over $\mathbb{Q}$, it is not necessarily attained. In the proof, for the induction to work we implicitly make use of the fact that, because the domain of the suprema and infima is finite, they are attained, and hence the set $J$ is non-empty. This allows the induction to work properly.

Next, we note that despite proving the theorem only in the case of using max-plus and min-plus MPs, it can readily be generalized to also include the case of the weights of the supremum being inside the previous infumum, and vice versa. In other words, if we are given a network defined recursively as $x_i^{(n+1)}(\mathbf{x}) = \max_k \min_j (x_j^{(n)}(\mathbf{x}) + w_{kj}), i \in [N^{(n+1)}]$ and $x_0^{(n)} = 0$ fixed, then using the same Lemmas and an induction argument, it can be shown that Theorem 1 holds also for this new type of network.

Finally, the two theorems seem contradicting. However, they are actually (almost) complementary to each other. In order to see why, we have to go from the finite domains that Theorem 1 works with, to the infinite domains that the Representation Theorem works with. To do so, we need the following lemma.

**Lemma E.2.** *Let $(f_n)$ be a sequence of Lipschitz functions defined on an interval $I$, and suppose that $f_n \to f$ pointwise on $I$. Assume that there exist constants $m, M \in \mathbb{R} \cup \{-\infty, +\infty\}$ such that for all $n \in \mathbb{N}$, it holds a.e. on $I$ that*

$$m \le f_n'(x) \le M.$$

*Then, wherever the derivative $f'(x)$ exists on $I$, it satisfies*

$$m \le f'(x) \le M.$$

*Proof.* Since each $f_n$ is Lipschitz on $I$, it is absolutely continuous on any compact subinterval $[a, b] \subset I$. Furthermore, since $f_n'(x)$ exists almost everywhere and is bounded as $m \le f_n'(x) \le M$, we have for any $a < b$ in $I$:

$$m(b - a) \le \int_a^b f_n'(x)\, dx = f_n(b) - f_n(a) \le M(b - a).$$

Taking the limit as $n \to \infty$ and using the pointwise convergence $f_n \to f$, we obtain:

$$m(b - a) \le f(b) - f(a) \le M(b - a),$$

which implies:

$$m \le \frac{f(b) - f(a)}{b - a} \le M \quad \text{for all } a < b \text{ in } I.$$

Thus, whenever $f'(x)$ exists, it must lie in the interval $[m, M]$. $\qquad\square$

**Example 2.** *Consider the sequence $f_n(x) = \frac{1}{n} \sin(nx)$, defined on an interval $I$ containing $0$. Each $f_n$ is Lipschitz on $I$ and converges pointwise to $f(x) = 0$ for all $x \in \mathbb{R}$. The derivatives are given by $f_n'(x) = \cos(nx)$, which do not converge uniformly.*

*In particular, $f_n'(0) = 1$ for all $n$, while $f'(0) = 0$. Thus, the derivative and limit cannot be interchanged. However, since $-1 \le f_n'(x) \le 1$ for all $x \in I$, Lemma E.2 applies, and we conclude that:*

$$-1 \le f'(x) \le 1 \quad \text{wherever } f'(x) \text{ exists.}$$

*Indeed, since $f(x) \equiv 0$, we have $f'(x) = 0$ everywhere it exists, and the bound is satisfied. Moreover, the bound $[-1, 1]$ is tight no matter how small an interval $I$ we choose around 0. Hence, Lemma E.2 cannot give us a better estimate on $f'(0)$.*

Take a function of the form $y(\mathbf{x}) = \sup_{r \in \mathbb{R}^n} \min_{j \in [n] \cup \{0\}} (x_j + w(j, r))$, like in the identities (5), (6) obtained from the Representation theorem, where $w(j, r)$ is continuous in terms of $r$. Due to continuity, we have that $y(\mathbf{x}) = \sup_{q \in \mathbb{Q}^n} \min_j (x_j + w(j, q))$, and it also holds that $y(\mathbf{x}) = \sup_{i \in \mathbb{N}} \min_j (x_j + w(j, q_i)) = \lim_{n \to \infty} \max_{i \in [n]} \min_j (x_j + w(j, q_i)) = \lim_{n \to \infty} y_n(\mathbf{x})$, where $q_i$ is a counting of the countable $\mathbb{Q}^n$. Indeed, the above holds because $y_n$ is an increasing sequence:

$$y_n = \max_{i \in [n]} \min_j (x_j + w(j, q_i)) \le \max_{i \in [n+1]} \min_j (x_j + w(j, q_i)) = y_{n+1} \Rightarrow$$

$$\lim_{n \to \infty} y_n(\mathbf{x}) = \sup_{n \in \mathbb{N}} y_n(\mathbf{x}) = \sup_{n \in \mathbb{N}} \max_{i \in [n]} \min_j (x_j + w(j, q_i)) = \sup_{i \in \mathbb{N}} \min_j (x_j + w(j, q_i)) = y(\mathbf{x}).$$

For every $y_n$ Theorem 1 holds. Hence for every $n$, $y_n$ is Lipschitz, and a.e. $\nabla y_n(\mathbf{x}) = 0$ or $\nabla y_n(\mathbf{x}) = e_{i_n}$ for some $i_n = i_n(\mathbf{x})$. Hence, a.e. it holds that $\partial_{x_j} y_n(\mathbf{x}) \ge 0$ and $\sum_j (\nabla y_n(\mathbf{x}))_j$ equals either 0 or 1. Now we have the following arguments:

1. Define $g_n(t) = y_n(x_j = t, \mathbf{x}_{-j})$. Since $y_n$ is Lipschitz, so is $g_n$. Moreover, we are given that $\partial_{x_j} y_n(\mathbf{x}) \geq 0$ a.e., which implies that $\frac{dg_n}{dt}(t) \geq 0$ a.e.. Now define $g(t) = y(x_j = t, \mathbf{x}_{-j})$. We have that $g_n \to g$ pointwise, since $y_n \to y$ pointwise. Therefore, we can apply Lemma E.2 to obtain $\frac{dg}{dt}(t) \geq 0$ wherever the derivative exists. Therefore,

$$\partial_{x_j} y(\mathbf{x}) \geq 0,$$

   if the derivative exists.

2. We are given that

$$\sum_j (\nabla y_n(\mathbf{x}))_j = \langle \mathbf{1}, \nabla y_n(\mathbf{x}) \rangle \in \{0, 1\} \quad \Rightarrow \quad \langle \mathbf{1}, \nabla y_n(\mathbf{x}) \rangle \leq 1, \quad \text{a.e.}$$

   Define $g_n(t) = y_n(\mathbf{x} + t\mathbf{1})$ and $g(t) = y(\mathbf{x} + t\mathbf{1})$. Then

$$\frac{dg_n}{dt}(t) = \langle \mathbf{1}, \nabla y_n(\mathbf{x} + t\mathbf{1}) \rangle \leq 1, \quad \text{a.e.}$$

   Since $y_n$ is Lipschitz, so is $g_n$. Since $y_n \to y$ pointwise, it also holds that $g_n \to g$ pointwise. Hence, Lemma E.2 applies, and we have that

$$\frac{dg}{dt}(0) = \langle \mathbf{1}, \nabla y(\mathbf{x}) \rangle \leq 1,$$

   if the derivative exists.

Based on the above arguments, we see that $\nabla y(\mathbf{x}) \succeq 0$ and $\langle \mathbf{1}, \nabla y(\mathbf{x}) \rangle \leq 1$, wherever the derivative exists. If we suppose that $y$ can represent a linear perceptron with weights $\alpha_1, \ldots, \alpha_n$, then it must hold that $\alpha_j = \partial_{x_j} y(\mathbf{x}) \geq 0$ and $\alpha_1 + \ldots + \alpha_n = \langle \mathbf{1}, \nabla y(\mathbf{x}) \rangle \leq 1$. **In other words, a byproduct of (the slight variation of) Theorem 1 is that functions of the form of the Representation Theorem can represent linear perceptrons only if the weights of the perceptrons are non-negative, and their sum is less than or equal to 1.**

