# OpenReview forum: "Training Deep Morphological Neural Networks as Universal Approximators"
_TMLR — Rejected by TMLR_

### Review · Reviewer_sQGh · 2026-04-01

**Summary Of Contributions:**

The paper starts from morphological perceptrons, which are models in which the algebraic operations of standard addition and multiplication are replaced by their tropical counterparts (maximum and addition), and studies how to construct deep morphological neural networks that can be trained on standard image classification tasks (MNIST, FMINST, CIFAR10). To preserve desirable properties such as sparsity and pruneability, a self-imposed constraint on stacking layers is that most learnable parameters shall be part of morphological layers rather than in auxiliary linear components. To address the limitations of purely morphological networks of lacking universal approximation and trainability, the paper introduces “linear” activations, which are coordinate-wise linear scalings or coordinate-wise scalings in a fixed randomly chosen orthonormal basis. A third approach considers a hybrid architecture alternating standard linear with morphological layers.  The paper proves that purely morphological deep networks are not universal approximators and shows that the proposed architectures with “linear” activations recover universal approximation. Empirically, the proposed models can be successfully trained on standard image classification tasks and exhibit strong pruneability.

**Strengths**

+ The paper is very well written and well structured, making the arguments easy to follow.

+ To my knowledge, this is the first work to successfully train deep morphological networks on image classification tasks (MNIST, Fashion-MNIST, and CIFAR-10).

+ The paper provides substantial theoretical contributions, including rigorous proofs. In particular, it shows that common morphological architectures (such as alternating max-plus and min-plus layers, or convex combinations of dilation and erosion) are not universal approximators. On the other hand, the proposed architectures recover universal approximation.

+ The experiments demonstrate strong performance under pruning the networks to a very small number of parameters, for which standard multilayer perceptrons suffer severe performance loss.

**Weaknesses**

- According to the experimental results, deep morphological networks do not generalize as well as standard neural networks. To improve generalization, the paper already applies regularization strategies (dropout and residual connections), but the proposed models are still outperformed by a standard multilayer perceptron. While it could be argued that the study of morphological layers is still in its infancy and that future improvements may close this gap, this raises questions about whether the intended inductive bias of morphological layers effectively translates into improved generalization in practice.

- As a result, the motivation for using morphological layers is not fully convincing from the presented results. These layers are motivated by incorporating operations such as dilation and erosion, which have been identified as useful in pre-deep learning research. However, their introduction here does not appear to lead to improved performance comparable to the gains achieved, for example, by convolutional networks over standard MLPs. It would be interesting to showcase a task on which the intrinsic bias of the proposed architectures aligns better than that of MLPs and leads to improved performance.

- The advantage of hybrid structures over standard networks is not entirely convincing. While the paper shows that, under large batch sizes, gradient descent converges faster for the hybrid network than for traditional ones, this result is restricted to that regime. In practice, practitioners often prefer smaller batch sizes. Moreover, it would be more informative to compare convergence when using individual batch sizes that are well chosen for the respective architectures.

**Audience:**

Yes

**Audience Explanation:**

The paper advances the study of neural network architectures based on morphological layers. While the performance of these (arguably less conventional) morphological networks still lags behind that of highly optimized standard neural networks, the approach is nonetheless valuable. The results are of interest to the community and contribute meaningfully to ongoing research in this direction.

**Broader Impact Concerns:**

No concerns

**Claims And Evidence:**

Yes

**Claims Explanation:**

The paper provides proofs and easy-to-follow arguments supporting its theoretical claims. The empirical results are well supported by the presented experiments.

**Requested Changes:**

All proposed changes are suggestions for minor improvements.

-	Please consider discussing the improvements of the hybrid networks (restricted to the large-batch regime) more carefully.

-	The beginning of section 3 states that  a “common belief is that such a network does not require activations between its layers”. Please clarify more precisely what is hypothesized int the related literature. As currently phrased, it suggests that it hypothesized that a simple stacking of morphological perceptrons might be sufficient for universal approximation, but it appears fairly obvious that this cannot hold (using a simple example calculation with or without tropical algebra).

-	It is well-known that ReLU networks are universal approximators and that their set of functions coincides with the set of tropical rational maps. It appears plausible that the class of functions represented by the proposed architecture also coincides with the set of ReLU network functions. A brief discussion of this relationship would be appreciated

---

> ### Author Response · Authors · 2026-05-15
> **Response to Reviewer sQGh**
>
> We thank the reviewer for the careful reading of the manuscript and for the positive assessment of both the theoretical and empirical contributions. We especially appreciate the recognition that the paper provides rigorous theoretical results while also demonstrating trainable deep morphological networks on standard image classification benchmarks.
>
> Before addressing the specific comments, we would like to note an important correction made during revision. We identified an implementation issue affecting the originally reported fully connected SNIP pruning experiments. After correcting it, we reran the SNIP experiments. The corrected results no longer support the previously stated pruning advantage over linear baselines. Instead, they show that the proposed fully connected morphological architectures remain trainable under aggressive pruning and do not require substantially larger active parameter counts than comparable linear networks. We have updated the abstract, introduction, pruning section, and conclusion accordingly. The theoretical results and the non-pruning experimental conclusions are unaffected.
>
> Regarding the discussion of the large-batch training used for Setting 3, we agree that the previous wording could be interpreted too strongly. The experiments suggest that morphological architectures may be sensitive to gradient noise, with larger batch sizes stabilizing optimization in this setting. We revised the corresponding discussion to reflect this interpretation more carefully and removed wording suggesting a broader convergence advantage.
>
> We also thank the reviewer for pointing out that the phrasing in Section 3 could be made more precise. We revised (and repositioned) the corresponding paragraph to clarify that the issue concerns architectural patterns frequently appearing in the morphological-network literature, namely the composition of morphological operators either directly or with standard non-trainable nonlinear activations. Theorem 3 (in its two forms found in the appendix) covers both of these cases in the case of DEPs and shows that such constructions suffer from expressivity limitations.
>
> We further expanded the discussion regarding the inductive bias implied by Theorem 4. In particular, the proof reduces universal approximation on bounded domains to approximation of piecewise affine functions, allowing comparison with classical maxout constructions. The resulting construction shows that the proposed MPM architecture can emulate the corresponding maxout representations with only an additional logarithmic factor in active parameters. More broadly, the hierarchical construction used suggests an inductive bias in which the slopes of the affine regions are constructed progressively, and intermediate results may be reused across affine regions, potentially reducing the width required in practice despite the additional logarithmic depth factor.
>
> Finally, following the reviewer’s suggestion, we expanded the discussion relating the proposed architectures to tropical geometry and ReLU/maxout networks. Rank-K maxout units correspond to tropical polynomials formed as a maximum of K affine functions, while differences of maxout units correspond to tropical rational mappings. From this viewpoint, ReLU/maxout architectures constrain the number of affine pieces while allowing each affine piece to remain fully expressive, whereas activated morphological layers impose the complementary structural constraint: each affine piece is highly restricted (via the diagonal scaling), but many such pieces participate in the representation. We believe this perspective helps clarify the differences of the proposed architectures relative to more classical piecewise-linear neural networks.

---

> > ### Comment · Reviewer_sQGh · 2026-05-21
> >
> > I thank the authors for their clarifying changes and for transparently reporting the implementation issue and the resulting correction to the pruning experiments.
> >
> > The revised SNIP experiments no longer appear to support an advantage of the proposed architectures over linear baselines under structural pruning. However, Table 5 still appears to show a notable advantage under ℓ1-based masking. Could the authors clarify how they interpret this discrepancy, and to what extent the remaining evidence should still be taken as support for improved compressibility or pruneability of morphological networks?
> >
> > Relatedly, the paper discusses prior work by Dimitriadis & Maragos (2012) as evidence for pruning benefits in morphological networks. Could the authors comment on why those reported gains do not appear to transfer to the deeper morphological architectures evaluated here under the corrected experimental setup?
> >
> > Pruneability appears to be a central empirical motivation for this work, both in the framing and in relation to prior literature. Given that the corrected SNIP results weaken the evidence for pruning benefits, it would be helpful to better understand what empirical advantage remains over standard architectures, especially since the paper does not identify a task where the proposed models clearly outperform conventional baselines.
> >
> > ---
> > On a side note, there is a typo in the new Table 6 where for RMPM the pruning ratio has a constant value

---

> > > ### Author Response · Authors · 2026-06-02
> > > **Response to Official Comment by Reviewer sQGh**
> > >
> > > We thank the reviewer again for the detailed feedback and for the time spent carefully reviewing both the original submission and the revised version.
> > >
> > > We agree that the corrected SNIP results no longer support a pruning advantage of the proposed architectures over linear baselines. For this reason, we revised both the framing and the conclusions of the paper accordingly. The remaining $\ell_1$-masking results should be interpreted more narrowly. We note that SNIP and related pruning methods were originally developed for linear networks, where removing a connection is modeled by setting its weight to zero. In morphological networks, removing a connection does not correspond to setting its weight to zero, and the associated importance scores are not aligned with the underlying pruning objective. To the best of our knowledge, there is currently no broadly adopted pruning method specifically designed for deep morphological networks.
> > >
> > > The original pruning hypothesis was motivated both by prior work reporting positive pruning results in morphological networks and by the theoretical construction underlying Theorem 4. Beyond universal approximation, the theorem suggests that MPM networks can represent complex functions while maintaining sparsity. This motivated our expectation that pruning behavior reported in earlier morphological architectures might extend to MPM networks as well.
> > >
> > > Given the mismatch between the pruning objective and the pruning methods employed, we do not view the current comparison as a definitive assessment of pruning behavior in morphological networks. Rather, the experiments now serve to show that the representational power of the MPMs is not achieved through a prohibitive increase in active parameters.
> > >
> > > A possible distinction from earlier pruning results (such as those of Dimitriadis & Maragos) is that they consider max-plus and min-plus neurons in isolation. In such architectures, there is a natural notion of connection importance, since moving a weight toward the corresponding neutral element ($-\infty$ for max-plus and $+\infty$ for min-plus) progressively removes its influence on the output. MPM units combine max-plus and min-plus components in order to increase representational power, making connection importance less clear-cut and therefore more difficult to capture through existing weight-ordering criteria.
> > >
> > > More broadly, the primary motivation of this work is not pruning. The paper studies the representational capabilities of deep morphological networks and establishes controlled augmentations that yield universal approximation without requiring a substantial increase in active parameters relative to comparable linear constructions. The proposed architectures further provide an empirical demonstration that deep morphological networks can be trained successfully in settings where the contribution of linear operations is deliberately restricted. As discussed in the revised manuscript, MPM units can emulate max-plus and min-plus neurons and, under suitable restrictions, reduce to AND/OR logical operations over subsets of their inputs. Investigating whether this inductive bias is beneficial for tasks involving structured logical or rule-based relationships is outside the scope of the present work, and is not experimentally evaluated.
> > >
> > > We also thank the reviewer for pointing out the typo in Table 6. The ratios should have been in 1-to-1 correspondence.

---

### Review · Reviewer_pAKt · 2026-04-02

**Summary Of Contributions:**

The authors study variants of morphological networks and hybrid version of morphological modules within MLPs and ConvNets. The authors start with several negative results on pure morphological architectures failing to approximate even fairly simple functions, setting up the idea that these nonlinearities should be used as components in a more traditional almost universal architecture instead. Afterwards, the authors showed that with hybrid architectures with only morphological non-linearities can indeed approximate universally. Finally, the paper ended with a number of experimental results on MNIST and CIFAR-10, as well as pruning abilities, which I find most interesting. I personally think the pruning abilities is the most attractive aspect of this architecture.

**Additional Comments:**

I should say that I have an overall positive impression of this submission, but I do want to mention a few of my own opinions that feels subjective but do not affect the decision of this paper.

Firstly, I don't find universal approximation results are particularly interesting. Surely negative results are crucial, but I believe most architectures can approximate all continuous functions on bounded domains given diverging number of parameters. However, the more interesting results are always about training dynamics, which universal approximation theory fails to address.

Secondly, I want to reiterate that MNIST experiments, while accessible to academic level of computing resources, are no longer reliable benchmarks beyond being toy examples. Unfortunately, if everything works on MNIST, then it's no longer effective at proving the potential of a new method. Arguably even CIFAR-10 has similar issues, but are at least more reliable. This is why I would like to see more extended experiments on CIFAR-10, and ideally even Tiny ImageNet if possible.

Finally, I believe the pruning ability is the most interesting part of morphological architectures. Perhaps this should be highlighted early on in the paper in a Figure environment. E.g. converting Table 2 to a figure would be nice, albeit on CIFAR would be better than MNIST, but I understand there's a performance gap. This would be much more attractive to read in my opinion.

**Audience:**

Yes

**Audience Explanation:**

While this paper does not get me too excited about morphological architectures, I believe it could have a place at TMLR as a paper that is demonstrating some interesting trade-offs. One is that the authors showed clearly strong pruning abilities (albeit on MNIST with MLPs only), also the worst performance on CIFAR-10 compared to ResNet-20. It appears that large batches are also required for training, which is a bit unfortunate at the moment, but I believe this could be potentially improved in future iterations. Also I should say negative results are also interesting, which deserves to be part of the literature.

**Claims And Evidence:**

Yes

**Claims Explanation:**

Most of the results match the claims made by the paper, with proofs and code supporting the results. However, I will point out some minor issues in the paper.

**Requested Changes:**

There are a few minor issues that I found, which all contain fairly straight forward fixes. Broadly speaking, I believe several results only work for bounded domains, which the authors should clarify. However this is not really problematic in my opinion, given most universal approximation results only work on bounded domains anyways.

Specifically, Definition A.0 suggested universal approximation works on $\mathbb{R}^n$ whenever the domain is not specified. However, Theorem 4 and Theorem 5's special case result should be clarified that these are also for bounded domains.

Furthermore, I believe the authors do not have theoretical results for setting 2, unless I'm reading this correctly. I would like the authors to either provide a proof for this setting, or simply remove the claims regarding all three constraint settings "accompanied by universal approximability results."

I would also like to ask the authors to provide some results on pruning for the ConvNet results on CIFAR-10. There's a sense that MNIST is too easy of a dataset, and I would like to see more concrete experiments on more challenging datasets for the pruning results to be more convincing. Perhaps the gap of test error to ResNet-20 might even be reduced as the network gets pruned. Ideally, I would like to see the authors expand on the experiments beyond MNIST, more CIFAR-10, and ideally even on Tiny ImageNet.

---

> ### Author Response · Authors · 2026-05-15
> **Response to Reviewer pAKt**
>
> We thank the reviewer for the careful reading of the manuscript and for the constructive feedback. We are encouraged by the reviewer’s positive assessment of the work and by the comment that the negative results are valuable for understanding the limitations of deep morphological architectures.
>
> Before addressing the specific comments, we would like to note an important correction made during revision. We identified an implementation issue affecting the originally reported fully connected SNIP pruning experiments. After correcting it, we reran the SNIP experiments. The corrected results no longer support the previously stated pruning advantage over linear baselines. Instead, they show that the proposed fully connected morphological architectures remain trainable under aggressive pruning and do not require substantially larger active parameter counts than comparable linear networks. This also directly relates to the reviewer’s concern that the universal approximation results may only hold with diverging parameter counts. We have updated the abstract, introduction, pruning section, and conclusion accordingly. The theoretical results and the non-pruning experimental conclusions are unaffected.
>
> We have also addressed the reviewer’s requested clarifications. First, we clarified throughout the paper that the universal approximation results are stated on bounded domains. In particular, we updated the paragraph surrounding Definition A.0 and explicitly clarified that the theorems are formulated over bounded domains.
>
> Second, the reviewer is correct that the original wording could be read as claiming universal approximation results for all three settings. In setting 2, the fixed orthonormal matrices are not controlled by us, and they are given non-deterministically. We have clarified the contributions to state more precisely that universal approximation results are given for the deterministic settings. We have also included a Corollary extending MPMs universal approximation result to the RMPMs.
>
> Regarding the request for pruning experiments on convolutional architectures and more challenging datasets, we agree that such experiments would be valuable. In the revised version, however, given the correction of the SNIP experiments described above, we chose to avoid expanding the pruning claims and instead restricted the discussion to the corrected fully connected setting. We agree that extending the pruning analysis to CIFAR-10, Tiny ImageNet, and convolutional architectures is an important direction for future work.
>
> Finally, we appreciate the reviewer’s broader comment that optimization and training dynamics are often more informative than universal approximation alone. We agree, and this is why the paper combines negative approximation results, sparse-gradient statements, trainability experiments, and large-batch optimization observations. We have revised the framing to emphasize that the theoretical results motivate the architectural modifications, while the experiments test whether these modified architectures can be trained and used at non-prohibitive parameter scales in practice.

---

### Review · Reviewer_yyYV · 2026-05-01

**Summary Of Contributions:**

The paper first studies the limitations of deep morphological neural networks, proving that simply stacking morphological layers is insufficient to approximate any function. This may seem surprising at first glance, as morphological operations are inherently non-linear (due to max and min operators).

The authors then propose three families of hybrid architectures that mix morphological and linear layers. They proved that this strategy makes the resulting networks universal approximators.

Finally, a well-designed empirical evaluation is provided on the common MNIST and Fashion-MNIST datasets. An interesting aspect of the proposed architectures is that they show to be more prunable than standard MLPs.

**Additional Comments:**

I think the 1D toy examples in the Appendix (Figure 6) are helpful for understanding the results, so I suggest referring to them in the main paper.

**Audience:**

Yes

**Audience Explanation:**

As far as I can say, the paper presents original advances for understanding the possibilities and limitations of (deep) morphological neural networks.

Also, for a reader non-knowledgeable of morphological neural networks, like myself, it is intellectually stimulating to be exposed to the fact that well-established rules governing classical deep neural networks vary in other settings (e.g., the effects of linear and non-linear activations). It fosters out-of-the-box thinking that may inspire new ideas in the future.

**Broader Impact Concerns:**

-

**Claims And Evidence:**

Yes

**Claims Explanation:**

First, I must state that I was unfamiliar with Morphological Neural Networks before reviewing this manuscript. Therefore, I can't be certain that the results are completely new, but I have no reason to doubt, given the clear overview of the literature.

Also, I haven't proofread the appendix, but the obtained mathematical results make a lot of sense. Reading the main paper, I could see why stacked morphological layers can be rewritten as a "shallow" multilayer perceptron, and how introducing linearities can enhance their representational power.

I appreciate that the results are not overstated. Both the theoretical and the empirical results are presented honestly. The experiment's description contains the relevant training details.

**Requested Changes:**

I think the paper is well-written, but the notation could be more explicitly defined
- The meaning of $N(n)$ should be explicated more clearly since the beginning of Section 3 (I think $N^{(n)}$ would be more consistent with the rest of the notation). It would also help to state the size of the weight matrix $\mathbf{W}^{(n)}$.
- Each network formula should be immediately preceded or followed by the complete list of learnable parameters and fixed ones. Currently, one must read back and forth to get the complete picture.
- The first equation of page 6 confuses me:  I think indices $i$ should be removed from $\mathbf{w}\_i^{(n)}$ and $\mathbf{m}\_i^{(n)},$ and $\textrm{diag}(\alpha_i^{(n)})$ should be $\textrm{diag}(\alpha_1^{(n)}, \ldots, \alpha_{N(n)}^{(n)}$). If I'm wrong, the notation still needs to be disambiguated!
- In Setting 3, what are the sizes of matrices $\mathbf{A}^{n}$ and vectors $\mathbf{b}^{n}$? Please also restate that they are learnable.

---

> ### Author Response · Authors · 2026-05-15
> **Response to Reviewer yyYV**
>
> We thank the reviewer for the careful reading of the manuscript and for the positive assessment of both the theoretical and empirical contributions. We especially appreciate the reviewer’s comments regarding the clarity and honesty of the presentation.
>
> Before addressing the specific comments, we would like to note an important correction made during revision. We identified an implementation issue affecting the originally reported fully connected SNIP pruning experiments. After correcting it, we reran the SNIP experiments. The corrected results no longer support the previously stated pruning advantage over linear baselines. Instead, they show that the proposed fully connected morphological architectures remain trainable under aggressive pruning and do not require substantially larger active parameter counts than comparable linear networks. We have updated the abstract, introduction, pruning section, and conclusion accordingly. The theoretical results and the non-pruning experimental conclusions are unaffected.
>
> We have addressed the requested changes as follows:
>
> * We clarified the notation used in Section 3. We now explicitly state the dimensions of the corresponding weight matrices and vectors. We also made the notation more consistent throughout the manuscript.
> * For each architectural setting, we now explicitly state which quantities are learnable parameters and which are fixed, together with their corresponding dimensions.
> * We corrected the notation in the first equation on page 6, including the indexing ambiguity pointed out by the reviewer. The index of the biases should be 0 instead of i, and we also corrected the diagonal matrix.
> * In Setting 3, we now explicitly specify the dimensions of the matrices and vectors involved and restate that they are learnable parameters.
>
> Following the reviewer’s suggestion, we added a reference in the main text to the 1D toy examples in the appendix (Figure 6), as they provide useful intuition regarding the theoretical results.

---

### Decision · Action_Editor_f2Ed · 2026-06-09

**Recommendation:** Reject

**Additional Comments:**

Deep morphological networks change standard algebra to the max-plus or min-plus operations. The key potential benefit is compressibility.

The paper advances the field by proposing sound modifications that increase expressivity. The results are significantly better on MNIST, Fashion-MNIST and CIFAR-10.

On the theoretical side, the paper proves properties of different proposed variants.

The key weakness is that despite those improvements morphological networks still do not show practical utility of morphological networks. The gap to a regular network is very large on CIFAR-10 (20%). It is also unclear how big the gap is in the case of alternatives such as binary or ternary networks or FP4 training.

I agree with Reviewers that the paper is a valuable contribution that fosters out-of-box thinking.

However, I believe that for the paper to be interesting for the audience of TMLR, larger changes are required than customary at the camera ready stage. Specifically,

1. Please add a Figure 1. It is hard to understand the morphological MLP from equations alone for a reader without any prior exposition to the field. The image could simply compare normal MLP with morphological MLP, but feel free to take any route that communicates the main idea of the paper visually to the reader.
2. Please expand on the benefit of morphological networks. The brief mention of compressibility should be expanded. Please bring in also background in binary/ternary networks/FP4 networks.
3. The experiments should compare to binary/ternary/FP4 networks (one of those at least).

I believe these changes are essential to make the paper interesting to TMLR audience. There is a too small community interested in just morphological networks, and comparison to adjacent fields is essential to interest those audiences as well.

The paper is invited to be resubmitted once these and other remarks made by Reviewers are addressed. Thank you for the submission, and I hope the comments will be helpful in improving the paper.

**Audience:**

No

**Audience Explanation:**

It would if the paper compared better to adjacent fields (see detailed comments later).

**Claims And Evidence:**

Yes

**Claims Explanation:**

The experiments clearly show improved expressivity and generalization of the proposed variants.

**Resubmission Of Major Revision:**

The authors may consider submitting a major revision at a later time.